# DETECTING MISCLASSIFICATION ERRORS IN NEURAL NETWORKS WITH A GAUSSIAN PROCESS MODEL

## ABSTRACT

As neural network classifiers are deployed in real-world applications, it is crucial that their predictions are not just accurate, but trustworthy as well. One practical solution is to assign confidence scores to each prediction, then filter out low-confidence predictions. However, existing confidence metrics are not yet sufficiently reliable for this role. This paper presents a new framework that produces more reliable confidence scores for detecting misclassification errors. This framework, RED, calibrates the classifier's inherent confidence indicators and estimates uncertainty of the calibrated confidence scores using Gaussian Processes. Empirical comparisons with other confidence estimation methods on 125 UCI datasets demonstrate that this approach is effective. An experiment on a vision task with a large deep learning architecture further confirms that the method can scale up, and a case study involving out-of-distribution and adversarial samples shows potential of the proposed method to improve robustness of neural network classifiers more broadly in the future.

## 1 INTRODUCTION

Classifiers based on Neural Networks (NNs) are widely deployed in many real-world applications (LeCun et al., 2015; Anjos et al., 2015; Alghoul et al., 2018; Shahid et al., 2019). Although good prediction accuracies are achieved, lack of safety guarantees becomes a severe issue when NNs are applied to safety-critical domains, e.g., healthcare (Selişteanu et al., 2018; Gupta et al., 2007; Shahid et al., 2019), finance (Dixon et al., 2017), self-driving (Janai et al., 2017; Hecker et al., 2018), etc.

One way to estimate trustworthiness of a classifier prediction is to use its inherent confidence-related score, e.g., the maximum class probability (Hendrycks & Gimpel, 2017), entropy of the softmax outputs (Williams & Renals, 1997), or difference between the highest and second highest activation outputs (Monteith & Martinez, 2010). However, these scores are unreliable and may even be misleading: high-confidence but erroneous predictions are frequently observed (Provost et al., 1998; Guo et al., 2017; Nguyen et al., 2015; Goodfellow et al., 2014; Amodei et al., 2016). In a practical setting, it is beneficial to have a detector that can raise a red flag whenever the predictions are suspicious. A human observer can then evaluate such predictions, making the classification system safer.

In order to construct such a detector, quantitative metrics for measuring predictive reliability under different circumstances are first developed, and a warning threshold then be set based on users' preferred precision-recall tradeoff. Existing such methods can be categorized into three types based on their focus: error detection, which aims to detect the natural misclassifications made by the classifier (Hendrycks & Gimpel, 2017; Jiang et al., 2018; Corbière et al., 2019); out-of-distribution (OOD) detection, which reports samples that are from different distributions compared to training data (Liang et al., 2018; Lee et al., 2018a; Devries & Taylor, 2018); and adversarial sample detection, which filters out samples from adversarial attacks (Lee et al., 2018b; Wang et al., 2019; Aigrain & Detyniecki, 2019).

Among these categories, error detection, also called misclassification detection (Jiang et al., 2018) or failure prediction (Corbière et al., 2019), is the most challenging (Aigrain & Detyniecki, 2019) and underexplored. For instance, Hendrycks & Gimpel (2017) defined a baseline based on maximum class probability after softmax layer. Although the baseline performs reasonably well in most testing cases, reduced efficacy in some scenaria indicates room for improvement (Hendrycks & Gimpel,

2017). Jiang et al. (2018) proposed Trust Score, which measures the similarity between the original classifier and a modified nearest-neighbor classifier. The main limitation of this method is scalability: the Trust Score may provide no or negative improvement over the baseline for high-dimensional data. ConfidNet (Corbière et al., 2019) builds a separate NN model to learn the true class probablity, i.e. softmax probability for the ground-truth class. However, ConfidNet itself is a standard NN, so its confidence scores may be unreliable or misleading: A random input may generate a random confidence score, and ConfidNet do not provide any information regarding uncertainty of these confidence scores. Moreover, none of these methods can differentiate natural classifier errors from risks caused by OOD or adversarial samples; if a detector could do that, it would be easier for practitioners to fix the problem, e.g., by retraining the original classifier or applying better preprocessing techniques to filter out OOD or adversarial data.

To meet these challenges, a new framework is developed in this paper for error detection in NN classifiers. The main idea is to utilize RIO (Residual, Input, Output; Qiu et al., 2020), a regression model based on Gaussian Processes, on top of the original NN classifier. Whereas the original RIO is limited to regression problems, the proposed approach extends it to misclassification detection by modifying its components. The new framework not only produces a calibrated confidence score based on the original maximum class probability, but also provides a quantitative uncertainty estimation of that score. Errors can therefore be detected more reliably. Note that the proposed method does not change the prediction accuracy of the original classification model. Instead, it provides a quantitative metric that makes it possible to detect misclassification errors. This framework, referred to as RED (RIO for Error Detection), is compared empirically to existing approaches on 125 UCI datasets and on a large-scale deep learning architecture. The results demonstrate that the approach is effective and robust. A further case study with OOD and adversarial samples shows the potential of using RED to diagnose the sources of mistakes as well, thereby leading to a possible comprehensive approach for improving trustworthiness of neural network classifiers in the future.

## 2 RELATED WORK

In the past two decades, a large volume of work was devoted to calibrating the confidence scores returned by classifiers. Early works include Platt Scaling (Platt, 1999; Niculescu-Mizil & Caruana, 2005), histogram binning (Zadrozny & Elkan, 2001), isotonic regression (Zadrozny & Elkan, 2002), with recent extensions like Temperature Scaling (Guo et al., 2017), Dirichlet calibration (Kull et al., 2019), and distance-based learning from errors (Xing et al., 2020). These methods focus on reducing the difference between reported class probability and true accuracy, and generally the rankings of samples are preserved after calibration. As a result, the separability between correct and incorrect predictions is not improved. In contrast, RED aims at deriving a score that can differentiate incorrect predictions from correct ones better.

A related direction of work is the development of classifiers with rejection/abstention option. These approaches either introduce new training pipelines/loss functions (Bartlett & Wegkamp, 2008; Yuan & Wegkamp, 2010; Cortes et al., 2016), or define mechanisms for learning rejection thresholds under certain risk levels (Dubuisson & Masson, 1993; Santos-Pereira & Pires, 2005; Chow, 2006; Geifman & El-Yaniv, 2017). In contrast to these methods, RED assumes an existing pretrained NN classifier, and provides an additional metric for detecting potential errors made by this classifier, without specifying a rejection threshold.

Designing metrics for detecting potential risks in NN classifiers has also become popular recently. While most approaches focus on detecting OOD (Liang et al., 2018; Lee et al., 2018a; Devries & Taylor, 2018) or adversarial examples (Lee et al., 2018b; Wang et al., 2019; Aigrain & Detyniecki, 2019), work on detecting natural errors, i.e., regular misclassifications not caused by external sources, is more limited. Ortega (1995) and Koppel & Engelson (1996) conducted early work in predicting whether a classifier is going to make mistakes, and Seewald & Fürnkranz (2001) built a meta-grading classifier based on similar ideas. However, these early works did not consider NN classifiers. More recently, Hendrycks & Gimpel (2017) and Haldimann et al. (2019) demonstrated that raw maximum class probability is an effective baseline in error detection, although its performance was reduced in some scenaria.

More elaborate techniques for error detection have also been developed recently. Mandelbaum & Weinshall (2017) proposed a confidence score based on the data embedding derived from the penul-

timate layer of a NN. However, their approach requires modifying the training procedure in order to achieve effective embeddings. Jiang et al. (2018) introduced Trust Score to measure the similarity between a base classifier and a modified nearest-neighbor classifier. Trust Score outperforms the maximum class probability baseline in many cases, but negative improvement over baseline can be observed in high-dimensional problems, implying poor scalability of local distance computations. ConfidNet (Corbière et al., 2019) learns to predict the class probability of true class with another NN, while Introspection-Net (Aigrain & Detyniecki, 2019) utilizes the logit activations of the original NN classifier to predict its correctness. Since both models themselves are standard NNs, the confidence scores returned by them may be arbitrarily high without any uncertainty information. Moreover, existing approaches for error detection cannot differentiate natural misclassification error from OOD or adversarial samples, making it difficult to diagnose the sources of risks. In contrast, RED explicitly reports its uncertainty about the estimated confidence score, providing more reliable error detection. The uncertainty information returned by RED may also be helpful in clarifying the cause of classifier mistakes, as will be demonstrated in this paper.

## 3 METHODOLOGY

This section gives the general problem statement, introduces the basic idea of original RIO, on which RED is built, and describes the technical details of RED.

### 3.1 PROBLEM STATEMENT

Consider a training dataset $\mathcal{D} = (\mathcal{X}, \mathbf{y}) = \{(\mathbf{x}_i, y_i)\}_{i=1}^N$, and a pretrained NN classifier that outputs a predicted label $\hat{y}_i$ and class probabilities for each class $\sigma_i = [\hat{p}_{i,1}, \hat{p}_{i,2}, \ldots, \hat{p}_{i,K}]$ given $\mathbf{x}_i$, where $N$ is the total number of training points and $K$ is the total number of classes. The problem is to develop a metric that can serve as a quantitative indicator for detecting natural misclassification errors made by the pretrained NN classifier.

### 3.2 RIO

The original RIO (Qiu et al., 2020) was developed to quantify point-prediction uncertainty in regression models. More specifically, RIO fits a Gaussian Process (GP) to predict the residuals, i.e. the differences between ground-truth and original model predictions. It utilizes an I/O kernel, i.e. a composite of an input kernel and an output kernel, thus taking into account both inputs and outputs of the original regression model. As a result, it measures the covariances between data points in both the original feature space and the original model output space. For each new data point, a trained RIO model takes the original input and output of the base regression model, and predicts a distribution of the residual, which can be added back to the original model prediction to obtain both a calibrated prediction and the corresponding predictive uncertainty.

In the original RIO work, SVGP (Hensman et al., 2013; 2015) was used as an approximate GP to improve the scalability of the approach. Both empirical results and theoretical analysis showed that RIO is able to consistently improve the prediction accuracy of base model as well as provide reliable uncertainty estimation. Moreover, RIO can be directly applied on top of any pre-trained models without retraining or modification. It therefore forms a promising foundation for improving reliability of error detection metrics as well.

### 3.3 RIO FOR ERROR DETECTION (RED)

Although RIO performs robustly in a wide variety of regression problems, it cannot be directly applied to classification models. A new framework, namely RED, is proposed to utilize RIO for error detection in classification domains.

Building on the fact that the original maximum class probability is a strong baseline for error detection (Hendrycks & Gimpel, 2017; Haldimann et al., 2019), the main idea of RED is to derive a more reliable confidence score by stacking RIO on top of the original maximum class probability. Since RIO was designed for single-output regression problems, it contains an output kernel only for scalar outputs. In RED, this original output kernel is extended to multiple outputs, i.e. to vector outputs

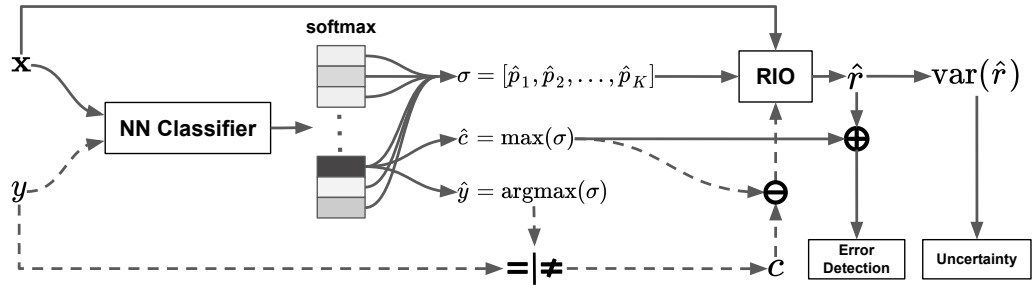

Figure 1: **The RED Training and Deployment Processes.** The solid pathways are active in both training and deployment phase, while the dashed pathways are active only in the training phase. During the training phase, a target confidence score $c$ is assigned to each training sample according to whether it is correctly predicted by the original NN classifier or not. A RIO model is then trained to predict the residual between the target confidence score $c$ and the original maximum class probability $\hat{c}$. The I/O kernel in RIO utilizes both the raw feature $\mathbf{x}$ and softmax outputs $\sigma$ to predict the residuals. In the deployment phase, given a new data point, the trained RIO model provides a Gaussian distribution of estimated residual $\hat{r}$ defined by the mean $\bar{\hat{r}}$ and variance $\mathrm{var}(\hat{r})$. Addition of $\hat{r}$ and $\hat{c}$ forms a calibrated confidence score for error detection, and $\mathrm{var}(\hat{r})$ indicates the corresponding uncertainty.

---

**Algorithm 1** RED Training and Deployment Procedures

---

**Require:**

   $(\mathcal{X}, \mathbf{y}) = \{(\mathbf{x}_i, y_i)\}_{i=1}^N$: training data

   $\hat{\mathbf{y}} = \{\hat{y}_i\}_{i=1}^N$: labels predicted by original NN classifier on training data

   $\boldsymbol{\sigma} = \{\sigma_i = [\hat{p}_{i,1}, \hat{p}_{i,2}, \ldots, \hat{p}_{i,K}]\}_{i=1}^N$: softmax outputs of original NN classifier on training data

   $\hat{\mathbf{c}} = \{\hat{c}_i = \max(\sigma_i)\}_{i=1}^N$: maximum class probability returned by original NN classifier on training data

   $\mathbf{x}_*$: data to be predicted

   $\sigma_*$: softmax outputs of original NN classifier on $\mathbf{x}_*$

   $\hat{c}_*$: maximum class probability returned by original NN classifier on $\mathbf{x}_*$

**Ensure:**

   $\hat{c}_*' \sim \mathcal{N}(\hat{c}_* + \bar{\hat{r}}_*, \mathrm{var}(\hat{r}_*))$: $\hat{c}_* + \bar{\hat{r}}_*$ can be used as confidence score for error detection, and $\mathrm{var}(\hat{r}_*)$ represents the uncertainty of returned confidence score

**Training Phase:**

1: obtain target confidence score $\mathbf{c} = \{c_i = \delta_{y_i, \hat{y}_i}\}_{i=1}^N$, where $\delta_{y_i, \hat{y}_i}$ is the Kronecker delta ($\delta_{y_i, \hat{y}_i} = 1$ if $y_i = \hat{y}_i$, otherwise $\delta_{y_i, \hat{y}_i} = 0$)

2: calculate residuals $\mathbf{r} = \{r_i = c_i - \hat{c}_i\}_{i=1}^N$

3: **for** each optimizer step **do**

4:    calculate covariance matrix $\mathbf{K}_c((\mathcal{X}, \boldsymbol{\sigma}), (\mathcal{X}, \boldsymbol{\sigma}))$, where each entry is given by $k_c((\mathbf{x}_i, \sigma_i), (\mathbf{x}_j, \sigma_j)) = k_{\mathrm{in}}(\mathbf{x}_i, \mathbf{x}_j) + k_{\mathrm{out}}(\sigma_i, \sigma_j)$, for $i, j = 1, 2, \ldots, N$

5:    optimize GP hyperparameters by maximizing log marginal likelihood $\log p(\mathbf{r}|\mathcal{X}, \boldsymbol{\sigma}) = -\frac{1}{2}\mathbf{r}^\top(\mathbf{K}_c((\mathcal{X}, \boldsymbol{\sigma}), (\mathcal{X}, \boldsymbol{\sigma})) + \sigma_n^2 \mathbf{I})^{-1}\mathbf{r} - \frac{1}{2}\log|\mathbf{K}_c((\mathcal{X}, \boldsymbol{\sigma}), (\mathcal{X}, \boldsymbol{\sigma})) + \sigma_n^2 \mathbf{I}| - \frac{n}{2}\log 2\pi$

**Deployment Phase:**

6: calculate residual mean $\bar{\hat{r}}_* = \mathbf{k}_*^\top(\mathbf{K}_c((\mathcal{X}, \boldsymbol{\sigma}), (\mathcal{X}, \boldsymbol{\sigma})) + \sigma_n^2 \mathbf{I})^{-1}\mathbf{r}$ and residual variance $\mathrm{var}(\hat{r}_*) = k_c((\mathbf{x}_*, \sigma_*), (\mathbf{x}_*, \sigma_*)) - \mathbf{k}_*^\top(\mathbf{K}_c((\mathcal{X}, \boldsymbol{\sigma}), (\mathcal{X}, \boldsymbol{\sigma})) + \sigma_n^2 \mathbf{I})^{-1}\mathbf{k}_*$, where $\mathbf{k}_*$ denotes the vector of kernel-based covariances (i.e., $k_c(\mathbf{x}_*, \mathbf{x}_i)$) between $\mathbf{x}_*$ and all the training points

7: return distribution of calibrated confidence score $\hat{c}_*' \sim \mathcal{N}(\hat{c}_* + \bar{\hat{r}}_*, \mathrm{var}(\hat{r}_*))$

---

such as those of the final softmax layer of a NN classifer, representing estimated class probabilities for each class. This modification allows RIO to access more information from the classifier outputs.

The targets for RIO training need to be redesigned as well. The raw targets of a classification problem are the ground-truth labels; they are in categorical space, while RIO works in continuous space. To solve this issue, RED constructs a different problem: Instead of predicting the labels directly, it predicts whether the original prediction is correct or not. A target confidence score is

assigned to each training data point accordingly. The residual between this target confidence score and the originally returned maximum class probability is calculated, and a RIO model is trained to predict these residuals. Given a new data point, the trained RIO model combined with the original NN classifier thus provides a calibrated confidence score for detecting misclassification errors.

Figure 1 illustrates the RED training and deployment processes conceptually, and Algorithm 1 specifies them in detail. In the training phase, the first step is to define a target confidence score $c_i$ for each training sample $(\mathbf{x}_i, y_i)$. For simplicity, all training samples that are correctly predicted by the original NN classifier receive 1 as the target confidence score, and those that are incorrectly predicted receive 0. The validation dataset during the original NN training is included in the training dataset for RED. After the target confidence scores are assigned, a regression problem is formulated for the RIO model: Given the original raw features $\{\mathbf{x}_i\}_{i=1}^N$ and the corresponding softmax outputs of the original NN classifier $\{\sigma_i = [\hat{p}_{i,1}, \hat{p}_{i,2}, \ldots, \hat{p}_{i,K}]\}_{i=1}^N$, predict the residuals $\mathbf{r} = \{r_i = c_i - \hat{c}_i\}_{i=1}^n$ between target confidence scores $\mathbf{c} = \{c_i\}_{i=1}^N$ and the original maximum class probabilities $\hat{\mathbf{c}} = \{\hat{c}_i = \max(\sigma_i)\}_{i=1}^N$.

The RIO model relies on an I/O kernel consisting of two components: the input kernel $k_{\text{in}}(\mathbf{x}_i, \mathbf{x}_j)$, which measures covariances in the raw feature space, and the modified multi-output kernel $k_{\text{out}}(\sigma_i, \sigma_j)$, which calculates covariances in the softmax output space. The hyperparameters of the I/O kernel are optimized to maximize the log marginal likelihood $\log p(\mathbf{r}|\mathcal{X}, \boldsymbol{\sigma})$. In the deployment phase, given a new data point $\mathbf{x}_*$, the trained RIO model provides a Gaussian distribution for the estimated residual $\hat{r}_* \sim \mathcal{N}(\bar{\hat{r}}_*, \text{var}(\hat{r}_*))$. By adding the estimated residual back to the original maximum class probability $\hat{c}_*$, a distribution of calibrated confidence score is obtained as $\hat{c}'_* \sim \mathcal{N}(\hat{c}_* + \bar{\hat{r}}_*, \text{var}(\hat{r}_*))$. The mean $\hat{c}_* + \bar{\hat{r}}_*$ can be directly used as a quantitative metric for error detection, and the variance $\text{var}(\hat{r}_*)$ represents the corresponding uncertainty of the confidence score.

## 4 EMPIRICAL EVALUATION

In this section, the error detection performance of RED is evaluated comprehensively on 125 UCI datasets, comparing it to other related methods. Its generality is then evaluated by applying it to two other base models, and its scale-up properties measured in a larger deep learning architecture and in a larger task. Finally, RED's potential to improve robustness more broadly is demonstrated in a case study focusing on OOD and adversarial samples.

### 4.1 COMPARISONS WITH RELATED APPROACHES

As a comprehensive evaluation of RED, an empirical comparison with seven related approaches on 125 UCI datasets (Dua & Graff, 2017) was performed. These approaches include maximum class probability baseline (MCP; Hendrycks & Gimpel, 2017), three state-of-the-art approaches, namely Trust Score (Jiang et al., 2018), ConfidNet (Corbière et al., 2019), and Introspection-Net (Aigrain & Detyniecki, 2019), as well as three earlier approaches, i.e. entropy of the original softmax outputs (Steinhardt & Liang, 2016), DNGO (Snoek et al., 2015), and the original SVGP (Hensman et al., 2013; 2015). The 125 UCI datasets include 121 datasets used by Klambauer et al. (2017) and four more recent ones. Full details about the datasets and parametric setup of all tested algorithms, and a downloadable link for source codes are provided in Appendix A.1.

Following the experimental setup of Hendrycks & Gimpel (2017); Corbière et al. (2019); Aigrain & Detyniecki (2019), the task for each algorithm is to provide a confidence score for each testing point. An error detector can then use a predefined fixed threshold on this score to decide which points are probably misclassified by the original NN classifier. For RED, the mean of calibrated confidence score $\hat{c}_* + \bar{\hat{r}}_*$ was used as the reported confidence score.

Five threshold-independent performance metrics were used to compare the methods: AUPR-Error, which computes the area under the Precision-Recall (AUPR) Curve when treating incorrect predictions as positive class during the detection; AUPR-Success, which is similar to AUPR-Error but uses correct predictions as positive class; AUROC, which computes the area under receiver operating characteristic (ROC) curve for the error detection task; AP-Error, which computes the average precision (AP) under different thresholds treating incorrect predictions as positive class; and AP-Success, which is similar to AP-Error but uses correct predictions as positive class. AUPR and AUROC are commonly used in prior work (Hendrycks & Gimpel, 2017; Corbière et al., 2019; Aigrain & De-

Table 1: Mean Rank on UCI datasets

| Method | AP-Error mean±std | AUPR-Error mean±std | AP-Success mean±std | AUPR-Success mean±std | AUROC mean±std |
|---|---|---|---|---|---|
| RED | **1.39±0.61**\* | **1.49±0.78**\* | **1.74±0.97**\* | **1.80±1.03**\* | **1.65±0.82**\* |
| MCP Baseline | 2.93± 0.89 | 3.06±0.92 | 2.77±1.07 | 2.75±1.11 | 2.80±1.08 |
| Trust Score | 3.92±2.45 | 3.86±2.50 | 3.64±2.25 | 3.61±2.25 | 3.76±2.31 |
| ConfidNet | 6.13±1.37 | 6.33±1.38 | 6.07±1.51 | 6.07±1.41 | 5.97±1.45 |
| Introspection-Net | 5.34±1.65 | 5.38±1.65 | 5.83±1.46 | 5.89±1.51 | 5.71±1.50 |
| Entropy | 3.47±1.08 | 3.59±1.19 | 3.19±1.26 | 3.23±1.32 | 3.26±1.28 |
| DNGO | 6.19±1.51 | 5.46±1.82 | 6.84±1.33 | 6.80±1.44 | 6.57±1.47 |
| SVGP | 6.59±1.60 | 6.80±1.49 | 5.89±1.54 | 5.83±1.49 | 6.24±1.61 |

The symbols \* indicates that the differences between the marked entry and all other counterparts are statistically significant at the 5% significance level for both paired $t$-test and Wilcoxon test. The best entries that are significantly better than all the others under both statistical test are marked in boldface.

Table 2: A Pairwise Comparison between RED and Other Methods on UCI datasets

| Method | AP-Error + / = / - | AUPR-Error + / = / - | AP-Success + / = / - | AUPR-Success + / = / - | AUROC + / = / - |
|---|---|---|---|---|---|
| MCP Baseline | 87 / 35 / 0 | 90 / 32 / 0 | 58 / 63 / 1 | 56 / 65 / 1 | 61 / 60 / 1 |
| Trust Score | 53 / 44 / 16 | 49 / 47 / 17 | 50 / 47 / 16 | 48 / 49 / 16 | 59 / 37 / 17 |
| ConfidNet | 100 / 22 / 0 | 100 / 22 / 0 | 106 / 16 / 0 | 106 / 16 / 0 | 109 / 13 / 0 |
| Introspection-Net | 93 / 29 / 0 | 90 / 32 / 0 | 98 / 24 / 0 | 98 / 24 / 0 | 101 / 21 / 0 |
| Entropy | 74 / 47 / 1 | 75 / 46 / 1 | 53 / 68 / 1 | 53 / 68 / 1 | 52 / 69 / 1 |
| DNGO | 92 / 17 / 0 | 73 / 31 / 5 | 99 / 10 / 0 | 97 / 12 / 0 | 98 / 11 / 0 |
| SVGP | 98 / 23 / 1 | 98 / 23 / 1 | 97 / 25 / 0 | 97 / 25 / 0 | 102 / 19 / 1 |

The columns labeled + show the number of datasets on which RED performs significantly better at the 5% significance level in a paired $t$-test, Wilcoxon test, or both; those labeled - represent the contrary case; those labeled = represent no statistical significance.

tyniecki, 2019), but as discussed by Davis & Goadrich (2006) and Flach & Kull (2015), AUPR may provide overly-optimistic measurement of performance. To compensate for this issue, AP-Error and AP-Success are included as additional metrics. Since the target for the confidence metrics is to detect misclassification errors, the following discussion will focus more on AP-Error and AUPR-Error.

Ten independent runs were conducted for each dataset. During each run, the dataset was randomly split into training dataset and testing dataset, and a standard NN classifier trained and evaluated on them. The same dataset split and trained NN classifier was used to evaluate all methods. Full details of the experimental setup are provided in Appendix A.1.

Table 1 shows the ranks of each algorithm averaged over all 125 UCI datasets. The rank of each algorithm on each dataset is based on the average performance over 10 independent runs. RED performs best on all metrics; the performance differences between RED and all other methods are statistically significant under paired $t$-test and Wilcoxon test. Trust Score has the highest standard deviation, suggesting that its performance varies significantly across different datasets.

As a more detailed comparison, Table 2 shows how often RED performs statistically significantly better, how often the performance is not significantly different, and how often it performs significantly worse than the other methods. RED is most often significantly better, and very rarely worse. In a handful of datasets Trust Score is better, but most often it is not.

To illustrate the performance of RED further compared to the baseline and the three state-of-the-art approaches, Figure 2 shows the distribution of the relative rank of RED, MCP baseline, Trust Score, ConfidNet and Introspection-Net as a function of the number of samples and the number of features in the dataset. These plots are based on the AP-Error metric; other metrics provide similar results. RED performs consistently well over different dataset sizes and feature dimensionalities. Trust Score performs best in several datasets, but occasionally also worst in both small and large datasets, making it a rather unreliable choice. ConfidNet generally exhibits worse performance on datasets with large dataset sizes and high feature dimensionalities, i.e. it does not scale well to larger problems.

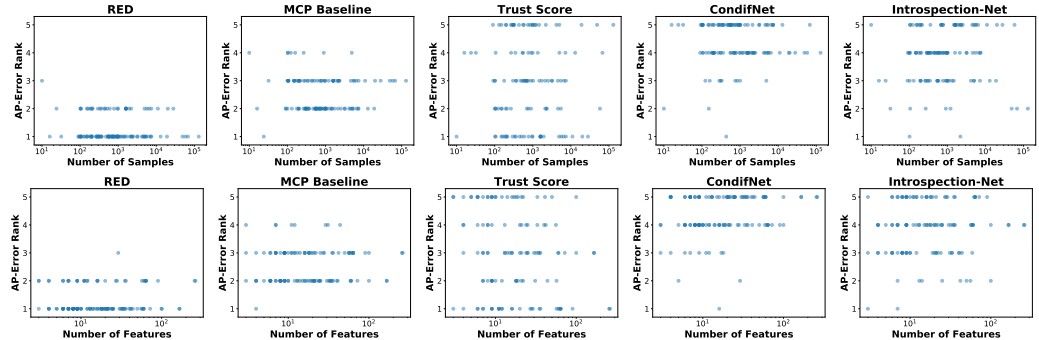

Figure 2: **Performance Ranks Across Dataset Sizes and Dimensionalities on UCI Datasets.**
Each plot represents the distribution of relative ranks for one method (each column) as a function
of the dataset size (top row) and the feature dimensionality (bottom row). Each dot in each plot
represents the relative rank in one dataset. RED performs consistently well over datasets of different
sizes and dimensionalities. Trust Score performs inconsistently, and ConfidNet performs poorly on
larger datasets.

Table 3: A Pairwise Comparison between RED and Other Methods on UCI datasets

| Method | AP-Error | AUPR-Error | AP-Success | AUPR-Success | AUROC |
| | + / = / - | + / = / - | + / = / - | + / = / - | + / = / - |
| --- | --- | --- | --- | --- | --- |
| BNN MCP | 102 / 20 / 0 | 104 / 18 / 0 | 95 / 26 / 1 | 88 / 33 / 1 | 95 / 26 / 1 |
| BNN Entropy | 67 / 53 / 2 | 68 / 52 / 2 | 48 / 66 / 8 | 48 / 66 / 8 | 53 / 64 / 5 |
| MC-Dropout MCP | 87 / 35 / 0 | 88 / 34 / 0 | 70 / 52 / 0 | 67 / 55 / 0 | 71 / 51 / 0 |
| MC-Dropout Entropy | 54 / 68 / 0 | 55 / 67 / 0 | 38 / 77 / 7 | 38 / 76 / 8 | 42 / 74 / 6 |

The columns labeled + show the number of datasets on which RED performs significantly better at the 5% significance level in a paired $t$-test, Wilcoxon test, or both; those labeled - represent the contrary case; those labeled = represent no statistical significance.

## 4.2 GENERALITY WRT. BASE MODELS

To evaluate generality and robustness of RED, it was applied to two other base models: an NN
classifier using MC-dropout technique (Gal & Ghahramani, 2016) and a Bayesian Neural Network
(BNN) classifier (Wen et al., 2018). They were each trained as base classifiers, and RED was
then applied to each of them (implementation details are provided in Appendix A.1). Experiments
analogous to those in Table 2 were performed on 125 UCI datasets in both cases. Table 3 shows
the pairwise comparisons between RED and the internal confidence scores returned by the base
models. MCP and Entropy represent the maximum class probability and entropy of softmax outputs,
respectively, after averaging over 100 test-time samplings. RED significantly improves MC-dropout
and BNN classifier in most datasets, demonstrating that it is a general technique that can be applied
to a variety of models.

## 4.3 SCALING UP TO LARGER ARCHITECTURES

To confirm that the RED approach scales up to large deep learning architectures, a VGG16 model
(Simonyan & Zisserman, 2015) was trained on the CIFAR-10 dataset using a state-of-the-art training
pipeline (see Appendix A.2 for details). In order to remove the influence of feature extraction in
image preprocessing and to make the comparison fair, all approaches used the same logit outputs of
the trained VGG16 model as their input features. 10 independent runs are performed. During each
run, a VGG16 model is trained, and all the methods are evaluated based on this VGG16 model.

Table 4 shows the results on the two main error detection performance metrics (note that the table
lists absolute values instead of rankings along each metric). Trust Score performs much better than
in previous literatures (Corbière et al., 2019). This difference may be due to the fact that logit outputs
are used as input features here, whereas Corbière et al. (2019) utilized a higher dimensional feature
space for Trust Score. Based on the results, RED significantly outperforms all the counterparts in
both metrics. This result demonstrates the advantages of RED in scaling up to larger architectures.

Table 4: A Comparison based on the VGG16 Network Architecture on the CIFAR-10 Task

| Metric | RED mean±std | MCP Baseline mean±std | Trust Score mean±std | ConfidNet mean±std | Introspection-Net mean±std | Entropy mean±std | DNGO mean±std | SVGP mean±std |
|---|---|---|---|---|---|---|---|---|
| AP-Error(%) | **49.88±1.99*** | 47.09±2.19 | 48.76±2.28 | 45.80±2.85 | 42.11±1.98 | 47.91±2.17 | 33.91±2.94 | 40.71±2.33 |
| AUPR-Error(%) | **49.79±2.00*** | 46.99±2.21 | 48.68±2.29 | 45.70±2.86 | 42.01±1.98 | 47.81±2.19 | 34.43±2.92 | 40.60±2.34 |

The symbols * indicates that the differences between the marked entry and all other counterparts are statistically significant at the 5% significance level for both paired $t$-test and Wilcoxon test. The best entries that are significantly better than all the others under both statistical test are marked in boldface.

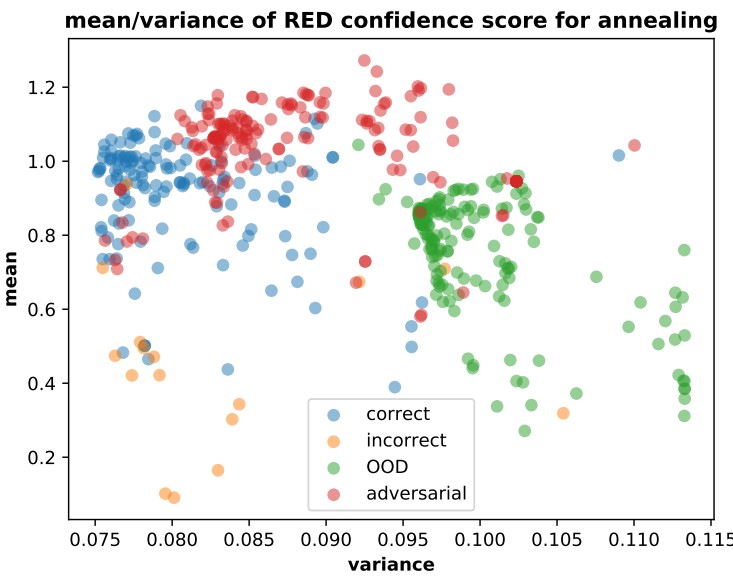

Figure 3: **Identifying OOD and Adversarial Samples Based on Mean and Variance of Confidence Scores.** Each dot represents one sample in the testing set in the UCI Annealing task. The horizontal axis denotes the variance of RED-calibrated confidence score, and the vertical axis denotes the mean. If an in-distribution sample is correctly classified by original NN classifier, it is marked as "correct", otherwise it is marked "incorrect". Mean is a good separator of correct and incorrect classifications. High variance, on the other hand, indicates that RED is uncertain about its confidence score, which can be used to identify OOD and adversarial samples. In this manner, RED can serve as a foundation for improving robustness of classifiers more broadly in the future.

## 4.4 A CASE STUDY WITH OOD AND ADVERSARIAL SAMPLES

In all experiments so far, the mean of calibrated confidence score $\hat{c}_* + \bar{\hat{r}}_*$ was used as RED's confidence score. Although good performance is observed in error detection by only using the mean, the variance of calibrated confidence score $\mathrm{var}(\hat{r}_*)$ may be helpful if the scenario is more complex, e.g., the dataset includes some OOD data, or even adversarial data.

A preliminary investigation of RED in such scenaria was performed by manually adding OOD and adversarial data into the test set of the UCI "annealing" dataset, on which RED detected errors well. The synthetic OOD and adversarial samples were created to be highly deceptive, aiming to evaluate the performance of RED under difficult circumstances. The OOD data were sampled from a Gaussian distribution with mean 0 and variance 1, and the number of added OOD data was the same as the number of samples in the original test set. Note that all data points from original dataset are normalized to have mean 0 and variance 1 for each feature dimension during preprocessing, so the OOD data and in-distribution data have similar scales. The adversarial data was created by adding negligible modifications to training samples that the original NN classifier predicted incorrectly with highest confidence. This process mimics an adversary that can arbitrarily alter the output of the NN classifier with minuscule changes to the input (Goodfellow et al., 2014).

Figure 3 shows the distribution of mean and variance of calibrated confidence scores for testing samples, including correctly and incorrectly labeled actual samples, as well as the synthetic OOD and adversarial samples. The mean is a good separator for correctly classified and incorrectly classified samples, which tend to cluster on the top and bottom half of the image, respectively. On the

other hand, variance is a promising indicator of OOD and adversarial samples. RED's confidence scores of in-distribution samples have low variance because they covary with the training samples. The variance thus represents RED's confidence in its confidence score. Samples with large variance indicate that RED is uncertain about its confidence score, which can be used as a basis for detecting OOD and adversarial samples.

Thus, although the main focus of this paper is to demonstrate RED on misclassification detection, the preliminary results in this subsection show that it provides a promising foundation for detecting other error types as well.

## 5 DISCUSSION AND FUTURE WORK

One interesting observation from the experiments is that RED almost never performs worse than the MCP baseline. This result suggests that there is almost no risk in applying RED on top of an existing NN classifier. Since RED is based on a GP model, the estimated residual $\hat{r}_*$ is close to zero if the predicted sample is far from the distribution of the original training samples, resulting in no change to the original MCP. In other words, RED does not make random changes to original MCP if it is very uncertain about the predicted sample, and this uncertainty is explicitly represented in the variance of the estimated confidence score. This property makes RED a particularly reliable technique for error detection.

Another interesting observation is that the variance is also helpful in detecting OOD and adversarial samples. This result follows from the design of the RIO uncertainty model. Since RIO in RED has an input kernel and an output kernel, lower estimated variance requires that the predicted sample is close to training samples in both the input feature space and the classifier output space. This requirement is difficult for OOD and adversarial attacks to achieve, providing a basis for detecting them.

In a real-world deployment, it is necessary to define a threshold for triggering error warning based on RED's confidence scores. A practical way is to use a validation dataset to determine how the precision/recall tradeoff changes over different thresholds. The user can then select a threshold based on their preference.

The most compelling direction of future work is to extend this capability of RED further. Instead of using a single dimensional confidence score for error detection, it is possible to use mean and variance simultaneously, leading to a two dimensional detection space. Further separation between OOD and adversarial samples may be possible by adding one more dimension, such as the ratio between input kernel output and output kernel output. Also, instead of using a hard target confidence score (i.e. either 0 or 1), it may be possible to define a soft target confidence that may be more informative. Further, RED may be used to calibrate other existing confidence metrics, such as the Trust Score, which may lead to a further improvement in detection performance.

## 6 CONCLUSION

This paper introduced a new framework, RED, for error detection in neural network classifiers that can produce a more reliable confidence score than previous methods. RED is able to not only provide a calibrated confidence score, but also report the uncertainty of the estimated confidence score. Experimental results show that RED's scores consistently outperform state-of-the-art methods in separating the misclassified samples from correctly classified samples. Preliminary experiments also demonstrate that the approach scales up to large deep learning architectures, and can form a basis for detecting OOD and adversarial samples as well. It is therefore a promising foundation for improving robustness of neural network classifiers.

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

# A APPENDIX

## A.1 EXPERIMENTAL SETUP FOR SECTION 4.1 AND SECTION 4.2

**General Information** 10 independent runs are performed for each dataset. During each run, the dataset is randomly split into a training set ($80\%$) and a testing set ($20\%$), then a fully connected feed-forward NN classifier with 2 hidden layers, each with 64 hidden neurons, are trained on the training set. The activation function is ReLU for all the hidden layers. The maximum number of epochs for training is 1000. $20\%$ of the training set is used as validation set, and the split is random at each independent run. An early stop is triggered if the loss on validation set has not be improved for 10 epochs. The optimizer is Adam with learning rate 0.001, $\beta_1 = 0.9$, and $\beta_2 = 0.999$. The loss function is cross entropy loss. During each independent run, the same random dataset split and trained base NN classifier is used for evaluating all algorithms. All source codes for reproducing the experimental results can be downloaded from (https://drive.google.com/drive/folders/1X5R6sEkjmucR7B4MvF-rY6QnCvos6a7n?usp=sharing).

**Dataset Description** In total, 125 UCI datasets are used in the experiments, among which 121 are from Klambauer et al. (2017), and 4 are recent datasets released in Dua & Graff (2017). All features in all datasets are normalized to have mean 0 and standard deviation 1. Full details regarding the number of samples N, number of features M, and number of classes K for each dataset are shown in Table 5.

**Parametric Setup for Algorithms**

- RED: SVGP (Hensman et al., 2013; 2015) is used as an approximator to original GP. The number of inducing points are 50. RBF kernel is used for both input and multi-output kernel. Automatic Relevance Determination (ARD) feature is turned on. The signal variances and length scales of all the kernels plus the noise variance are the trainable hyperparameters. The optimizer is L-BFGS-B with default parameters as in Scipy.optimize documentation (https://docs.scipy.org/doc/scipy/reference/optimize.minimize-lbfgsb.html), and the

maximum number of iterations is set as 1000. The optimization process runs until the L-BFGS-B optimizer decides to stop. To overcome the sensitivity of GP optimization to initialization of the hyperparameters (Ulapane et al., 2020), 20 random initialization of the hyperparameters are tried for each independent run. For each random initialization, the signal variances are generated from a uniform distribution within interval $[0, 1]$, and the lengthscales are generated from a uniform distribution within interval $[0, 10]$. For 10 initializations, the hyperparameters of input kernel are first optimized while the multi-output kernel is temporarily turned off, then after the optimizer stops, the multi-output kernel is turned on, and both the two kernels are optimized simultaneously. For the other 10 initializations, both kernels are optimized simultaneouly from the start. The average performance of the 3 best optimized model in terms of corresponding metrics are used as the final performance of RED on each independent run. During our preliminary investigation, several statistic metrics on training set is effective in picking the true best-performing model out of these 20 trials, e.g., the gap between average estimated confidence scores of correctly classified training samples and incorrectly classified training samples, the scale of optimized noise variance of SVGP model, the ratio between sum of signal variances and noise variance after optimization, etc. Since improving initialization and optimization of GP hyperparameters is out of the scope of this work, we simply use average performance of the best 3 models (top $15\%$) in comparison.

- MCP baseline: The maximum class probability of softmax outputs of the base NN classifier is used as the confidence score of MCP baseline. The setup of the base NN classifier is provided above.

- Trust Score: k=10, $\alpha = 0$, without filtering. This is the same as the default setup in https://github.com/google/TrustScore.

- ConfidNet: During training, the input to ConfidNet is the raw feature, and the target is the class probability of the ground-truth class returned by base NN classifier. The architecture of ConfidNet is a fully connected feed-forward NN regressor with 2 hidden layers, each with 64 hidden neurons. The activation function is ReLU for all the hidden layers. The maximum number of epochs for training is 1000. An early stop is triggered if the loss on validation data has not be improved for 10 epochs. The optimizer is RMSprop with learning rate 0.001, and the loss function is mean squared error (MSE).

- Introspection-Net: During training, the input to Introspection-Net is the logit outputs of base NN classifier, and the target is 1 for correctly classified sample, and 0 for incorrectly classified sample. The architecture of ConfidNet is a fully connected feed-forward NN regressor with 2 hidden layers, each with 64 hidden neurons. The activation function is ReLU for all the hidden layers. The maximum number of epochs for training is 1000. An early stop is triggered if the loss on validation data has not be improved for 10 epochs. The optimizer is RMSprop with learning rate 0.001, and the loss function is mean squared error (MSE).

- Entropy: The entropy of softmax outputs of the base NN classifier is used as the confidence score of Entropy. THe setup of the base NN classifier is provided above.

- DNGO: A Bayesian linear regression layer similar to that of Snoek et al. (2015) is added after the logits layer of the original NN classifier to predict whether an original prediction is correct or not (1 for correct and 0 for incorrect). Default parametric setup, as in https://github.com/automl/pybnn/blob/master/pybnn/dngo.py, is used.

- SVGP: The original SVGP without output kernel is used to predict directly whether a prediction made by the base NN classifier is correct or not (1 for correct and 0 for incorrect). All other parameters are identical to those in RED.

- BNN MCP: The standard dense layers in the base NN classifier described in RED setup above is replaced with Flipout layers (Wen et al., 2018). All other parameters are identical with those in RED. The maximum class probability averaging over 100 test-time samplings is used as the confidence score for error detection.

- BNN Entropy: The same setup as with BNN MCP, except now the entropy of softmax outputs averaging over 100 test-time samplings is used as the confidence score for error detection.

- MC-Dropout MCP: A dropout layer with dropout rate of 0.5 is added after each dense layer of the base NN classifier described in the RED setup. All other parameters are identical with those in RED. The maximum class probability averaging over 100 test-time Monte-Carlo samplings is used as the confidence score for error detection.

- MC-Dropout Entropy: The same setup as with MC-Dropout MCP, except now the entropy of softmax outputs is averaged over 100 test-time Monte-Carlo samplings and used as confidence score for error detection.

## A.2 EXPERIMENTAL SETUP FOR SECTION 4.3

**Setup of VGG16 Training** The standard VGG16 architecture (Simonyan & Zisserman, 2015) is used. The training pipeline is based on the default setup described in https://github.com/geifmany/cifar-vgg/blob/master/cifar10vgg.py. For the CIFAR-10 dataset, 40,000 samples are used as the training set, 10,000 as the validation set, and 10,000 as the testing set.

**Parametric Setup for Algorithms** All algorithms use the logit outputs of the trained VGG16 model as input features. The maximum class probability of softmax outputs of the trained VGG16 model is used as the confidence score of MCP baseline. The parameters for RED, Trust Score, Entropy, DNGO and SVGP are identical to those in the UCI experiments. For ConfidNet and Introspection-Net, all parameters are the same as in the UCI experiments, except for that the number of hidden neurons for all hidden layers is increased to 128.

## A.3 DETAILED RESULTS FOR SECTION 4.1 AND SECTION 4.2

This subsection shows all the detailed results for experiments performed in Section 4.1 and Section 4.2. The results are averaged over 10 independent runs in terms of AP-Error, AP-Success, AUPR-Error, AUPR-Success, and AUROC. Detailed results for Section 4.1 are shown in Table 5, Table 6, Table 7, Table 8, and Table9. Detailed results for Section 4.2 are shown in Table 10, Table 11, Table 12, Table 13, and Table14. Each table is corresponding to one performance metric. The column "N", "M", and "K" denotes the number of samples, number of features, and number of classes for corresponding datasets. To save space, "MCP", "Intro-Net", "MC-D" stands for "MCP Baseline", "Introspection-Net", and "MC-Dropout", respectively. For datasets that the original NN classifier achieves 100% accuracy, the entries are marked as "NA". For dataset splits that the number of samples in one particular class is too small for Trust Score to calculate neighborhood distance, the entries are marked as "NA".

Table 5: Comparison between RED and Counterparts Using AP-Error

| Dataset | N | M | K | RED | MCP | Trust Score | ConfidNet | Intro-Net | Entropy | DNGO | SVGP |
|---|---|---|---|---|---|---|---|---|---|---|---|
| abalone | 4177 | 8 | 3 | 0.546 | 0.539 | 0.480 | 0.518 | 0.528 | 0.529 | 0.496 | 0.482 |
| acute-inflammation | 120 | 6 | 2 | NA | NA | NA | NA | NA | NA | NA | NA |
| acute-nephritis | 120 | 6 | 2 | NA | NA | NA | NA | NA | NA | NA | NA |
| adult | 48842 | 14 | 2 | 0.419 | 0.411 | 0.328 | 0.391 | 0.412 | 0.411 | 0.338 | 0.358 |
| annealing | 898 | 31 | 5 | 0.558 | 0.392 | 0.453 | 0.350 | 0.399 | 0.368 | 0.314 | 0.324 |
| arrhythmia | 452 | 262 | 13 | 0.583 | 0.580 | 0.618 | 0.409 | 0.413 | 0.560 | 0.426 | 0.475 |
| audiology-std | 196 | 59 | 18 | 0.779 | 0.721 | NA | 0.428 | 0.518 | 0.731 | 0.590 | 0.556 |
| balance-scale | 625 | 4 | 3 | 0.888 | 0.816 | 0.540 | 0.046 | 0.155 | 0.730 | 0.319 | 0.128 |
| balloons | 16 | 4 | 2 | 0.840 | 0.806 | 0.704 | 0.694 | 0.806 | 0.806 | 0.615 | 0.583 |
| bank | 4521 | 16 | 2 | 0.397 | 0.393 | 0.390 | 0.372 | 0.339 | 0.393 | 0.108 | 0.383 |
| bioconcentration | 779 | 9 | 3 | 0.493 | 0.470 | 0.510 | 0.426 | 0.461 | 0.440 | 0.408 | 0.385 |
| blood | 748 | 4 | 2 | 0.390 | 0.389 | 0.272 | 0.382 | 0.372 | 0.389 | 0.319 | 0.362 |
| breast-cancer | 286 | 9 | 2 | 0.532 | 0.503 | 0.381 | 0.467 | 0.431 | 0.503 | 0.441 | 0.471 |
| breast-cancer-wisc | 699 | 9 | 2 | 0.423 | 0.393 | 0.405 | 0.313 | 0.245 | 0.393 | 0.175 | 0.152 |
| breast-cancer-wisc-diag | 569 | 30 | 2 | 0.626 | 0.599 | 0.500 | 0.089 | 0.374 | 0.601 | 0.089 | 0.119 |
| breast-cancer-wisc-prog | 198 | 33 | 2 | 0.408 | 0.403 | 0.296 | 0.343 | 0.373 | 0.403 | 0.288 | 0.266 |
| breast-tissue | 106 | 9 | 6 | 0.813 | 0.743 | 0.800 | 0.696 | 0.601 | 0.749 | 0.744 | 0.722 |
| car | 1728 | 6 | 4 | 0.571 | 0.462 | 0.256 | 0.140 | 0.053 | 0.457 | 0.117 | 0.098 |
| cardiotocography-10clases | 2126 | 21 | 10 | 0.513 | 0.492 | 0.400 | 0.310 | 0.302 | 0.472 | 0.303 | 0.290 |
| cardiotocography-3clases | 2126 | 21 | 3 | 0.439 | 0.386 | 0.393 | 0.273 | 0.282 | 0.382 | 0.125 | 0.284 |
| chess-krvk | 28056 | 6 | 18 | 0.550 | 0.523 | 0.715 | 0.486 | 0.449 | 0.515 | 0.386 | 0.405 |
| chess-krvkp | 3196 | 36 | 2 | 0.347 | 0.339 | 0.163 | 0.037 | 0.087 | 0.339 | 0.038 | 0.054 |
| climate | 540 | 20 | 2 | 0.456 | 0.428 | 0.276 | 0.093 | 0.394 | 0.428 | 0.095 | 0.132 |
| congressional-voting | 435 | 16 | 2 | 0.468 | 0.465 | 0.429 | 0.474 | 0.459 | 0.464 | 0.478 | 0.470 |
| conn-bench-sonar-mines-rocks | 208 | 60 | 2 | 0.545 | 0.531 | 0.697 | 0.296 | 0.343 | 0.531 | 0.391 | 0.304 |
| conn-bench-vowel-deterding | 990 | 11 | 11 | 0.689 | 0.523 | 0.875 | 0.148 | 0.170 | 0.493 | 0.271 | 0.200 |

*Continued on next page.*

| Dataset | N | M | K | RED | MCP | Trust Score | ConfidNet | Intro-Net | Entropy | DNGO | SVGP |
|---|---|---|---|---|---|---|---|---|---|---|---|
| connect-4 | 67557 | 42 | 2 | 0.406 | 0.390 | 0.326 | 0.259 | 0.395 | 0.390 | 0.321 | 0.155 |
| contrac | 1473 | 9 | 3 | 0.596 | 0.590 | 0.529 | 0.552 | 0.584 | 0.577 | 0.573 | 0.548 |
| credit-approval | 690 | 15 | 2 | 0.365 | 0.359 | 0.336 | 0.264 | 0.331 | 0.359 | 0.302 | 0.336 |
| cylinder-bands | 512 | 35 | 2 | 0.494 | 0.482 | 0.506 | 0.345 | 0.422 | 0.482 | 0.383 | 0.304 |
| dermatology | 366 | 34 | 6 | 0.690 | 0.648 | 0.605 | 0.072 | 0.253 | 0.686 | 0.327 | 0.130 |
| echocardiogram | 131 | 10 | 2 | 0.510 | 0.413 | 0.325 | 0.353 | 0.407 | 0.413 | 0.306 | 0.375 |
| ecoli | 336 | 7 | 8 | 0.433 | 0.421 | 0.472 | 0.283 | 0.284 | 0.374 | 0.278 | 0.286 |
| energy-y1 | 768 | 8 | 3 | 0.472 | 0.380 | 0.289 | 0.195 | 0.187 | 0.364 | 0.144 | 0.244 |
| energy-y2 | 768 | 8 | 3 | 0.498 | 0.434 | 0.498 | 0.322 | 0.278 | 0.421 | 0.302 | 0.328 |
| fertility | 100 | 9 | 2 | 0.267 | 0.238 | 0.239 | 0.193 | 0.217 | 0.238 | 0.169 | 0.160 |
| flags | 194 | 28 | 8 | 0.767 | 0.717 | 0.784 | 0.666 | 0.664 | 0.712 | 0.649 | 0.659 |
| glass | 214 | 9 | 6 | 0.583 | 0.525 | 0.657 | 0.507 | 0.412 | 0.489 | 0.498 | 0.414 |
| haberman-survival | 306 | 3 | 2 | 0.452 | 0.446 | 0.386 | 0.412 | 0.425 | 0.446 | 0.360 | 0.499 |
| hayes-roth | 160 | 3 | 3 | 0.688 | 0.576 | 0.728 | 0.603 | 0.572 | 0.576 | 0.657 | 0.536 |
| heart-cleveland | 303 | 13 | 5 | 0.759 | 0.748 | 0.707 | 0.689 | 0.705 | 0.732 | 0.627 | 0.730 |
| heart-hungarian | 294 | 12 | 2 | 0.436 | 0.418 | 0.442 | 0.261 | 0.397 | 0.418 | 0.356 | 0.351 |
| heart-switzerland | 123 | 12 | 5 | 0.735 | 0.692 | 0.726 | 0.697 | 0.662 | 0.700 | 0.705 | 0.649 |
| heart-va | 200 | 12 | 5 | 0.757 | 0.732 | 0.676 | 0.678 | 0.732 | 0.746 | 0.677 | 0.639 |
| hepatitis | 155 | 19 | 2 | 0.542 | 0.441 | 0.495 | 0.423 | 0.330 | 0.441 | 0.408 | 0.369 |
| hill-valley | 1212 | 100 | 2 | 0.657 | 0.546 | 0.478 | 0.508 | 0.594 | 0.546 | 0.554 | 0.546 |
| horse-colic | 368 | 25 | 2 | 0.444 | 0.431 | 0.426 | 0.263 | 0.394 | 0.431 | 0.304 | 0.247 |
| ilpd-indian-liver | 583 | 9 | 2 | 0.470 | 0.459 | 0.427 | 0.434 | 0.443 | 0.459 | 0.375 | 0.445 |
| image-segmentation | 2310 | 18 | 7 | 0.539 | 0.455 | 0.471 | 0.251 | 0.214 | 0.430 | 0.218 | 0.139 |
| ionosphere | 351 | 33 | 2 | 0.543 | 0.493 | 0.335 | 0.158 | 0.405 | 0.493 | 0.300 | 0.171 |
| iris | 150 | 4 | 3 | 0.686 | 0.633 | 0.747 | 0.154 | 0.202 | 0.605 | 0.360 | 0.227 |
| led-display | 1000 | 7 | 10 | 0.572 | 0.571 | 0.301 | 0.553 | 0.512 | 0.547 | 0.508 | 0.470 |
| lenses | 24 | 4 | 3 | 0.875 | 0.900 | 0.786 | 0.665 | 0.790 | 0.900 | 0.622 | 0.587 |
| letter | 20000 | 16 | 26 | 0.495 | 0.461 | 0.785 | 0.245 | 0.081 | 0.465 | 0.109 | 0.164 |
| libras | 360 | 90 | 15 | 0.602 | 0.512 | 0.657 | 0.307 | 0.301 | 0.479 | 0.498 | 0.279 |
| low-res-spect | 531 | 100 | 9 | 0.552 | 0.515 | 0.454 | 0.304 | 0.305 | 0.504 | 0.297 | 0.343 |
| lung-cancer | 32 | 56 | 3 | 0.887 | 0.819 | 0.695 | 0.690 | 0.841 | 0.859 | 0.641 | 0.582 |
| lymphography | 148 | 18 | 4 | 0.660 | 0.546 | 0.619 | 0.278 | 0.386 | 0.545 | 0.353 | 0.363 |
| magic | 19020 | 10 | 2 | 0.366 | 0.355 | 0.294 | 0.309 | 0.350 | 0.355 | 0.240 | 0.233 |
| mammographic | 961 | 5 | 2 | 0.471 | 0.468 | 0.288 | 0.414 | 0.454 | 0.468 | 0.386 | 0.402 |
| messidor | 1151 | 19 | 2 | 0.465 | 0.449 | 0.359 | 0.364 | 0.426 | 0.449 | 0.411 | 0.350 |
| miniboone | 130064 | 50 | 2 | 0.380 | 0.361 | 0.190 | 0.299 | 0.371 | 0.361 | 0.251 | 0.162 |
| molec-biol-promoter | 106 | 57 | 2 | 0.550 | 0.541 | 0.786 | 0.330 | 0.505 | 0.541 | 0.432 | 0.437 |
| molec-biol-splice | 3190 | 60 | 3 | 0.461 | 0.455 | 0.375 | 0.193 | 0.208 | 0.464 | 0.396 | 0.174 |
| monks-1 | 556 | 6 | 2 | 0.618 | 0.444 | 0.167 | 0.242 | 0.343 | 0.444 | 0.110 | 0.222 |
| monks-2 | 601 | 6 | 2 | 0.524 | 0.466 | 0.611 | 0.417 | 0.426 | 0.466 | 0.389 | 0.387 |
| monks-3 | 554 | 6 | 2 | 0.648 | 0.561 | 0.369 | 0.319 | 0.453 | 0.561 | 0.395 | 0.175 |
| mushroom | 8124 | 21 | 2 | NA | NA | NA | NA | NA | NA | NA | NA |
| musk-1 | 476 | 166 | 2 | 0.518 | 0.508 | 0.501 | 0.101 | 0.354 | 0.508 | 0.295 | 0.103 |
| musk-2 | 6598 | 166 | 2 | 0.409 | 0.372 | 0.083 | 0.009 | 0.077 | 0.372 | 0.006 | 0.007 |
| nursery | 12960 | 8 | 5 | 0.586 | 0.448 | 0.074 | 0.035 | 0.127 | 0.447 | 0.135 | 0.173 |
| oocytes_merluccius_nucleus_4d | 1022 | 41 | 2 | 0.442 | 0.401 | 0.277 | 0.296 | 0.383 | 0.401 | 0.385 | 0.330 |
| oocytes_merluccius_states_2f | 1022 | 25 | 3 | 0.505 | 0.488 | 0.413 | 0.256 | 0.358 | 0.477 | 0.214 | 0.262 |
| oocytes_trisopterus_nucleus_2f | 912 | 25 | 2 | 0.481 | 0.451 | 0.342 | 0.275 | 0.460 | 0.451 | 0.354 | 0.300 |
| oocytes_trisopterus_states_5b | 912 | 32 | 3 | 0.445 | 0.423 | 0.346 | 0.155 | 0.291 | 0.441 | 0.316 | 0.114 |
| optical | 5620 | 62 | 10 | 0.452 | 0.429 | 0.582 | 0.048 | 0.036 | 0.422 | 0.079 | 0.068 |
| ozone | 2536 | 72 | 2 | 0.308 | 0.293 | 0.270 | 0.207 | 0.146 | 0.293 | 0.032 | 0.025 |
| page-blocks | 5473 | 10 | 5 | 0.450 | 0.386 | 0.399 | 0.330 | 0.275 | 0.371 | 0.254 | 0.186 |
| parkinsons | 195 | 22 | 2 | 0.562 | 0.516 | 0.541 | 0.155 | 0.239 | 0.516 | 0.276 | 0.316 |
| pendigits | 10992 | 16 | 10 | 0.491 | 0.362 | 0.684 | 0.178 | 0.132 | 0.363 | 0.104 | 0.059 |
| phishing | 1353 | 9 | 3 | 0.453 | 0.433 | 0.369 | 0.347 | 0.371 | 0.412 | 0.338 | 0.334 |
| pima | 768 | 8 | 2 | 0.501 | 0.492 | 0.422 | 0.412 | 0.486 | 0.492 | 0.426 | 0.336 |
| pittsburg-bridges-MATERIAL | 106 | 7 | 3 | 0.600 | 0.501 | 0.485 | 0.360 | 0.511 | 0.506 | 0.336 | 0.392 |
| pittsburg-bridges-REL-L | 103 | 7 | 3 | 0.593 | 0.536 | 0.654 | 0.522 | 0.549 | 0.529 | 0.501 | 0.498 |
| pittsburg-bridges-SPAN | 92 | 7 | 3 | 0.582 | 0.550 | 0.543 | 0.367 | 0.507 | 0.563 | 0.472 | 0.449 |
| pittsburg-bridges-T-OR-D | 102 | 7 | 2 | 0.472 | 0.397 | 0.456 | 0.391 | 0.506 | 0.397 | 0.375 | 0.352 |
| pittsburg-bridges-TYPE | 105 | 7 | 6 | 0.809 | 0.736 | 0.758 | 0.684 | 0.662 | 0.692 | 0.702 | 0.737 |
| planning | 182 | 12 | 2 | 0.366 | 0.364 | 0.387 | 0.348 | 0.331 | 0.364 | 0.312 | 0.259 |
| plant-margin | 1600 | 64 | 100 | 0.586 | 0.572 | 0.696 | 0.331 | 0.266 | 0.569 | 0.305 | 0.183 |
| plant-shape | 1600 | 64 | 100 | 0.693 | 0.666 | 0.759 | 0.524 | 0.477 | 0.643 | 0.517 | 0.477 |
| plant-texture | 1599 | 64 | 100 | 0.607 | 0.596 | 0.708 | 0.308 | 0.291 | 0.610 | 0.282 | 0.221 |
| post-operative | 90 | 8 | 3 | 0.428 | 0.416 | 0.289 | 0.377 | 0.413 | 0.374 | 0.441 | 0.418 |
| primary-tumor | 330 | 17 | 15 | 0.794 | 0.783 | 0.693 | 0.725 | 0.721 | 0.779 | 0.683 | 0.715 |
| ringnorm | 7400 | 20 | 2 | 0.348 | 0.344 | 0.275 | 0.042 | 0.116 | 0.344 | 0.112 | 0.027 |
| seeds | 210 | 7 | 3 | 0.583 | 0.526 | 0.526 | 0.344 | 0.333 | 0.475 | 0.397 | 0.345 |
| semeion | 1593 | 256 | 10 | 0.541 | 0.527 | 0.704 | 0.092 | 0.237 | 0.564 | 0.180 | 0.142 |
| soybean | 683 | 35 | 18 | 0.573 | 0.516 | 0.448 | 0.317 | 0.191 | 0.505 | 0.234 | 0.319 |
| spambase | 4601 | 57 | 2 | 0.316 | 0.292 | 0.299 | 0.179 | 0.238 | 0.292 | 0.203 | 0.139 |
| spect | 265 | 22 | 2 | 0.520 | 0.506 | 0.384 | 0.443 | 0.471 | 0.506 | 0.419 | 0.434 |
| spectf | 267 | 44 | 2 | 0.478 | 0.427 | 0.429 | 0.251 | 0.428 | 0.427 | 0.258 | 0.346 |
| statlog-australian-credit | 690 | 14 | 2 | 0.473 | 0.455 | 0.418 | 0.417 | 0.416 | 0.455 | 0.407 | 0.430 |
| statlog-german-credit | 1000 | 24 | 2 | 0.423 | 0.422 | 0.383 | 0.329 | 0.389 | 0.422 | 0.354 | 0.254 |
| statlog-heart | 270 | 13 | 2 | 0.471 | 0.438 | 0.388 | 0.322 | 0.462 | 0.438 | 0.350 | 0.285 |

*Continued on next page.*

| Dataset | N | M | K | RED | MCP | Trust Score | ConfidNet | Intro-Net | Entropy | DNGO | SVGP |
|---|---|---|---|---|---|---|---|---|---|---|---|
| statlog-image | 2310 | 18 | 7 | 0.561 | 0.464 | 0.484 | 0.240 | 0.114 | 0.473 | 0.274 | 0.183 |
| statlog-landsat | 6435 | 36 | 6 | 0.430 | 0.398 | 0.554 | 0.322 | 0.312 | 0.409 | 0.288 | 0.255 |
| statlog-shuttle | 58000 | 9 | 7 | 0.551 | 0.318 | 0.547 | 0.233 | 0.157 | 0.310 | 0.139 | 0.247 |
| statlog-vehicle | 846 | 18 | 4 | 0.542 | 0.526 | 0.319 | 0.392 | 0.361 | 0.536 | 0.346 | 0.364 |
| steel-plates | 1941 | 27 | 7 | 0.529 | 0.517 | 0.533 | 0.452 | 0.437 | 0.514 | 0.437 | 0.404 |
| synthetic-control | 600 | 60 | 6 | 0.647 | 0.541 | 0.462 | 0.052 | 0.117 | 0.498 | 0.199 | 0.031 |
| teaching | 151 | 5 | 3 | 0.646 | 0.604 | 0.540 | 0.632 | 0.593 | 0.612 | 0.676 | 0.569 |
| thyroid | 7200 | 21 | 3 | 0.442 | 0.413 | 0.160 | 0.126 | 0.190 | 0.409 | 0.015 | 0.114 |
| tic-tac-toe | 958 | 9 | 2 | 0.633 | 0.538 | 1.000 | 0.160 | 0.513 | 0.538 | 0.385 | 0.480 |
| titanic | 2201 | 3 | 2 | 0.314 | 0.313 | 0.202 | 0.310 | 0.315 | 0.313 | 0.227 | 0.321 |
| trains | 10 | 29 | 2 | 0.778 | 0.750 | 0.917 | 0.833 | 0.750 | 0.750 | 0.625 | 0.750 |
| twonorm | 7400 | 20 | 2 | 0.371 | 0.357 | 0.404 | 0.160 | 0.290 | 0.357 | 0.176 | 0.037 |
| vertebral-column-2clases | 310 | 6 | 2 | 0.501 | 0.490 | 0.346 | 0.378 | 0.432 | 0.490 | 0.389 | 0.273 |
| vertebral-column-3clases | 310 | 6 | 3 | 0.542 | 0.518 | 0.308 | 0.489 | 0.396 | 0.529 | 0.439 | 0.307 |
| wall-following | 5456 | 24 | 4 | 0.418 | 0.378 | 0.304 | 0.208 | 0.248 | 0.370 | 0.261 | 0.139 |
| waveform | 5000 | 21 | 3 | 0.465 | 0.423 | 0.331 | 0.357 | 0.394 | 0.423 | 0.200 | 0.162 |
| waveform-noise | 5000 | 40 | 3 | 0.428 | 0.398 | 0.322 | 0.230 | 0.326 | 0.397 | 0.215 | 0.167 |
| wine | 178 | 13 | 3 | 0.894 | 0.742 | 0.661 | 0.101 | 0.458 | 0.825 | 0.613 | 0.284 |
| wine-quality-red | 1599 | 11 | 6 | 0.534 | 0.522 | 0.540 | 0.494 | 0.491 | 0.496 | 0.523 | 0.453 |
| wine-quality-white | 4898 | 11 | 7 | 0.541 | 0.529 | 0.626 | 0.534 | 0.523 | 0.516 | 0.524 | 0.480 |
| yeast | 1484 | 8 | 10 | 0.570 | 0.563 | 0.510 | 0.506 | 0.498 | 0.548 | 0.410 | 0.912 |
| zoo | 101 | 16 | 7 | 0.910 | 0.804 | 0.793 | 0.297 | 0.546 | 0.893 | 0.443 | 0.694 |

*Continued from previous page.*

Table 6: Comparison between RED and Counterparts Using AP-Success

| Dataset | N | M | K | RED | MCP | Trust Score | ConfidNet | Intro-Net | Entropy | DNGO | SVGP |
|---|---|---|---|---|---|---|---|---|---|---|---|
| abalone | 4177 | 8 | 3 | 0.853 | 0.850 | 0.821 | 0.845 | 0.844 | 0.849 | 0.814 | 0.840 |
| acute-inflammation | 120 | 6 | 2 | NA | NA | NA | NA | NA | NA | NA | NA |
| acute-nephritis | 120 | 6 | 2 | NA | NA | NA | NA | NA | NA | NA | NA |
| adult | 48842 | 14 | 2 | 0.968 | 0.967 | 0.944 | 0.964 | 0.961 | 0.967 | 0.927 | 0.961 |
| annealing | 898 | 31 | 5 | 0.975 | 0.899 | 0.943 | 0.909 | 0.916 | 0.902 | 0.898 | 0.913 |
| arrhythmia | 452 | 262 | 13 | 0.817 | 0.810 | 0.809 | 0.654 | 0.668 | 0.803 | 0.673 | 0.778 |
| audiology-std | 196 | 59 | 18 | 0.946 | 0.926 | NA | 0.625 | 0.740 | 0.927 | 0.756 | 0.782 |
| balance-scale | 625 | 4 | 3 | 0.999 | 0.999 | 0.997 | 0.972 | 0.983 | 0.999 | 0.985 | 0.976 |
| balloons | 16 | 4 | 2 | 0.843 | 0.806 | 0.806 | 0.769 | 0.806 | 0.806 | 0.625 | 0.667 |
| bank | 4521 | 16 | 2 | 0.977 | 0.980 | 0.976 | 0.968 | 0.937 | 0.980 | 0.892 | 0.979 |
| bioconcentration | 779 | 9 | 3 | 0.746 | 0.728 | 0.782 | 0.717 | 0.726 | 0.721 | 0.693 | 0.698 |
| blood | 748 | 4 | 2 | 0.892 | 0.892 | 0.873 | 0.902 | 0.883 | 0.892 | 0.869 | 0.900 |
| breast-cancer | 286 | 9 | 2 | 0.820 | 0.820 | 0.765 | 0.796 | 0.775 | 0.820 | 0.767 | 0.785 |
| breast-cancer-wisc | 699 | 9 | 2 | 0.998 | 0.997 | 0.998 | 0.994 | 0.985 | 0.997 | 0.987 | 0.986 |
| breast-cancer-wisc-diag | 569 | 30 | 2 | 0.996 | 0.996 | 0.997 | 0.970 | 0.981 | 0.998 | 0.973 | 0.981 |
| breast-cancer-wisc-prog | 198 | 33 | 2 | 0.856 | 0.860 | 0.797 | 0.824 | 0.835 | 0.860 | 0.787 | 0.795 |
| breast-tissue | 106 | 9 | 6 | 0.909 | 0.878 | 0.898 | 0.863 | 0.767 | 0.872 | 0.800 | 0.849 |
| car | 1728 | 6 | 4 | 0.999 | 0.999 | 0.999 | 0.992 | 0.988 | 0.999 | 0.984 | 0.990 |
| cardiotocography-10clases | 2126 | 21 | 10 | 0.959 | 0.959 | 0.927 | 0.890 | 0.872 | 0.958 | 0.885 | 0.901 |
| cardiotocography-3clases | 2126 | 21 | 3 | 0.990 | 0.991 | 0.988 | 0.972 | 0.964 | 0.991 | 0.935 | 0.979 |
| chess-krvk | 28056 | 6 | 18 | 0.900 | 0.890 | 0.934 | 0.867 | 0.838 | 0.889 | 0.782 | 0.801 |
| chess-krvkp | 3196 | 36 | 2 | 1.000 | 1.000 | 0.995 | 0.990 | 0.994 | 1.000 | 0.993 | 0.996 |
| climate | 540 | 20 | 2 | 0.991 | 0.989 | 0.976 | 0.938 | 0.976 | 0.987 | 0.945 | 0.950 |
| congressional-voting | 435 | 16 | 2 | 0.693 | 0.690 | 0.672 | 0.692 | 0.684 | 0.689 | 0.691 | 0.689 |
| conn-bench-sonar-mines-rocks | 208 | 60 | 2 | 0.962 | 0.963 | 0.980 | 0.815 | 0.901 | 0.963 | 0.874 | 0.897 |
| conn-bench-vowel-deterding | 990 | 11 | 11 | 1.000 | 1.000 | 1.000 | 0.980 | 0.984 | 1.000 | 0.989 | 0.989 |
| connect-4 | 67557 | 42 | 2 | 0.969 | 0.969 | 0.938 | 0.942 | 0.958 | 0.969 | 0.927 | 0.868 |
| contrac | 1473 | 9 | 3 | 0.717 | 0.714 | 0.581 | 0.682 | 0.675 | 0.705 | 0.658 | 0.669 |
| credit-approval | 690 | 15 | 2 | 0.943 | 0.941 | 0.916 | 0.909 | 0.936 | 0.941 | 0.896 | 0.928 |
| cylinder-bands | 512 | 35 | 2 | 0.865 | 0.862 | 0.872 | 0.789 | 0.809 | 0.862 | 0.767 | 0.758 |
| dermatology | 366 | 34 | 6 | 0.999 | 0.999 | 0.997 | 0.975 | 0.981 | 0.999 | 0.977 | 0.951 |
| echocardiogram | 131 | 10 | 2 | 0.944 | 0.936 | 0.925 | 0.916 | 0.924 | 0.936 | 0.888 | 0.929 |
| ecoli | 336 | 7 | 8 | 0.959 | 0.956 | 0.969 | 0.911 | 0.879 | 0.954 | 0.886 | 0.919 |
| energy-y1 | 768 | 8 | 3 | 0.992 | 0.994 | 0.992 | 0.975 | 0.974 | 0.994 | 0.959 | 0.983 |
| energy-y2 | 768 | 8 | 3 | 0.986 | 0.986 | 0.985 | 0.962 | 0.946 | 0.986 | 0.963 | 0.978 |
| fertility | 100 | 9 | 2 | 0.953 | 0.928 | 0.948 | 0.920 | 0.929 | 0.928 | 0.903 | 0.939 |
| flags | 194 | 28 | 8 | 0.739 | 0.716 | 0.817 | 0.573 | 0.617 | 0.720 | 0.580 | 0.599 |
| glass | 214 | 9 | 6 | 0.854 | 0.811 | 0.915 | 0.779 | 0.738 | 0.810 | 0.741 | 0.783 |
| haberman-survival | 306 | 3 | 2 | 0.850 | 0.849 | 0.854 | 0.821 | 0.835 | 0.849 | 0.818 | 0.835 |
| hayes-roth | 160 | 3 | 3 | 0.931 | 0.881 | 0.959 | 0.887 | 0.864 | 0.887 | 0.881 | 0.837 |
| heart-cleveland | 303 | 13 | 5 | 0.856 | 0.859 | 0.836 | 0.800 | 0.822 | 0.861 | 0.696 | 0.838 |
| heart-hungarian | 294 | 12 | 2 | 0.925 | 0.908 | 0.927 | 0.847 | 0.915 | 0.908 | 0.844 | 0.869 |
| heart-switzerland | 123 | 12 | 5 | 0.500 | 0.480 | 0.478 | 0.451 | 0.452 | 0.490 | 0.471 | 0.446 |
| heart-va | 200 | 12 | 5 | 0.526 | 0.499 | 0.462 | 0.439 | 0.504 | 0.511 | 0.417 | 0.392 |
| hepatitis | 155 | 19 | 2 | 0.965 | 0.965 | 0.975 | 0.930 | 0.896 | 0.965 | 0.897 | 0.915 |
| hill-valley | 1212 | 100 | 2 | 0.763 | 0.704 | 0.559 | 0.653 | 0.741 | 0.704 | 0.698 | 0.693 |
| horse-colic | 368 | 25 | 2 | 0.927 | 0.918 | 0.920 | 0.833 | 0.913 | 0.918 | 0.836 | 0.871 |

*Continued on next page.*

| Dataset | N | M | K | RED | MCP | Trust Score | ConfidNet | Intro-Net | Entropy | DNGO | SVGP |
|---|---|---|---|---|---|---|---|---|---|---|---|
| ilpd-indian-liver | 583 | 9 | 2 | 0.878 | 0.874 | 0.814 | 0.860 | 0.863 | 0.874 | 0.800 | 0.871 |
| image-segmentation | 2310 | 18 | 7 | 0.997 | 0.997 | 0.998 | 0.983 | 0.984 | 0.997 | 0.981 | 0.985 |
| ionosphere | 351 | 33 | 2 | 0.990 | 0.988 | 0.976 | 0.900 | 0.973 | 0.988 | 0.931 | 0.934 |
| iris | 150 | 4 | 3 | 0.992 | 0.991 | 0.997 | 0.938 | 0.917 | 0.991 | 0.950 | 0.955 |
| led-display | 1000 | 7 | 10 | 0.883 | 0.883 | 0.759 | 0.874 | 0.850 | 0.880 | 0.835 | 0.847 |
| lenses | 24 | 4 | 3 | 0.982 | 0.960 | 0.944 | 0.871 | 0.883 | 0.960 | 0.859 | 0.844 |
| letter | 20000 | 16 | 26 | 0.995 | 0.996 | 0.999 | 0.980 | 0.947 | 0.996 | 0.963 | 0.982 |
| libras | 360 | 90 | 15 | 0.956 | 0.943 | 0.956 | 0.808 | 0.807 | 0.941 | 0.863 | 0.850 |
| low-res-spect | 531 | 100 | 9 | 0.977 | 0.968 | 0.953 | 0.908 | 0.892 | 0.968 | 0.902 | 0.950 |
| lung-cancer | 32 | 56 | 3 | 0.771 | 0.734 | 0.650 | 0.547 | 0.743 | 0.739 | 0.496 | 0.459 |
| lymphography | 148 | 18 | 4 | 0.941 | 0.925 | 0.958 | 0.809 | 0.861 | 0.924 | 0.865 | 0.826 |
| magic | 19020 | 10 | 2 | 0.966 | 0.966 | 0.954 | 0.961 | 0.959 | 0.966 | 0.907 | 0.946 |
| mammographic | 961 | 5 | 2 | 0.922 | 0.919 | 0.882 | 0.910 | 0.911 | 0.919 | 0.882 | 0.926 |
| messidor | 1151 | 19 | 2 | 0.863 | 0.861 | 0.764 | 0.806 | 0.830 | 0.861 | 0.808 | 0.806 |
| miniboone | 130064 | 50 | 2 | 0.992 | 0.992 | 0.983 | 0.988 | 0.986 | 0.992 | 0.972 | 0.981 |
| molec-biol-promoter | 106 | 57 | 2 | 0.892 | 0.888 | 0.953 | 0.724 | 0.880 | 0.888 | 0.784 | 0.822 |
| molec-biol-splice | 3190 | 60 | 3 | 0.953 | 0.953 | 0.926 | 0.833 | 0.846 | 0.952 | 0.901 | 0.838 |
| monks-1 | 556 | 6 | 2 | 0.998 | 0.997 | 0.984 | 0.988 | 0.989 | 0.997 | 0.981 | 0.995 |
| monks-2 | 601 | 6 | 2 | 0.885 | 0.857 | 0.918 | 0.809 | 0.810 | 0.857 | 0.761 | 0.808 |
| monks-3 | 554 | 6 | 2 | 0.999 | 0.999 | 0.993 | 0.981 | 0.993 | 0.999 | 0.981 | 0.991 |
| mushroom | 8124 | 21 | 2 | NA | NA | NA | NA | NA | NA | NA | NA |
| musk-1 | 476 | 166 | 2 | 0.984 | 0.983 | 0.986 | 0.912 | 0.957 | 0.983 | 0.942 | 0.934 |
| musk-2 | 6598 | 166 | 2 | 1.000 | 1.000 | 0.998 | 0.995 | 0.995 | 1.000 | 0.994 | 0.994 |
| nursery | 12960 | 8 | 5 | 1.000 | 1.000 | 1.000 | 0.999 | 0.999 | 1.000 | 0.999 | 1.000 |
| oocytes_merluccius_nucleus_4d | 1022 | 41 | 2 | 0.939 | 0.930 | 0.881 | 0.892 | 0.904 | 0.930 | 0.890 | 0.907 |
| oocytes_merluccius_states_2f | 1022 | 25 | 3 | 0.991 | 0.991 | 0.991 | 0.967 | 0.970 | 0.991 | 0.950 | 0.975 |
| oocytes_trisopterus_nucleus_2f | 912 | 25 | 2 | 0.940 | 0.931 | 0.908 | 0.872 | 0.923 | 0.931 | 0.873 | 0.876 |
| oocytes_trisopterus_states_5b | 912 | 32 | 3 | 0.991 | 0.990 | 0.985 | 0.952 | 0.967 | 0.990 | 0.955 | 0.945 |
| optical | 5620 | 62 | 10 | 0.999 | 0.999 | 0.999 | 0.981 | 0.981 | 0.999 | 0.982 | 0.992 |
| ozone | 2536 | 72 | 2 | 0.994 | 0.996 | 0.993 | 0.988 | 0.966 | 0.996 | 0.968 | 0.951 |
| page-blocks | 5473 | 10 | 5 | 0.998 | 0.997 | 0.993 | 0.993 | 0.989 | 0.997 | 0.989 | 0.994 |
| parkinsons | 195 | 22 | 2 | 0.994 | 0.992 | 0.995 | 0.939 | 0.944 | 0.992 | 0.962 | 0.964 |
| pendigits | 10992 | 16 | 10 | 0.999 | 0.999 | 1.000 | 0.992 | 0.992 | 0.999 | 0.994 | 0.993 |
| phishing | 1353 | 9 | 3 | 0.975 | 0.973 | 0.958 | 0.959 | 0.946 | 0.972 | 0.929 | 0.952 |
| pima | 768 | 8 | 2 | 0.898 | 0.894 | 0.875 | 0.848 | 0.873 | 0.894 | 0.857 | 0.870 |
| pittsburg-bridges-MATERIAL | 106 | 7 | 3 | 0.976 | 0.969 | 0.953 | 0.921 | 0.933 | 0.969 | 0.892 | 0.939 |
| pittsburg-bridges-REL-L | 103 | 7 | 3 | 0.790 | 0.780 | 0.845 | 0.766 | 0.744 | 0.783 | 0.735 | 0.773 |
| pittsburg-bridges-SPAN | 92 | 7 | 3 | 0.851 | 0.808 | 0.785 | 0.678 | 0.756 | 0.805 | 0.782 | 0.716 |
| pittsburg-bridges-T-OR-D | 102 | 7 | 2 | 0.957 | 0.943 | 0.924 | 0.926 | 0.946 | 0.943 | 0.932 | 0.920 |
| pittsburg-bridges-TYPE | 105 | 7 | 6 | 0.818 | 0.692 | 0.747 | 0.651 | 0.635 | 0.664 | 0.737 | 0.766 |
| planning | 182 | 12 | 2 | 0.729 | 0.722 | 0.737 | 0.730 | 0.728 | 0.722 | 0.734 | 0.692 |
| plant-margin | 1600 | 64 | 100 | 0.951 | 0.950 | 0.966 | 0.828 | 0.817 | 0.950 | 0.816 | 0.731 |
| plant-shape | 1600 | 64 | 100 | 0.861 | 0.849 | 0.880 | 0.718 | 0.671 | 0.838 | 0.697 | 0.673 |
| plant-texture | 1599 | 64 | 100 | 0.953 | 0.950 | 0.958 | 0.809 | 0.804 | 0.950 | 0.789 | 0.791 |
| post-operative | 90 | 8 | 3 | 0.683 | 0.673 | 0.739 | 0.684 | 0.685 | 0.670 | 0.727 | 0.687 |
| primary-tumor | 330 | 17 | 15 | 0.762 | 0.752 | 0.654 | 0.640 | 0.633 | 0.750 | 0.546 | 0.642 |
| ringnorm | 7400 | 20 | 2 | 0.999 | 0.999 | 0.989 | 0.982 | 0.977 | 0.999 | 0.981 | 0.976 |
| seeds | 210 | 7 | 3 | 0.988 | 0.986 | 0.992 | 0.964 | 0.961 | 0.985 | 0.935 | 0.947 |
| semeion | 1593 | 256 | 10 | 0.994 | 0.994 | 0.996 | 0.927 | 0.950 | 0.994 | 0.943 | 0.963 |
| soybean | 683 | 35 | 18 | 0.996 | 0.995 | 0.993 | 0.970 | 0.942 | 0.995 | 0.944 | 0.974 |
| spambase | 4601 | 57 | 2 | 0.989 | 0.988 | 0.984 | 0.965 | 0.966 | 0.988 | 0.962 | 0.974 |
| spect | 265 | 22 | 2 | 0.811 | 0.802 | 0.767 | 0.731 | 0.788 | 0.801 | 0.686 | 0.733 |
| spectf | 267 | 44 | 2 | 0.955 | 0.951 | 0.942 | 0.913 | 0.919 | 0.953 | 0.879 | 0.949 |
| statlog-australian-credit | 690 | 14 | 2 | 0.701 | 0.692 | 0.656 | 0.668 | 0.678 | 0.692 | 0.629 | 0.662 |
| statlog-german-credit | 1000 | 24 | 2 | 0.890 | 0.891 | 0.873 | 0.837 | 0.850 | 0.891 | 0.821 | 0.782 |
| statlog-heart | 270 | 13 | 2 | 0.940 | 0.935 | 0.906 | 0.882 | 0.917 | 0.935 | 0.853 | 0.863 |
| statlog-image | 2310 | 18 | 7 | 0.998 | 0.998 | 0.998 | 0.986 | 0.980 | 0.998 | 0.984 | 0.990 |
| statlog-landsat | 6435 | 36 | 6 | 0.986 | 0.987 | 0.991 | 0.964 | 0.959 | 0.987 | 0.953 | 0.971 |
| statlog-shuttle | 58000 | 9 | 7 | 1.000 | 1.000 | 1.000 | 0.998 | 0.998 | 0.999 | 0.999 | 0.999 |
| statlog-vehicle | 846 | 18 | 4 | 0.966 | 0.967 | 0.927 | 0.929 | 0.913 | 0.968 | 0.870 | 0.922 |
| steel-plates | 1941 | 27 | 7 | 0.919 | 0.918 | 0.928 | 0.882 | 0.850 | 0.916 | 0.827 | 0.870 |
| synthetic-control | 600 | 60 | 6 | 1.000 | 1.000 | 0.999 | 0.986 | 0.988 | 1.000 | 0.988 | 0.986 |
| teaching | 151 | 5 | 3 | 0.686 | 0.648 | 0.598 | 0.657 | 0.640 | 0.634 | 0.682 | 0.632 |
| thyroid | 7200 | 21 | 3 | 0.999 | 0.999 | 0.996 | 0.992 | 0.984 | 0.999 | 0.985 | 0.996 |
| tic-tac-toe | 958 | 9 | 2 | 1.000 | 1.000 | 1.000 | 0.999 | 0.998 | 0.999 | 0.994 | 0.999 |
| titanic | 2201 | 3 | 2 | 0.877 | 0.876 | 0.805 | 0.874 | 0.873 | 0.876 | 0.821 | 0.877 |
| trains | 10 | 29 | 2 | 0.778 | 0.750 | 0.917 | 0.833 | 0.750 | 0.750 | 0.625 | 0.750 |
| twonorm | 7400 | 20 | 2 | 0.998 | 0.999 | 0.998 | 0.991 | 0.991 | 0.999 | 0.981 | 0.973 |
| vertebral-column-2clases | 310 | 6 | 2 | 0.977 | 0.977 | 0.953 | 0.952 | 0.942 | 0.977 | 0.917 | 0.937 |
| vertebral-column-3clases | 310 | 6 | 3 | 0.978 | 0.977 | 0.943 | 0.959 | 0.933 | 0.978 | 0.912 | 0.953 |
| wall-following | 5456 | 24 | 4 | 0.980 | 0.980 | 0.972 | 0.928 | 0.949 | 0.980 | 0.945 | 0.933 |
| waveform | 5000 | 21 | 3 | 0.968 | 0.966 | 0.947 | 0.947 | 0.941 | 0.966 | 0.866 | 0.848 |
| waveform-noise | 5000 | 40 | 3 | 0.962 | 0.962 | 0.935 | 0.881 | 0.900 | 0.962 | 0.872 | 0.844 |
| wine | 178 | 13 | 3 | 0.999 | 0.999 | 0.998 | 0.948 | 0.990 | 0.999 | 0.969 | 0.978 |
| wine-quality-red | 1599 | 11 | 6 | 0.735 | 0.727 | 0.751 | 0.705 | 0.696 | 0.719 | 0.695 | 0.697 |
| wine-quality-white | 4898 | 11 | 7 | 0.673 | 0.663 | 0.745 | 0.650 | 0.655 | 0.656 | 0.652 | 0.611 |
| yeast | 1484 | 8 | 10 | 0.756 | 0.751 | 0.706 | 0.718 | 0.692 | 0.745 | 0.601 | 0.579 |

*Continued on next page.*

| Dataset | N | M | K | RED | MCP | Trust Score | ConfidNet | Intro-Net | Entropy | DNGO | SVGP |
|---|---|---|---|---|---|---|---|---|---|---|---|
| zoo | 101 | 16 | 7 | 0.999 | 0.998 | 0.997 | 0.928 | 0.963 | 0.999 | 0.963 | 0.950 |

*Continued from previous page.*

Table 7: Comparison between RED and Counterparts Using AUPR-Error

| Dataset | N | M | K | RED | MCP | Trust Score | ConfidNet | Intro-Net | Entropy | DNGO | SVGP |
|---|---|---|---|---|---|---|---|---|---|---|---|
| abalone | 4177 | 8 | 3 | 0.543 | 0.537 | 0.478 | 0.516 | 0.526 | 0.527 | 0.498 | 0.480 |
| acute-inflammation | 120 | 6 | 2 | NA | NA | NA | NA | NA | NA | NA | NA |
| acute-nephritis | 120 | 6 | 2 | NA | NA | NA | NA | NA | NA | NA | NA |
| adult | 48842 | 14 | 2 | 0.419 | 0.410 | 0.328 | 0.390 | 0.412 | 0.410 | 0.372 | 0.357 |
| annealing | 898 | 31 | 5 | 0.547 | 0.375 | 0.440 | 0.331 | 0.385 | 0.350 | 0.347 | 0.352 |
| arrhythmia | 452 | 262 | 13 | 0.573 | 0.570 | 0.606 | 0.396 | 0.399 | 0.547 | 0.412 | 0.465 |
| audiology-std | 196 | 59 | 18 | 0.766 | 0.704 | NA | 0.401 | 0.494 | 0.712 | 0.564 | 0.532 |
| balance-scale | 625 | 4 | 3 | 0.877 | 0.794 | 0.501 | 0.030 | 0.124 | 0.685 | 0.396 | 0.085 |
| balloons | 16 | 4 | 2 | 0.785 | 0.736 | 0.796 | 0.569 | 0.736 | 0.736 | 0.479 | 0.458 |
| bank | 4521 | 16 | 2 | 0.391 | 0.387 | 0.385 | 0.367 | 0.333 | 0.387 | 0.554 | 0.377 |
| bioconcentration | 779 | 9 | 3 | 0.485 | 0.461 | 0.501 | 0.418 | 0.453 | 0.431 | 0.398 | 0.376 |
| blood | 748 | 4 | 2 | 0.377 | 0.375 | 0.312 | 0.367 | 0.357 | 0.377 | 0.345 | 0.348 |
| breast-cancer | 286 | 9 | 2 | 0.518 | 0.483 | 0.354 | 0.447 | 0.405 | 0.483 | 0.421 | 0.452 |
| breast-cancer-wisc | 699 | 9 | 2 | 0.374 | 0.341 | 0.347 | 0.253 | 0.184 | 0.341 | 0.367 | 0.121 |
| breast-cancer-wisc-diag | 569 | 30 | 2 | 0.594 | 0.561 | 0.438 | 0.070 | 0.348 | 0.562 | 0.170 | 0.100 |
| breast-cancer-wisc-prog | 198 | 33 | 2 | 0.374 | 0.369 | 0.254 | 0.310 | 0.344 | 0.369 | 0.259 | 0.236 |
| breast-tissue | 106 | 9 | 6 | 0.795 | 0.717 | 0.771 | 0.660 | 0.556 | 0.722 | 0.717 | 0.690 |
| car | 1728 | 6 | 4 | 0.537 | 0.419 | 0.236 | 0.111 | 0.041 | 0.413 | 0.317 | 0.081 |
| cardiotocography-10clases | 2126 | 21 | 10 | 0.507 | 0.486 | 0.394 | 0.301 | 0.295 | 0.465 | 0.297 | 0.284 |
| cardiotocography-3clases | 2126 | 21 | 3 | 0.429 | 0.372 | 0.377 | 0.259 | 0.268 | 0.369 | 0.493 | 0.272 |
| chess-krvk | 28056 | 6 | 18 | 0.550 | 0.523 | 0.715 | 0.485 | 0.448 | 0.514 | 0.386 | 0.404 |
| chess-krvkp | 3196 | 36 | 2 | 0.308 | 0.299 | 0.141 | 0.024 | 0.067 | 0.299 | 0.245 | 0.036 |
| climate | 540 | 20 | 2 | 0.421 | 0.395 | 0.239 | 0.081 | 0.362 | 0.395 | 0.296 | 0.110 |
| congressional-voting | 435 | 16 | 2 | 0.503 | 0.503 | 0.659 | 0.505 | 0.543 | 0.488 | 0.503 | 0.511 |
| conn-bench-sonar-mines-rocks | 208 | 60 | 2 | 0.507 | 0.487 | 0.666 | 0.260 | 0.305 | 0.487 | 0.352 | 0.264 |
| conn-bench-vowel-deterding | 990 | 11 | 11 | 0.636 | 0.455 | 0.865 | 0.123 | 0.144 | 0.431 | 0.250 | 0.152 |
| connect-4 | 67557 | 42 | 2 | 0.405 | 0.389 | 0.326 | 0.258 | 0.394 | 0.389 | 0.371 | 0.155 |
| contrac | 1473 | 9 | 3 | 0.591 | 0.585 | 0.525 | 0.546 | 0.579 | 0.573 | 0.569 | 0.544 |
| credit-approval | 690 | 15 | 2 | 0.344 | 0.337 | 0.316 | 0.241 | 0.311 | 0.337 | 0.289 | 0.316 |
| cylinder-bands | 512 | 35 | 2 | 0.481 | 0.464 | 0.493 | 0.331 | 0.406 | 0.464 | 0.373 | 0.288 |
| dermatology | 366 | 34 | 6 | 0.603 | 0.572 | 0.573 | 0.046 | 0.194 | 0.607 | 0.317 | 0.121 |
| echocardiogram | 131 | 10 | 2 | 0.463 | 0.350 | 0.274 | 0.303 | 0.357 | 0.350 | 0.236 | 0.325 |
| ecoli | 336 | 7 | 8 | 0.409 | 0.396 | 0.443 | 0.244 | 0.259 | 0.344 | 0.251 | 0.251 |
| energy-y1 | 768 | 8 | 3 | 0.442 | 0.333 | 0.258 | 0.163 | 0.165 | 0.312 | 0.165 | 0.211 |
| energy-y2 | 768 | 8 | 3 | 0.481 | 0.409 | 0.479 | 0.298 | 0.260 | 0.396 | 0.290 | 0.307 |
| fertility | 100 | 9 | 2 | 0.218 | 0.200 | 0.180 | 0.153 | 0.165 | 0.200 | 0.121 | 0.098 |
| flags | 194 | 28 | 8 | 0.760 | 0.703 | 0.774 | 0.651 | 0.649 | 0.697 | 0.632 | 0.647 |
| glass | 214 | 9 | 6 | 0.564 | 0.501 | 0.637 | 0.480 | 0.385 | 0.462 | 0.471 | 0.384 |
| haberman-survival | 306 | 3 | 2 | 0.426 | 0.420 | 0.363 | 0.389 | 0.400 | 0.420 | 0.334 | 0.477 |
| hayes-roth | 160 | 3 | 3 | 0.672 | 0.560 | 0.796 | 0.569 | 0.537 | 0.553 | 0.649 | 0.510 |
| heart-cleveland | 303 | 13 | 5 | 0.752 | 0.739 | 0.694 | 0.675 | 0.694 | 0.719 | 0.612 | 0.720 |
| heart-hungarian | 294 | 12 | 2 | 0.406 | 0.388 | 0.416 | 0.234 | 0.360 | 0.388 | 0.326 | 0.317 |
| heart-switzerland | 123 | 12 | 5 | 0.720 | 0.672 | 0.711 | 0.677 | 0.639 | 0.680 | 0.685 | 0.618 |
| heart-va | 200 | 12 | 5 | 0.748 | 0.720 | 0.658 | 0.664 | 0.721 | 0.736 | 0.665 | 0.621 |
| hepatitis | 155 | 19 | 2 | 0.499 | 0.383 | 0.435 | 0.375 | 0.273 | 0.383 | 0.365 | 0.317 |
| hill-valley | 1212 | 100 | 2 | 0.654 | 0.540 | 0.472 | 0.502 | 0.591 | 0.541 | 0.545 | 0.540 |
| horse-colic | 368 | 25 | 2 | 0.422 | 0.407 | 0.399 | 0.245 | 0.369 | 0.407 | 0.289 | 0.225 |
| ilpd-indian-liver | 583 | 9 | 2 | 0.458 | 0.445 | 0.410 | 0.419 | 0.428 | 0.445 | 0.433 | 0.429 |
| image-segmentation | 2310 | 18 | 7 | 0.527 | 0.434 | 0.449 | 0.229 | 0.200 | 0.409 | 0.208 | 0.123 |
| ionosphere | 351 | 33 | 2 | 0.513 | 0.452 | 0.303 | 0.128 | 0.364 | 0.452 | 0.286 | 0.148 |
| iris | 150 | 4 | 3 | 0.623 | 0.553 | 0.684 | 0.105 | 0.173 | 0.511 | 0.337 | 0.189 |
| led-display | 1000 | 7 | 10 | 0.574 | 0.573 | 0.566 | 0.557 | 0.512 | 0.548 | 0.504 | 0.469 |
| lenses | 24 | 4 | 3 | 0.833 | 0.887 | 0.798 | 0.572 | 0.759 | 0.887 | 0.530 | 0.499 |
| letter | 20000 | 16 | 26 | 0.493 | 0.459 | 0.784 | 0.242 | 0.125 | 0.463 | 0.110 | 0.162 |
| libras | 360 | 90 | 15 | 0.586 | 0.485 | 0.639 | 0.287 | 0.282 | 0.449 | 0.482 | 0.258 |
| low-res-spect | 531 | 100 | 9 | 0.536 | 0.493 | 0.437 | 0.281 | 0.288 | 0.479 | 0.274 | 0.314 |
| lung-cancer | 32 | 56 | 3 | 0.870 | 0.784 | 0.609 | 0.633 | 0.812 | 0.836 | 0.572 | 0.479 |
| lymphography | 148 | 18 | 4 | 0.641 | 0.512 | 0.582 | 0.228 | 0.339 | 0.514 | 0.302 | 0.323 |
| magic | 19020 | 10 | 2 | 0.364 | 0.353 | 0.293 | 0.307 | 0.348 | 0.353 | 0.293 | 0.232 |
| mammographic | 961 | 5 | 2 | 0.458 | 0.456 | 0.277 | 0.399 | 0.441 | 0.456 | 0.402 | 0.384 |
| messidor | 1151 | 19 | 2 | 0.458 | 0.442 | 0.350 | 0.356 | 0.417 | 0.442 | 0.404 | 0.341 |
| miniboone | 130064 | 50 | 2 | 0.379 | 0.361 | 0.190 | 0.299 | 0.370 | 0.361 | 0.349 | 0.162 |
| molec-biol-promoter | 106 | 57 | 2 | 0.506 | 0.497 | 0.765 | 0.292 | 0.436 | 0.497 | 0.399 | 0.396 |
| molec-biol-splice | 3190 | 60 | 3 | 0.457 | 0.450 | 0.372 | 0.189 | 0.203 | 0.459 | 0.398 | 0.171 |
| monks-1 | 556 | 6 | 2 | 0.576 | 0.359 | 0.126 | 0.199 | 0.305 | 0.359 | 0.153 | 0.149 |
| monks-2 | 601 | 6 | 2 | 0.512 | 0.452 | 0.622 | 0.405 | 0.412 | 0.452 | 0.376 | 0.374 |
| monks-3 | 554 | 6 | 2 | 0.603 | 0.509 | 0.290 | 0.269 | 0.403 | 0.509 | 0.450 | 0.123 |
| mushroom | 8124 | 21 | 2 | NA | NA | NA | NA | NA | NA | NA | NA |

*Continued on next page.*

| Dataset | N | M | K | RED | MCP | Trust Score | ConfidNet | Intro-Net | Entropy | DNGO | SVGP |
|---|---|---|---|---|---|---|---|---|---|---|---|
| musk-1 | 476 | 166 | 2 | 0.482 | 0.470 | 0.476 | 0.081 | 0.319 | 0.470 | 0.273 | 0.089 |
| musk-2 | 6598 | 166 | 2 | 0.371 | 0.339 | 0.064 | 0.007 | 0.061 | 0.339 | 0.503 | 0.006 |
| nursery | 12960 | 8 | 5 | 0.550 | 0.388 | 0.070 | 0.027 | 0.129 | 0.386 | 0.368 | 0.131 |
| oocytes_merluccius_nucleus_4d | 1022 | 41 | 2 | 0.431 | 0.389 | 0.264 | 0.284 | 0.368 | 0.389 | 0.378 | 0.318 |
| oocytes_merluccius_states_2f | 1022 | 25 | 3 | 0.486 | 0.466 | 0.387 | 0.234 | 0.338 | 0.453 | 0.304 | 0.244 |
| oocytes_trisopterus_nucleus_2f | 912 | 25 | 2 | 0.470 | 0.439 | 0.329 | 0.262 | 0.447 | 0.439 | 0.349 | 0.285 |
| oocytes_trisopterus_states_5b | 912 | 32 | 3 | 0.420 | 0.394 | 0.321 | 0.137 | 0.266 | 0.417 | 0.316 | 0.101 |
| optical | 5620 | 62 | 10 | 0.437 | 0.411 | 0.572 | 0.043 | 0.037 | 0.402 | 0.070 | 0.067 |
| ozone | 2536 | 72 | 2 | 0.288 | 0.269 | 0.253 | 0.189 | 0.127 | 0.269 | 0.516 | 0.023 |
| page-blocks | 5473 | 10 | 5 | 0.439 | 0.374 | 0.388 | 0.317 | 0.263 | 0.359 | 0.255 | 0.173 |
| parkinsons | 195 | 22 | 2 | 0.490 | 0.456 | 0.477 | 0.108 | 0.197 | 0.456 | 0.264 | 0.277 |
| pendigits | 10992 | 16 | 10 | 0.480 | 0.344 | 0.677 | 0.168 | 0.124 | 0.343 | 0.102 | 0.050 |
| phishing | 1353 | 9 | 3 | 0.439 | 0.416 | 0.357 | 0.332 | 0.355 | 0.399 | 0.330 | 0.322 |
| pima | 768 | 8 | 2 | 0.489 | 0.480 | 0.408 | 0.399 | 0.477 | 0.480 | 0.414 | 0.322 |
| pittsburg-bridges-MATERIAL | 106 | 7 | 3 | 0.553 | 0.444 | 0.405 | 0.308 | 0.459 | 0.449 | 0.272 | 0.336 |
| pittsburg-bridges-REL-L | 103 | 7 | 3 | 0.565 | 0.498 | 0.624 | 0.485 | 0.519 | 0.491 | 0.464 | 0.452 |
| pittsburg-bridges-SPAN | 92 | 7 | 3 | 0.536 | 0.504 | 0.494 | 0.319 | 0.461 | 0.521 | 0.416 | 0.412 |
| pittsburg-bridges-T-OR-D | 102 | 7 | 2 | 0.388 | 0.299 | 0.376 | 0.320 | 0.462 | 0.299 | 0.298 | 0.274 |
| pittsburg-bridges-TYPE | 105 | 7 | 6 | 0.797 | 0.713 | 0.741 | 0.658 | 0.634 | 0.664 | 0.668 | 0.716 |
| planning | 182 | 12 | 2 | 0.339 | 0.337 | 0.362 | 0.315 | 0.295 | 0.337 | 0.281 | 0.229 |
| plant-margin | 1600 | 64 | 100 | 0.580 | 0.565 | 0.692 | 0.323 | 0.258 | 0.562 | 0.297 | 0.179 |
| plant-shape | 1600 | 64 | 100 | 0.690 | 0.662 | 0.757 | 0.519 | 0.475 | 0.639 | 0.513 | 0.472 |
| plant-texture | 1599 | 64 | 100 | 0.602 | 0.590 | 0.704 | 0.300 | 0.323 | 0.605 | 0.275 | 0.216 |
| post-operative | 90 | 8 | 3 | 0.391 | 0.375 | 0.230 | 0.329 | 0.368 | 0.334 | 0.398 | 0.368 |
| primary-tumor | 330 | 17 | 15 | 0.789 | 0.777 | 0.683 | 0.717 | 0.712 | 0.772 | 0.673 | 0.704 |
| ringnorm | 7400 | 20 | 2 | 0.334 | 0.329 | 0.269 | 0.040 | 0.111 | 0.329 | 0.285 | 0.025 |
| seeds | 210 | 7 | 3 | 0.515 | 0.447 | 0.453 | 0.282 | 0.284 | 0.405 | 0.340 | 0.314 |
| semeion | 1593 | 256 | 10 | 0.530 | 0.514 | 0.698 | 0.083 | 0.224 | 0.554 | 0.166 | 0.132 |
| soybean | 683 | 35 | 18 | 0.546 | 0.476 | 0.418 | 0.282 | 0.164 | 0.469 | 0.209 | 0.289 |
| spambase | 4601 | 57 | 2 | 0.307 | 0.282 | 0.290 | 0.170 | 0.228 | 0.282 | 0.211 | 0.131 |
| spect | 265 | 22 | 2 | 0.503 | 0.487 | 0.381 | 0.425 | 0.451 | 0.487 | 0.397 | 0.419 |
| spectf | 267 | 44 | 2 | 0.447 | 0.386 | 0.395 | 0.216 | 0.392 | 0.387 | 0.286 | 0.304 |
| statlog-australian-credit | 690 | 14 | 2 | 0.465 | 0.446 | 0.408 | 0.408 | 0.406 | 0.446 | 0.395 | 0.421 |
| statlog-german-credit | 1000 | 24 | 2 | 0.413 | 0.410 | 0.373 | 0.320 | 0.378 | 0.410 | 0.352 | 0.244 |
| statlog-heart | 270 | 13 | 2 | 0.443 | 0.406 | 0.351 | 0.288 | 0.435 | 0.406 | 0.329 | 0.255 |
| statlog-image | 2310 | 18 | 7 | 0.541 | 0.465 | 0.465 | 0.220 | 0.100 | 0.452 | 0.260 | 0.166 |
| statlog-landsat | 6435 | 36 | 6 | 0.425 | 0.393 | 0.551 | 0.317 | 0.306 | 0.404 | 0.291 | 0.251 |
| statlog-shuttle | 58000 | 9 | 7 | 0.534 | 0.294 | 0.527 | 0.214 | 0.183 | 0.281 | 0.347 | 0.229 |
| statlog-vehicle | 846 | 18 | 4 | 0.531 | 0.515 | 0.304 | 0.376 | 0.347 | 0.524 | 0.335 | 0.350 |
| steel-plates | 1941 | 27 | 7 | 0.523 | 0.511 | 0.528 | 0.446 | 0.430 | 0.508 | 0.432 | 0.397 |
| synthetic-control | 600 | 60 | 6 | 0.590 | 0.453 | 0.396 | 0.031 | 0.076 | 0.402 | 0.209 | 0.020 |
| teaching | 151 | 5 | 3 | 0.626 | 0.578 | 0.503 | 0.614 | 0.568 | 0.586 | 0.653 | 0.544 |
| thyroid | 7200 | 21 | 3 | 0.428 | 0.397 | 0.148 | 0.116 | 0.178 | 0.390 | 0.508 | 0.101 |
| tic-tac-toe | 958 | 9 | 2 | 0.562 | 0.473 | 1.000 | 0.132 | 0.461 | 0.473 | 0.423 | 0.418 |
| titanic | 2201 | 3 | 2 | 0.386 | 0.385 | 0.599 | 0.372 | 0.386 | 0.385 | 0.437 | 0.370 |
| trains | 10 | 29 | 2 | 0.667 | 0.625 | 0.875 | 0.750 | 0.625 | 0.625 | 0.438 | 0.625 |
| twonorm | 7400 | 20 | 2 | 0.359 | 0.345 | 0.394 | 0.152 | 0.278 | 0.345 | 0.270 | 0.034 |
| vertebral-column-2clases | 310 | 6 | 2 | 0.468 | 0.456 | 0.302 | 0.345 | 0.402 | 0.456 | 0.349 | 0.243 |
| vertebral-column-3clases | 310 | 6 | 3 | 0.505 | 0.478 | 0.279 | 0.460 | 0.366 | 0.495 | 0.411 | 0.271 |
| wall-following | 5456 | 24 | 4 | 0.414 | 0.373 | 0.299 | 0.204 | 0.243 | 0.364 | 0.263 | 0.135 |
| waveform | 5000 | 21 | 3 | 0.462 | 0.419 | 0.327 | 0.353 | 0.390 | 0.419 | 0.537 | 0.159 |
| waveform-noise | 5000 | 40 | 3 | 0.424 | 0.394 | 0.318 | 0.227 | 0.322 | 0.394 | 0.511 | 0.165 |
| wine | 178 | 13 | 3 | 0.857 | 0.669 | 0.606 | 0.056 | 0.413 | 0.794 | 0.602 | 0.181 |
| wine-quality-red | 1599 | 11 | 6 | 0.530 | 0.518 | 0.537 | 0.489 | 0.486 | 0.491 | 0.519 | 0.448 |
| wine-quality-white | 4898 | 11 | 7 | 0.539 | 0.527 | 0.625 | 0.532 | 0.521 | 0.514 | 0.523 | 0.478 |
| yeast | 1484 | 8 | 10 | 0.565 | 0.557 | 0.506 | 0.502 | 0.493 | 0.542 | 0.405 | 0.911 |
| zoo | 101 | 16 | 7 | 0.888 | 0.744 | 0.752 | 0.183 | 0.473 | 0.842 | 0.385 | 0.657 |

*Continued from previous page.*

Table 8: Comparison between RED and Counterparts Using AUPR-Success

| Dataset | N | M | K | RED | MCP | Trust Score | ConfidNet | Intro-Net | Entropy | DNGO | SVGP |
|---|---|---|---|---|---|---|---|---|---|---|---|
| abalone | 4177 | 8 | 3 | 0.852 | 0.850 | 0.821 | 0.844 | 0.844 | 0.849 | 0.806 | 0.840 |
| acute-inflammation | 120 | 6 | 2 | NA | NA | NA | NA | NA | NA | NA | NA |
| acute-nephritis | 120 | 6 | 2 | NA | NA | NA | NA | NA | NA | NA | NA |
| adult | 48842 | 14 | 2 | 0.968 | 0.967 | 0.944 | 0.964 | 0.961 | 0.967 | 0.897 | 0.961 |
| annealing | 898 | 31 | 5 | 0.975 | 0.903 | 0.942 | 0.907 | 0.914 | 0.899 | 0.908 | 0.919 |
| arrhythmia | 452 | 262 | 13 | 0.814 | 0.806 | 0.806 | 0.645 | 0.659 | 0.799 | 0.665 | 0.775 |
| audiology-std | 196 | 59 | 18 | 0.945 | 0.925 | NA | 0.609 | 0.725 | 0.925 | 0.741 | 0.769 |
| balance-scale | 625 | 4 | 3 | 0.999 | 0.999 | 0.997 | 0.971 | 0.983 | 0.999 | 0.986 | 0.976 |
| balloons | 16 | 4 | 2 | 0.773 | 0.736 | 0.861 | 0.690 | 0.736 | 0.736 | 0.516 | 0.542 |
| bank | 4521 | 16 | 2 | 0.977 | 0.980 | 0.976 | 0.968 | 0.936 | 0.980 | 0.946 | 0.979 |
| bioconcentration | 779 | 9 | 3 | 0.742 | 0.723 | 0.779 | 0.712 | 0.722 | 0.716 | 0.687 | 0.693 |
| blood | 748 | 4 | 2 | 0.891 | 0.891 | 0.872 | 0.901 | 0.882 | 0.891 | 0.881 | 0.899 |

*Continued on next page.*

| Dataset | N | M | K | RED | MCP | Trust Score | ConfidNet | Intro-Net | Entropy | DNGO | SVGP |
|---|---|---|---|---|---|---|---|---|---|---|---|
| breast-cancer | 286 | 9 | 2 | 0.815 | 0.814 | 0.757 | 0.788 | 0.767 | 0.814 | 0.758 | 0.777 |
| breast-cancer-wisc | 699 | 9 | 2 | 0.998 | 0.997 | 0.998 | 0.994 | 0.984 | 0.997 | 0.992 | 0.986 |
| breast-cancer-wisc-diag | 569 | 30 | 2 | 0.996 | 0.997 | 0.997 | 0.969 | 0.980 | 0.998 | 0.970 | 0.981 |
| breast-cancer-wisc-prog | 198 | 33 | 2 | 0.850 | 0.855 | 0.787 | 0.817 | 0.828 | 0.855 | 0.779 | 0.785 |
| breast-tissue | 106 | 9 | 6 | 0.902 | 0.869 | 0.893 | 0.855 | 0.751 | 0.863 | 0.782 | 0.838 |
| car | 1728 | 6 | 4 | 0.999 | 0.999 | 0.999 | 0.992 | 0.988 | 0.999 | 0.989 | 0.989 |
| cardiotocography-10clases | 2126 | 21 | 10 | 0.959 | 0.959 | 0.927 | 0.889 | 0.871 | 0.958 | 0.884 | 0.901 |
| cardiotocography-3clases | 2126 | 21 | 3 | 0.990 | 0.991 | 0.988 | 0.971 | 0.963 | 0.991 | 0.963 | 0.979 |
| chess-krvk | 28056 | 6 | 18 | 0.900 | 0.890 | 0.934 | 0.867 | 0.837 | 0.889 | 0.782 | 0.801 |
| chess-krvkp | 3196 | 36 | 2 | 1.000 | 1.000 | 0.995 | 0.990 | 0.994 | 1.000 | 0.972 | 0.996 |
| climate | 540 | 20 | 2 | 0.991 | 0.989 | 0.976 | 0.937 | 0.975 | 0.986 | 0.958 | 0.949 |
| congressional-voting | 435 | 16 | 2 | 0.701 | 0.701 | 0.706 | 0.704 | 0.694 | 0.701 | 0.713 | 0.698 |
| conn-bench-sonar-mines-rocks | 208 | 60 | 2 | 0.961 | 0.963 | 0.980 | 0.806 | 0.898 | 0.963 | 0.852 | 0.894 |
| conn-bench-vowel-deterding | 990 | 11 | 11 | 1.000 | 1.000 | 1.000 | 0.979 | 0.984 | 1.000 | 0.991 | 0.989 |
| connect-4 | 67557 | 42 | 2 | 0.969 | 0.969 | 0.938 | 0.942 | 0.958 | 0.969 | 0.883 | 0.867 |
| contrac | 1473 | 9 | 3 | 0.716 | 0.713 | 0.578 | 0.680 | 0.672 | 0.704 | 0.654 | 0.667 |
| credit-approval | 690 | 15 | 2 | 0.942 | 0.941 | 0.915 | 0.908 | 0.936 | 0.941 | 0.888 | 0.927 |
| cylinder-bands | 512 | 35 | 2 | 0.864 | 0.860 | 0.870 | 0.784 | 0.804 | 0.860 | 0.769 | 0.752 |
| dermatology | 366 | 34 | 6 | 0.999 | 0.999 | 0.997 | 0.975 | 0.980 | 0.999 | 0.976 | 0.950 |
| echocardiogram | 131 | 10 | 2 | 0.941 | 0.934 | 0.922 | 0.913 | 0.920 | 0.934 | 0.881 | 0.925 |
| ecoli | 336 | 7 | 8 | 0.958 | 0.955 | 0.969 | 0.908 | 0.874 | 0.953 | 0.881 | 0.916 |
| energy-y1 | 768 | 8 | 3 | 0.992 | 0.993 | 0.992 | 0.975 | 0.973 | 0.993 | 0.960 | 0.983 |
| energy-y2 | 768 | 8 | 3 | 0.986 | 0.986 | 0.985 | 0.961 | 0.945 | 0.986 | 0.960 | 0.978 |
| fertility | 100 | 9 | 2 | 0.951 | 0.923 | 0.946 | 0.914 | 0.924 | 0.923 | 0.893 | 0.936 |
| flags | 194 | 28 | 8 | 0.731 | 0.707 | 0.812 | 0.551 | 0.599 | 0.710 | 0.560 | 0.585 |
| glass | 214 | 9 | 6 | 0.850 | 0.804 | 0.912 | 0.767 | 0.725 | 0.802 | 0.729 | 0.775 |
| haberman-survival | 306 | 3 | 2 | 0.844 | 0.843 | 0.851 | 0.815 | 0.828 | 0.843 | 0.813 | 0.829 |
| hayes-roth | 160 | 3 | 3 | 0.932 | 0.880 | 0.965 | 0.882 | 0.854 | 0.885 | 0.864 | 0.821 |
| heart-cleveland | 303 | 13 | 5 | 0.853 | 0.856 | 0.832 | 0.792 | 0.818 | 0.858 | 0.683 | 0.835 |
| heart-hungarian | 294 | 12 | 2 | 0.923 | 0.905 | 0.926 | 0.841 | 0.912 | 0.905 | 0.838 | 0.862 |
| heart-switzerland | 123 | 12 | 5 | 0.464 | 0.440 | 0.429 | 0.399 | 0.414 | 0.448 | 0.435 | 0.408 |
| heart-va | 200 | 12 | 5 | 0.500 | 0.470 | 0.438 | 0.413 | 0.480 | 0.483 | 0.390 | 0.364 |
| hepatitis | 155 | 19 | 2 | 0.964 | 0.964 | 0.974 | 0.925 | 0.888 | 0.964 | 0.891 | 0.910 |
| hill-valley | 1212 | 100 | 2 | 0.761 | 0.702 | 0.554 | 0.650 | 0.739 | 0.702 | 0.702 | 0.690 |
| horse-colic | 368 | 25 | 2 | 0.926 | 0.916 | 0.917 | 0.827 | 0.911 | 0.916 | 0.825 | 0.869 |
| ilpd-indian-liver | 583 | 9 | 2 | 0.878 | 0.873 | 0.811 | 0.859 | 0.862 | 0.873 | 0.816 | 0.869 |
| image-segmentation | 2310 | 18 | 7 | 0.997 | 0.997 | 0.998 | 0.983 | 0.984 | 0.997 | 0.977 | 0.985 |
| ionosphere | 351 | 33 | 2 | 0.990 | 0.988 | 0.976 | 0.897 | 0.973 | 0.988 | 0.928 | 0.933 |
| iris | 150 | 4 | 3 | 0.992 | 0.991 | 0.997 | 0.935 | 0.912 | 0.990 | 0.945 | 0.954 |
| led-display | 1000 | 7 | 10 | 0.888 | 0.888 | 0.773 | 0.877 | 0.849 | 0.885 | 0.824 | 0.845 |
| lenses | 24 | 4 | 3 | 0.979 | 0.947 | 0.944 | 0.831 | 0.845 | 0.947 | 0.818 | 0.815 |
| letter | 20000 | 16 | 26 | 0.995 | 0.996 | 0.999 | 0.980 | 0.888 | 0.996 | 0.964 | 0.982 |
| libras | 360 | 90 | 15 | 0.955 | 0.942 | 0.955 | 0.802 | 0.800 | 0.940 | 0.859 | 0.848 |
| low-res-spect | 531 | 100 | 9 | 0.977 | 0.968 | 0.952 | 0.905 | 0.889 | 0.968 | 0.898 | 0.949 |
| lung-cancer | 32 | 56 | 3 | 0.728 | 0.687 | 0.572 | 0.461 | 0.703 | 0.688 | 0.393 | 0.356 |
| lymphography | 148 | 18 | 4 | 0.939 | 0.922 | 0.956 | 0.797 | 0.851 | 0.921 | 0.856 | 0.817 |
| magic | 19020 | 10 | 2 | 0.966 | 0.966 | 0.954 | 0.960 | 0.959 | 0.966 | 0.838 | 0.946 |
| mammographic | 961 | 5 | 2 | 0.922 | 0.918 | 0.881 | 0.909 | 0.910 | 0.918 | 0.865 | 0.925 |
| messidor | 1151 | 19 | 2 | 0.862 | 0.861 | 0.762 | 0.805 | 0.829 | 0.861 | 0.795 | 0.805 |
| miniboone | 130064 | 50 | 2 | 0.992 | 0.992 | 0.983 | 0.988 | 0.986 | 0.992 | 0.933 | 0.981 |
| molec-biol-promoter | 106 | 57 | 2 | 0.886 | 0.882 | 0.951 | 0.702 | 0.873 | 0.882 | 0.766 | 0.811 |
| molec-biol-splice | 3190 | 60 | 3 | 0.953 | 0.953 | 0.926 | 0.832 | 0.845 | 0.952 | 0.892 | 0.837 |
| monks-1 | 556 | 6 | 2 | 0.998 | 0.997 | 0.982 | 0.988 | 0.989 | 0.997 | 0.983 | 0.995 |
| monks-2 | 601 | 6 | 2 | 0.884 | 0.855 | 0.925 | 0.806 | 0.807 | 0.855 | 0.751 | 0.806 |
| monks-3 | 554 | 6 | 2 | 0.999 | 0.999 | 0.992 | 0.980 | 0.993 | 0.999 | 0.977 | 0.991 |
| mushroom | 8124 | 21 | 2 | NA | NA | NA | NA | NA | NA | NA | NA |
| musk-1 | 476 | 166 | 2 | 0.984 | 0.983 | 0.986 | 0.910 | 0.956 | 0.983 | 0.909 | 0.934 |
| musk-2 | 6598 | 166 | 2 | 1.000 | 1.000 | 0.998 | 0.995 | 0.995 | 1.000 | 0.997 | 0.994 |
| nursery | 12960 | 8 | 5 | 1.000 | 1.000 | 1.000 | 0.999 | 0.999 | 1.000 | 0.999 | 1.000 |
| oocytes_merluccius_nucleus_4d | 1022 | 41 | 2 | 0.939 | 0.929 | 0.881 | 0.891 | 0.903 | 0.929 | 0.881 | 0.907 |
| oocytes_merluccius_states_2f | 1022 | 25 | 3 | 0.991 | 0.991 | 0.991 | 0.967 | 0.970 | 0.991 | 0.952 | 0.975 |
| oocytes_trisopterus_nucleus_2f | 912 | 25 | 2 | 0.939 | 0.931 | 0.907 | 0.870 | 0.922 | 0.931 | 0.858 | 0.874 |
| oocytes_trisopterus_states_5b | 912 | 32 | 3 | 0.991 | 0.990 | 0.985 | 0.951 | 0.967 | 0.990 | 0.932 | 0.945 |
| optical | 5620 | 62 | 10 | 0.999 | 0.999 | 0.999 | 0.981 | 0.978 | 0.999 | 0.980 | 0.992 |
| ozone | 2536 | 72 | 2 | 0.994 | 0.996 | 0.993 | 0.988 | 0.966 | 0.996 | 0.984 | 0.951 |
| page-blocks | 5473 | 10 | 5 | 0.998 | 0.997 | 0.993 | 0.993 | 0.989 | 0.997 | 0.966 | 0.994 |
| parkinsons | 195 | 22 | 2 | 0.994 | 0.992 | 0.995 | 0.936 | 0.942 | 0.992 | 0.960 | 0.963 |
| pendigits | 10992 | 16 | 10 | 0.999 | 0.999 | 1.000 | 0.992 | 0.992 | 0.999 | 0.994 | 0.993 |
| phishing | 1353 | 9 | 3 | 0.975 | 0.973 | 0.958 | 0.959 | 0.945 | 0.972 | 0.919 | 0.952 |
| pima | 768 | 8 | 2 | 0.897 | 0.893 | 0.873 | 0.845 | 0.871 | 0.893 | 0.839 | 0.869 |
| pittsburg-bridges-MATERIAL | 106 | 7 | 3 | 0.975 | 0.968 | 0.950 | 0.916 | 0.928 | 0.968 | 0.881 | 0.936 |
| pittsburg-bridges-REL-L | 103 | 7 | 3 | 0.776 | 0.766 | 0.832 | 0.741 | 0.725 | 0.769 | 0.715 | 0.755 |
| pittsburg-bridges-SPAN | 92 | 7 | 3 | 0.842 | 0.793 | 0.766 | 0.649 | 0.734 | 0.790 | 0.767 | 0.695 |
| pittsburg-bridges-T-OR-D | 102 | 7 | 2 | 0.956 | 0.941 | 0.919 | 0.924 | 0.945 | 0.941 | 0.931 | 0.916 |
| pittsburg-bridges-TYPE | 105 | 7 | 6 | 0.805 | 0.658 | 0.718 | 0.612 | 0.596 | 0.631 | 0.713 | 0.748 |
| planning | 182 | 12 | 2 | 0.715 | 0.709 | 0.723 | 0.714 | 0.713 | 0.709 | 0.720 | 0.677 |
| plant-margin | 1600 | 64 | 100 | 0.951 | 0.949 | 0.966 | 0.827 | 0.816 | 0.950 | 0.815 | 0.729 |

*Continued on next page.*

| Dataset | N | M | K | RED | MCP | Trust Score | ConfidNet | Intro-Net | Entropy | DNGO | SVGP |
|---|---|---|---|---|---|---|---|---|---|---|---|
| plant-shape | 1600 | 64 | 100 | 0.860 | 0.849 | 0.879 | 0.715 | 0.668 | 0.837 | 0.693 | 0.669 |
| plant-texture | 1599 | 64 | 100 | 0.953 | 0.950 | 0.958 | 0.807 | 0.780 | 0.950 | 0.786 | 0.790 |
| post-operative | 90 | 8 | 3 | 0.653 | 0.642 | 0.719 | 0.656 | 0.660 | 0.639 | 0.702 | 0.654 |
| primary-tumor | 330 | 17 | 15 | 0.757 | 0.746 | 0.646 | 0.627 | 0.623 | 0.745 | 0.531 | 0.629 |
| ringnorm | 7400 | 20 | 2 | 0.999 | 0.999 | 0.989 | 0.982 | 0.977 | 0.999 | 0.959 | 0.976 |
| seeds | 210 | 7 | 3 | 0.987 | 0.986 | 0.991 | 0.962 | 0.960 | 0.985 | 0.929 | 0.945 |
| semeion | 1593 | 256 | 10 | 0.994 | 0.994 | 0.996 | 0.926 | 0.949 | 0.994 | 0.943 | 0.963 |
| soybean | 683 | 35 | 18 | 0.996 | 0.995 | 0.993 | 0.970 | 0.940 | 0.995 | 0.942 | 0.973 |
| spambase | 4601 | 57 | 2 | 0.989 | 0.988 | 0.983 | 0.965 | 0.966 | 0.988 | 0.941 | 0.974 |
| spect | 265 | 22 | 2 | 0.805 | 0.795 | 0.762 | 0.721 | 0.783 | 0.795 | 0.673 | 0.724 |
| spectf | 267 | 44 | 2 | 0.954 | 0.951 | 0.941 | 0.911 | 0.917 | 0.952 | 0.880 | 0.948 |
| statlog-australian-credit | 690 | 14 | 2 | 0.696 | 0.687 | 0.649 | 0.662 | 0.672 | 0.687 | 0.621 | 0.657 |
| statlog-german-credit | 1000 | 24 | 2 | 0.889 | 0.890 | 0.872 | 0.835 | 0.848 | 0.890 | 0.771 | 0.779 |
| statlog-heart | 270 | 13 | 2 | 0.939 | 0.934 | 0.904 | 0.878 | 0.914 | 0.934 | 0.829 | 0.860 |
| statlog-image | 2310 | 18 | 7 | 0.998 | 0.998 | 0.998 | 0.986 | 0.979 | 0.998 | 0.984 | 0.990 |
| statlog-landsat | 6435 | 36 | 6 | 0.986 | 0.987 | 0.991 | 0.963 | 0.959 | 0.987 | 0.948 | 0.971 |
| statlog-shuttle | 58000 | 9 | 7 | 1.000 | 1.000 | 1.000 | 0.998 | 0.907 | 0.999 | 0.999 | 0.999 |
| statlog-vehicle | 846 | 18 | 4 | 0.965 | 0.967 | 0.927 | 0.928 | 0.912 | 0.968 | 0.868 | 0.921 |
| steel-plates | 1941 | 27 | 7 | 0.919 | 0.916 | 0.928 | 0.881 | 0.848 | 0.915 | 0.824 | 0.870 |
| synthetic-control | 600 | 60 | 6 | 1.000 | 1.000 | 0.999 | 0.985 | 0.988 | 1.000 | 0.987 | 0.986 |
| teaching | 151 | 5 | 3 | 0.668 | 0.625 | 0.569 | 0.630 | 0.625 | 0.611 | 0.662 | 0.615 |
| thyroid | 7200 | 21 | 3 | 0.999 | 0.999 | 0.996 | 0.992 | 0.984 | 0.999 | 0.992 | 0.996 |
| tic-tac-toe | 958 | 9 | 2 | 1.000 | 0.999 | 1.000 | 0.999 | 0.998 | 0.999 | 0.994 | 0.999 |
| titanic | 2201 | 3 | 2 | 0.895 | 0.894 | 0.889 | 0.890 | 0.891 | 0.894 | 0.878 | 0.895 |
| trains | 10 | 29 | 2 | 0.667 | 0.625 | 0.875 | 0.750 | 0.625 | 0.625 | 0.438 | 0.625 |
| twonorm | 7400 | 20 | 2 | 0.998 | 0.999 | 0.998 | 0.991 | 0.991 | 0.999 | 0.924 | 0.973 |
| vertebral-column-2clases | 310 | 6 | 2 | 0.977 | 0.977 | 0.952 | 0.952 | 0.940 | 0.977 | 0.894 | 0.936 |
| vertebral-column-3clases | 310 | 6 | 3 | 0.977 | 0.977 | 0.943 | 0.958 | 0.931 | 0.977 | 0.908 | 0.952 |
| wall-following | 5456 | 24 | 4 | 0.980 | 0.980 | 0.972 | 0.928 | 0.949 | 0.980 | 0.941 | 0.932 |
| waveform | 5000 | 21 | 3 | 0.968 | 0.966 | 0.947 | 0.947 | 0.940 | 0.966 | 0.928 | 0.847 |
| waveform-noise | 5000 | 40 | 3 | 0.962 | 0.962 | 0.935 | 0.880 | 0.900 | 0.962 | 0.926 | 0.844 |
| wine | 178 | 13 | 3 | 0.999 | 0.999 | 0.998 | 0.945 | 0.990 | 0.999 | 0.968 | 0.978 |
| wine-quality-red | 1599 | 11 | 6 | 0.734 | 0.726 | 0.750 | 0.702 | 0.692 | 0.717 | 0.692 | 0.695 |
| wine-quality-white | 4898 | 11 | 7 | 0.672 | 0.662 | 0.745 | 0.648 | 0.654 | 0.655 | 0.650 | 0.610 |
| yeast | 1484 | 8 | 10 | 0.754 | 0.749 | 0.704 | 0.716 | 0.689 | 0.742 | 0.598 | 0.571 |
| zoo | 101 | 16 | 7 | 0.999 | 0.997 | 0.997 | 0.919 | 0.958 | 0.999 | 0.958 | 0.943 |

*Continued from previous page.*

Table 9: Comparison between RED and Counterparts Using AUROC

| Dataset | N | M | K | RED | MCP | Trust Score | ConfidNet | Intro-Net | Entropy | DNGO | SVGP |
|---|---|---|---|---|---|---|---|---|---|---|---|
| abalone | 4177 | 8 | 3 | 0.734 | 0.730 | 0.680 | 0.719 | 0.726 | 0.726 | 0.697 | 0.700 |
| acute-inflammation | 120 | 6 | 2 | NA | NA | NA | NA | NA | NA | NA | NA |
| acute-nephritis | 120 | 6 | 2 | NA | NA | NA | NA | NA | NA | NA | NA |
| adult | 48842 | 14 | 2 | 0.841 | 0.837 | 0.759 | 0.828 | 0.830 | 0.837 | 0.751 | 0.814 |
| annealing | 898 | 31 | 5 | 0.865 | 0.699 | 0.776 | 0.724 | 0.723 | 0.694 | 0.619 | 0.675 |
| arrhythmia | 452 | 262 | 13 | 0.712 | 0.715 | 0.727 | 0.505 | 0.517 | 0.703 | 0.528 | 0.639 |
| audiology-std | 196 | 59 | 18 | 0.883 | 0.848 | NA | 0.439 | 0.605 | 0.852 | 0.666 | 0.672 |
| balance-scale | 625 | 4 | 3 | 0.986 | 0.987 | 0.909 | 0.510 | 0.621 | 0.985 | 0.712 | 0.536 |
| balloons | 16 | 4 | 2 | 0.741 | 0.667 | 0.722 | 0.556 | 0.667 | 0.667 | 0.312 | 0.333 |
| bank | 4521 | 16 | 2 | 0.854 | 0.861 | 0.840 | 0.822 | 0.750 | 0.861 | 0.500 | 0.854 |
| bioconcentration | 779 | 9 | 3 | 0.621 | 0.607 | 0.660 | 0.572 | 0.595 | 0.588 | 0.547 | 0.541 |
| blood | 748 | 4 | 2 | 0.705 | 0.705 | 0.634 | 0.716 | 0.687 | 0.705 | 0.665 | 0.704 |
| breast-cancer | 286 | 9 | 2 | 0.682 | 0.677 | 0.587 | 0.637 | 0.609 | 0.677 | 0.607 | 0.620 |
| breast-cancer-wisc | 699 | 9 | 2 | 0.944 | 0.936 | 0.945 | 0.873 | 0.759 | 0.936 | 0.797 | 0.727 |
| breast-cancer-wisc-diag | 569 | 30 | 2 | 0.925 | 0.925 | 0.921 | 0.558 | 0.682 | 0.940 | 0.551 | 0.575 |
| breast-cancer-wisc-prog | 198 | 33 | 2 | 0.640 | 0.644 | 0.521 | 0.566 | 0.595 | 0.644 | 0.514 | 0.484 |
| breast-tissue | 106 | 9 | 6 | 0.864 | 0.815 | 0.850 | 0.788 | 0.651 | 0.811 | 0.761 | 0.785 |
| car | 1728 | 6 | 4 | 0.974 | 0.968 | 0.934 | 0.717 | 0.615 | 0.968 | 0.653 | 0.691 |
| cardiotocography-10clases | 2126 | 21 | 10 | 0.839 | 0.835 | 0.756 | 0.670 | 0.622 | 0.829 | 0.660 | 0.663 |
| cardiotocography-3clases | 2126 | 21 | 3 | 0.902 | 0.901 | 0.882 | 0.804 | 0.761 | 0.900 | 0.563 | 0.813 |
| chess-krvk | 28056 | 6 | 18 | 0.778 | 0.759 | 0.868 | 0.722 | 0.685 | 0.753 | 0.616 | 0.642 |
| chess-krvkp | 3196 | 36 | 2 | 0.967 | 0.963 | 0.777 | 0.573 | 0.642 | 0.963 | 0.568 | 0.688 |
| climate | 540 | 20 | 2 | 0.898 | 0.858 | 0.760 | 0.449 | 0.744 | 0.854 | 0.554 | 0.572 |
| congressional-voting | 435 | 16 | 2 | 0.603 | 0.600 | 0.562 | 0.607 | 0.595 | 0.598 | 0.619 | 0.612 |
| conn-bench-sonar-mines-rocks | 208 | 60 | 2 | 0.833 | 0.833 | 0.912 | 0.488 | 0.643 | 0.833 | 0.634 | 0.625 |
| conn-bench-vowel-deterding | 990 | 11 | 11 | 0.990 | 0.979 | 0.995 | 0.582 | 0.624 | 0.979 | 0.716 | 0.759 |
| connect-4 | 67557 | 42 | 2 | 0.836 | 0.831 | 0.744 | 0.727 | 0.816 | 0.831 | 0.736 | 0.522 |
| contrac | 1473 | 9 | 3 | 0.663 | 0.658 | 0.556 | 0.619 | 0.642 | 0.645 | 0.626 | 0.608 |
| credit-approval | 690 | 15 | 2 | 0.741 | 0.741 | 0.683 | 0.649 | 0.703 | 0.741 | 0.625 | 0.694 |
| cylinder-bands | 512 | 35 | 2 | 0.703 | 0.702 | 0.713 | 0.579 | 0.637 | 0.702 | 0.561 | 0.516 |
| dermatology | 366 | 34 | 6 | 0.980 | 0.974 | 0.934 | 0.589 | 0.766 | 0.976 | 0.750 | 0.333 |
| echocardiogram | 131 | 10 | 2 | 0.758 | 0.709 | 0.655 | 0.634 | 0.687 | 0.709 | 0.597 | 0.659 |
| ecoli | 336 | 7 | 8 | 0.791 | 0.785 | 0.806 | 0.670 | 0.556 | 0.773 | 0.592 | 0.667 |

*Continued on next page.*

| Dataset | N | M | K | RED | MCP | Trust Score | ConfidNet | Intro-Net | Entropy | DNGO | SVGP |
|---|---|---|---|---|---|---|---|---|---|---|---|
| energy-y1 | 768 | 8 | 3 | 0.896 | 0.886 | 0.848 | 0.691 | 0.650 | 0.886 | 0.604 | 0.777 |
| energy-y2 | 768 | 8 | 3 | 0.892 | 0.880 | 0.874 | 0.777 | 0.668 | 0.880 | 0.802 | 0.834 |
| fertility | 100 | 9 | 2 | 0.588 | 0.469 | 0.563 | 0.382 | 0.485 | 0.469 | 0.413 | 0.508 |
| flags | 194 | 28 | 8 | 0.727 | 0.696 | 0.791 | 0.603 | 0.618 | 0.698 | 0.598 | 0.574 |
| glass | 214 | 9 | 6 | 0.719 | 0.659 | 0.823 | 0.648 | 0.560 | 0.651 | 0.603 | 0.581 |
| haberman-survival | 306 | 3 | 2 | 0.694 | 0.692 | 0.651 | 0.645 | 0.671 | 0.692 | 0.636 | 0.684 |
| hayes-roth | 160 | 3 | 3 | 0.846 | 0.746 | 0.909 | 0.785 | 0.749 | 0.759 | 0.811 | 0.674 |
| heart-cleveland | 303 | 13 | 5 | 0.806 | 0.811 | 0.783 | 0.758 | 0.769 | 0.811 | 0.680 | 0.786 |
| heart-hungarian | 294 | 12 | 2 | 0.750 | 0.724 | 0.742 | 0.555 | 0.730 | 0.724 | 0.599 | 0.650 |
| heart-switzerland | 123 | 12 | 5 | 0.584 | 0.545 | 0.580 | 0.545 | 0.496 | 0.554 | 0.554 | 0.499 |
| heart-va | 200 | 12 | 5 | 0.629 | 0.606 | 0.544 | 0.522 | 0.603 | 0.618 | 0.505 | 0.455 |
| hepatitis | 155 | 19 | 2 | 0.822 | 0.797 | 0.845 | 0.703 | 0.619 | 0.797 | 0.624 | 0.677 |
| hill-valley | 1212 | 100 | 2 | 0.712 | 0.636 | 0.512 | 0.585 | 0.669 | 0.636 | 0.636 | 0.627 |
| horse-colic | 368 | 25 | 2 | 0.743 | 0.731 | 0.749 | 0.530 | 0.717 | 0.731 | 0.563 | 0.568 |
| ilpd-indian-liver | 583 | 9 | 2 | 0.719 | 0.709 | 0.650 | 0.686 | 0.695 | 0.709 | 0.614 | 0.709 |
| image-segmentation | 2310 | 18 | 7 | 0.935 | 0.936 | 0.946 | 0.770 | 0.742 | 0.935 | 0.734 | 0.749 |
| ionosphere | 351 | 33 | 2 | 0.909 | 0.886 | 0.779 | 0.504 | 0.786 | 0.886 | 0.630 | 0.535 |
| iris | 150 | 4 | 3 | 0.942 | 0.928 | 0.963 | 0.553 | 0.471 | 0.926 | 0.693 | 0.612 |
| led-display | 1000 | 7 | 10 | 0.762 | 0.762 | 0.545 | 0.749 | 0.717 | 0.753 | 0.706 | 0.700 |
| lenses | 24 | 4 | 3 | 0.917 | 0.875 | 0.863 | 0.677 | 0.740 | 0.875 | 0.625 | 0.583 |
| letter | 20000 | 16 | 26 | 0.932 | 0.940 | 0.977 | 0.795 | 0.526 | 0.940 | 0.646 | 0.775 |
| libras | 360 | 90 | 15 | 0.845 | 0.812 | 0.865 | 0.524 | 0.535 | 0.803 | 0.681 | 0.562 |
| low-res-spect | 531 | 100 | 9 | 0.864 | 0.837 | 0.791 | 0.618 | 0.559 | 0.836 | 0.616 | 0.717 |
| lung-cancer | 32 | 56 | 3 | 0.759 | 0.686 | 0.573 | 0.444 | 0.700 | 0.725 | 0.378 | 0.304 |
| lymphography | 148 | 18 | 4 | 0.805 | 0.753 | 0.825 | 0.503 | 0.622 | 0.748 | 0.623 | 0.566 |
| magic | 19020 | 10 | 2 | 0.810 | 0.807 | 0.753 | 0.788 | 0.796 | 0.807 | 0.639 | 0.720 |
| mammographic | 961 | 5 | 2 | 0.764 | 0.761 | 0.650 | 0.729 | 0.742 | 0.761 | 0.693 | 0.755 |
| messidor | 1151 | 19 | 2 | 0.697 | 0.690 | 0.565 | 0.609 | 0.662 | 0.690 | 0.646 | 0.601 |
| miniboone | 130064 | 50 | 2 | 0.905 | 0.903 | 0.806 | 0.878 | 0.882 | 0.903 | 0.781 | 0.783 |
| molec-biol-promoter | 106 | 57 | 2 | 0.735 | 0.727 | 0.885 | 0.414 | 0.719 | 0.727 | 0.505 | 0.576 |
| molec-biol-splice | 3190 | 60 | 3 | 0.810 | 0.810 | 0.722 | 0.517 | 0.547 | 0.811 | 0.717 | 0.501 |
| monks-1 | 556 | 6 | 2 | 0.959 | 0.936 | 0.619 | 0.715 | 0.779 | 0.936 | 0.597 | 0.862 |
| monks-2 | 601 | 6 | 2 | 0.751 | 0.698 | 0.814 | 0.616 | 0.632 | 0.698 | 0.602 | 0.598 |
| monks-3 | 554 | 6 | 2 | 0.972 | 0.966 | 0.845 | 0.750 | 0.872 | 0.966 | 0.820 | 0.825 |
| mushroom | 8124 | 21 | 2 | NA | NA | NA | NA | NA | NA | NA | NA |
| musk-1 | 476 | 166 | 2 | 0.863 | 0.860 | 0.863 | 0.449 | 0.724 | 0.860 | 0.635 | 0.503 |
| musk-2 | 6598 | 166 | 2 | 0.984 | 0.981 | 0.794 | 0.513 | 0.627 | 0.982 | 0.500 | 0.417 |
| nursery | 12960 | 8 | 5 | 0.993 | 0.955 | 0.925 | 0.601 | 0.603 | 0.965 | 0.674 | 0.832 |
| oocytes_merluccius_nucleus_4d | 1022 | 41 | 2 | 0.779 | 0.758 | 0.627 | 0.652 | 0.717 | 0.758 | 0.708 | 0.693 |
| oocytes_merluccius_states_2f | 1022 | 25 | 3 | 0.907 | 0.909 | 0.907 | 0.763 | 0.795 | 0.908 | 0.706 | 0.793 |
| oocytes_trisopterus_nucleus_2f | 912 | 25 | 2 | 0.786 | 0.766 | 0.693 | 0.619 | 0.759 | 0.766 | 0.665 | 0.638 |
| oocytes_trisopterus_states_5b | 912 | 32 | 3 | 0.898 | 0.895 | 0.834 | 0.638 | 0.744 | 0.895 | 0.704 | 0.546 |
| optical | 5620 | 62 | 10 | 0.970 | 0.964 | 0.973 | 0.561 | 0.534 | 0.964 | 0.573 | 0.703 |
| ozone | 2536 | 72 | 2 | 0.899 | 0.905 | 0.851 | 0.773 | 0.568 | 0.906 | 0.500 | 0.315 |
| page-blocks | 5473 | 10 | 5 | 0.952 | 0.942 | 0.905 | 0.906 | 0.855 | 0.941 | 0.792 | 0.861 |
| parkinsons | 195 | 22 | 2 | 0.918 | 0.881 | 0.928 | 0.574 | 0.561 | 0.881 | 0.729 | 0.666 |
| pendigits | 10992 | 16 | 10 | 0.951 | 0.948 | 0.972 | 0.681 | 0.635 | 0.947 | 0.650 | 0.660 |
| phishing | 1353 | 9 | 3 | 0.859 | 0.852 | 0.788 | 0.798 | 0.762 | 0.847 | 0.726 | 0.766 |
| pima | 768 | 8 | 2 | 0.754 | 0.751 | 0.700 | 0.673 | 0.730 | 0.751 | 0.697 | 0.653 |
| pittsburg-bridges-MATERIAL | 106 | 7 | 3 | 0.848 | 0.771 | 0.776 | 0.586 | 0.706 | 0.772 | 0.535 | 0.692 |
| pittsburg-bridges-REL-L | 103 | 7 | 3 | 0.643 | 0.611 | 0.739 | 0.612 | 0.587 | 0.614 | 0.569 | 0.604 |
| pittsburg-bridges-SPAN | 92 | 7 | 3 | 0.689 | 0.634 | 0.631 | 0.403 | 0.574 | 0.631 | 0.579 | 0.488 |
| pittsburg-bridges-T-OR-D | 102 | 7 | 2 | 0.778 | 0.740 | 0.694 | 0.669 | 0.750 | 0.740 | 0.712 | 0.653 |
| pittsburg-bridges-TYPE | 105 | 7 | 6 | 0.799 | 0.713 | 0.743 | 0.657 | 0.636 | 0.673 | 0.723 | 0.728 |
| planning | 182 | 12 | 2 | 0.479 | 0.468 | 0.509 | 0.492 | 0.473 | 0.468 | 0.474 | 0.400 |
| plant-margin | 1600 | 64 | 100 | 0.846 | 0.844 | 0.888 | 0.605 | 0.559 | 0.844 | 0.583 | 0.391 |
| plant-shape | 1600 | 64 | 100 | 0.794 | 0.780 | 0.823 | 0.647 | 0.587 | 0.760 | 0.626 | 0.586 |
| plant-texture | 1599 | 64 | 100 | 0.854 | 0.848 | 0.883 | 0.580 | 0.560 | 0.850 | 0.534 | 0.491 |
| post-operative | 90 | 8 | 3 | 0.443 | 0.433 | 0.414 | 0.433 | 0.446 | 0.410 | 0.517 | 0.480 |
| primary-tumor | 330 | 17 | 15 | 0.769 | 0.765 | 0.676 | 0.689 | 0.673 | 0.764 | 0.621 | 0.689 |
| ringnorm | 7400 | 20 | 2 | 0.953 | 0.952 | 0.708 | 0.587 | 0.565 | 0.952 | 0.623 | 0.507 |
| seeds | 210 | 7 | 3 | 0.913 | 0.890 | 0.913 | 0.746 | 0.724 | 0.881 | 0.717 | 0.610 |
| semeion | 1593 | 256 | 10 | 0.930 | 0.926 | 0.953 | 0.499 | 0.662 | 0.928 | 0.615 | 0.657 |
| soybean | 683 | 35 | 18 | 0.945 | 0.942 | 0.916 | 0.765 | 0.598 | 0.937 | 0.614 | 0.807 |
| spambase | 4601 | 57 | 2 | 0.864 | 0.856 | 0.809 | 0.710 | 0.724 | 0.856 | 0.692 | 0.713 |
| spect | 265 | 22 | 2 | 0.668 | 0.660 | 0.581 | 0.562 | 0.620 | 0.660 | 0.524 | 0.558 |
| spectf | 267 | 44 | 2 | 0.798 | 0.788 | 0.740 | 0.650 | 0.727 | 0.789 | 0.621 | 0.760 |
| statlog-australian-credit | 690 | 14 | 2 | 0.585 | 0.569 | 0.543 | 0.537 | 0.539 | 0.569 | 0.510 | 0.525 |
| statlog-german-credit | 1000 | 24 | 2 | 0.715 | 0.720 | 0.679 | 0.620 | 0.676 | 0.720 | 0.636 | 0.513 |
| statlog-heart | 270 | 13 | 2 | 0.762 | 0.746 | 0.693 | 0.624 | 0.731 | 0.746 | 0.580 | 0.565 |
| statlog-image | 2310 | 18 | 7 | 0.943 | 0.947 | 0.941 | 0.818 | 0.683 | 0.947 | 0.777 | 0.784 |
| statlog-landsat | 6435 | 36 | 6 | 0.888 | 0.887 | 0.919 | 0.795 | 0.770 | 0.889 | 0.755 | 0.787 |
| statlog-shuttle | 58000 | 9 | 7 | 0.995 | 0.919 | 0.998 | 0.763 | 0.589 | 0.911 | 0.599 | 0.778 |
| statlog-vehicle | 846 | 18 | 4 | 0.854 | 0.856 | 0.719 | 0.759 | 0.716 | 0.858 | 0.682 | 0.730 |
| steel-plates | 1941 | 27 | 7 | 0.795 | 0.790 | 0.805 | 0.736 | 0.695 | 0.787 | 0.688 | 0.692 |
| synthetic-control | 600 | 60 | 6 | 0.984 | 0.980 | 0.956 | 0.497 | 0.547 | 0.977 | 0.671 | 0.497 |
| teaching | 151 | 5 | 3 | 0.645 | 0.596 | 0.555 | 0.618 | 0.574 | 0.585 | 0.662 | 0.553 |

*Continued on next page.*

| Dataset | N | M | K | RED | MCP | Trust Score | ConfidNet | Intro-Net | Entropy | DNGO | SVGP |
|---|---|---|---|---|---|---|---|---|---|---|---|
| thyroid | 7200 | 21 | 3 | 0.970 | 0.963 | 0.849 | 0.782 | 0.646 | 0.968 | 0.500 | 0.809 |
| tic-tac-toe | 958 | 9 | 2 | 0.986 | 0.933 | 1.000 | 0.876 | 0.894 | 0.933 | 0.684 | 0.942 |
| titanic | 2201 | 3 | 2 | 0.673 | 0.671 | 0.508 | 0.662 | 0.671 | 0.671 | 0.553 | 0.674 |
| trains | 10 | 29 | 2 | 0.556 | 0.500 | 0.833 | 0.667 | 0.500 | 0.500 | 0.250 | 0.500 |
| twonorm | 7400 | 20 | 2 | 0.952 | 0.952 | 0.941 | 0.817 | 0.812 | 0.952 | 0.662 | 0.499 |
| vertebral-column-2clases | 310 | 6 | 2 | 0.860 | 0.857 | 0.770 | 0.771 | 0.765 | 0.857 | 0.735 | 0.707 |
| vertebral-column-3clases | 310 | 6 | 3 | 0.877 | 0.873 | 0.715 | 0.819 | 0.710 | 0.877 | 0.726 | 0.750 |
| wall-following | 5456 | 24 | 4 | 0.850 | 0.847 | 0.793 | 0.641 | 0.706 | 0.846 | 0.700 | 0.603 |
| waveform | 5000 | 21 | 3 | 0.846 | 0.834 | 0.761 | 0.773 | 0.791 | 0.833 | 0.563 | 0.500 |
| waveform-noise | 5000 | 40 | 3 | 0.823 | 0.819 | 0.727 | 0.601 | 0.692 | 0.818 | 0.581 | 0.505 |
| wine | 178 | 13 | 3 | 0.991 | 0.975 | 0.954 | 0.345 | 0.799 | 0.980 | 0.751 | 0.646 |
| wine-quality-red | 1599 | 11 | 6 | 0.647 | 0.638 | 0.644 | 0.615 | 0.607 | 0.617 | 0.625 | 0.573 |
| wine-quality-white | 4898 | 11 | 7 | 0.621 | 0.612 | 0.677 | 0.604 | 0.605 | 0.597 | 0.600 | 0.553 |
| yeast | 1484 | 8 | 10 | 0.683 | 0.678 | 0.620 | 0.627 | 0.611 | 0.665 | 0.486 | 0.792 |
| zoo | 101 | 16 | 7 | 0.986 | 0.970 | 0.958 | 0.659 | 0.779 | 0.987 | 0.594 | 0.783 |

*Continued from previous page.*

Table 10: Comparison between RED and Counterparts Using AP-Error

| Dataset | N | M | K | RED+BNN | BNN MCP | BNN Entropy | RED+MC-D | MC-D MCP | MC-D Entropy |
|---|---|---|---|---|---|---|---|---|---|
| abalone | 4177 | 8 | 3 | 0.529 | 0.522 | 0.513 | 0.516 | 0.509 | 0.503 |
| acute-inflammation | 120 | 6 | 2 | NA | NA | NA | NA | NA | NA |
| acute-nephritis | 120 | 6 | 2 | NA | NA | NA | NA | NA | NA |
| adult | 48842 | 14 | 2 | 0.415 | 0.407 | 0.406 | 0.416 | 0.405 | 0.405 |
| annealing | 898 | 31 | 5 | 0.788 | 0.424 | 0.338 | 0.590 | 0.421 | 0.375 |
| arrhythmia | 452 | 262 | 13 | 0.719 | 0.614 | 0.734 | 0.551 | 0.543 | 0.540 |
| audiology-std | 196 | 59 | 18 | 0.768 | 0.684 | 0.756 | 0.760 | 0.659 | 0.698 |
| balance-scale | 625 | 4 | 3 | 0.820 | 0.532 | 0.581 | 0.685 | 0.548 | 0.621 |
| balloons | 16 | 4 | 2 | 0.825 | 0.810 | 0.810 | 0.852 | 0.824 | 0.824 |
| bank | 4521 | 16 | 2 | 0.527 | 0.443 | 0.448 | 0.431 | 0.410 | 0.413 |
| bioconcentration | 779 | 9 | 3 | 0.562 | 0.487 | 0.486 | 0.462 | 0.440 | 0.418 |
| blood | 748 | 4 | 2 | 0.515 | 0.471 | 0.498 | 0.441 | 0.441 | 0.443 |
| breast-cancer | 286 | 9 | 2 | 0.504 | 0.396 | 0.421 | 0.518 | 0.486 | 0.497 |
| breast-cancer-wisc | 699 | 9 | 2 | 0.512 | 0.473 | 0.447 | 0.454 | 0.425 | 0.418 |
| breast-cancer-wisc-diag | 569 | 30 | 2 | 0.430 | 0.362 | 0.446 | 0.545 | 0.476 | 0.543 |
| breast-cancer-wisc-prog | 198 | 33 | 2 | 0.553 | 0.491 | 0.521 | 0.504 | 0.497 | 0.491 |
| breast-tissue | 106 | 9 | 6 | 0.847 | 0.777 | 0.797 | 0.885 | 0.841 | 0.832 |
| car | 1728 | 6 | 4 | 0.604 | 0.495 | 0.339 | 0.656 | 0.480 | 0.326 |
| cardiotocography-10clases | 2126 | 21 | 10 | 0.548 | 0.515 | 0.463 | 0.529 | 0.488 | 0.464 |
| cardiotocography-3clases | 2126 | 21 | 3 | 0.499 | 0.424 | 0.421 | 0.468 | 0.400 | 0.405 |
| chess-krvk | 28056 | 6 | 18 | 0.514 | 0.501 | 0.474 | 0.751 | 0.675 | 0.648 |
| chess-krvkp | 3196 | 36 | 2 | 0.392 | 0.369 | 0.385 | 0.443 | 0.417 | 0.424 |
| climate | 540 | 20 | 2 | 0.553 | 0.506 | 0.532 | 0.442 | 0.416 | 0.472 |
| congressional-voting | 435 | 16 | 2 | 0.517 | 0.466 | 0.472 | 0.486 | 0.485 | 0.496 |
| conn-bench-sonar-mines-rocks | 208 | 60 | 2 | 0.497 | 0.487 | 0.509 | 0.536 | 0.537 | 0.555 |
| conn-bench-vowel-deterding | 990 | 11 | 11 | 0.672 | 0.466 | 0.365 | 0.683 | 0.559 | 0.471 |
| connect-4 | 67557 | 42 | 2 | 0.408 | 0.402 | 0.401 | 0.490 | 0.410 | 0.407 |
| contrac | 1473 | 9 | 3 | 0.634 | 0.629 | 0.622 | 0.605 | 0.599 | 0.594 |
| credit-approval | 690 | 15 | 2 | 0.388 | 0.359 | 0.365 | 0.414 | 0.402 | 0.409 |
| cylinder-bands | 512 | 35 | 2 | 0.517 | 0.436 | 0.422 | 0.459 | 0.462 | 0.469 |
| dermatology | 366 | 34 | 6 | 0.633 | 0.637 | 0.510 | 0.757 | 0.726 | 0.645 |
| echocardiogram | 131 | 10 | 2 | 0.440 | 0.407 | 0.443 | 0.533 | 0.488 | 0.507 |
| ecoli | 336 | 7 | 8 | 0.520 | 0.484 | 0.496 | 0.416 | 0.410 | 0.411 |
| energy-y1 | 768 | 8 | 3 | 0.868 | 0.419 | 0.367 | 0.571 | 0.388 | 0.394 |
| energy-y2 | 768 | 8 | 3 | 0.709 | 0.411 | 0.347 | 0.542 | 0.464 | 0.437 |
| fertility | 100 | 9 | 2 | 0.213 | 0.178 | 0.205 | 0.329 | 0.299 | 0.301 |
| flags | 194 | 28 | 8 | 0.794 | 0.750 | 0.746 | 0.764 | 0.718 | 0.728 |
| glass | 214 | 9 | 6 | 0.664 | 0.511 | 0.499 | 0.541 | 0.480 | 0.476 |
| haberman-survival | 306 | 3 | 2 | 0.476 | 0.337 | 0.349 | 0.492 | 0.489 | 0.482 |
| hayes-roth | 160 | 3 | 3 | 0.671 | 0.568 | 0.560 | 0.713 | 0.572 | 0.594 |
| heart-cleveland | 303 | 13 | 5 | 0.772 | 0.744 | 0.760 | 0.709 | 0.687 | 0.695 |
| heart-hungarian | 294 | 12 | 2 | 0.494 | 0.460 | 0.463 | 0.417 | 0.387 | 0.411 |
| heart-switzerland | 123 | 12 | 5 | 0.755 | 0.700 | 0.656 | 0.647 | 0.619 | 0.585 |
| heart-va | 200 | 12 | 5 | 0.833 | 0.812 | 0.828 | 0.753 | 0.715 | 0.733 |
| hepatitis | 155 | 19 | 2 | 0.610 | 0.559 | 0.585 | 0.616 | 0.582 | 0.564 |
| hill-valley | 1212 | 100 | 2 | 0.513 | 0.497 | 0.533 | 0.570 | 0.565 | 0.548 |
| horse-colic | 368 | 25 | 2 | 0.393 | 0.343 | 0.365 | 0.379 | 0.372 | 0.397 |
| ilpd-indian-liver | 583 | 9 | 2 | 0.499 | 0.395 | 0.428 | 0.481 | 0.470 | 0.472 |
| image-segmentation | 2310 | 18 | 7 | 0.543 | 0.372 | 0.371 | 0.533 | 0.456 | 0.406 |
| ionosphere | 351 | 33 | 2 | 0.367 | 0.316 | 0.448 | 0.436 | 0.360 | 0.394 |
| iris | 150 | 4 | 3 | 0.821 | 0.675 | 0.466 | 0.735 | 0.694 | 0.705 |
| led-display | 1000 | 7 | 10 | 0.577 | 0.566 | 0.521 | 0.574 | 0.571 | 0.523 |
| lenses | 24 | 4 | 3 | 0.805 | 0.662 | 0.674 | 0.979 | 0.875 | 0.854 |
| letter | 20000 | 16 | 26 | 0.461 | 0.455 | 0.401 | 0.705 | 0.627 | 0.563 |

*Continued on next page.*

| Dataset | N | M | K | RED+BNN | BNN MCP | BNN Entropy | RED+MC-D | MC-D MCP | MC-D Entropy |
|---|---|---|---|---|---|---|---|---|---|
| libras | 360 | 90 | 15 | 0.698 | 0.587 | 0.507 | 0.554 | 0.433 | 0.377 |
| low-res-spect | 531 | 100 | 9 | 0.623 | 0.545 | 0.604 | 0.575 | 0.541 | 0.563 |
| lung-cancer | 32 | 56 | 3 | 0.775 | 0.734 | 0.716 | 0.863 | 0.805 | 0.773 |
| lymphography | 148 | 18 | 4 | 0.456 | 0.427 | 0.412 | 0.507 | 0.492 | 0.467 |
| magic | 19020 | 10 | 2 | 0.358 | 0.354 | 0.356 | 0.358 | 0.350 | 0.353 |
| mammographic | 961 | 5 | 2 | 0.444 | 0.434 | 0.425 | 0.435 | 0.431 | 0.432 |
| messidor | 1151 | 19 | 2 | 0.472 | 0.455 | 0.443 | 0.414 | 0.401 | 0.399 |
| miniboone | 130064 | 50 | 2 | 0.373 | 0.360 | 0.360 | 0.374 | 0.364 | 0.366 |
| molec-biol-promoter | 106 | 57 | 2 | 0.642 | 0.520 | 0.569 | 0.560 | 0.491 | 0.500 |
| molec-biol-splice | 3190 | 60 | 3 | 0.441 | 0.428 | 0.399 | 0.471 | 0.434 | 0.445 |
| monks-1 | 556 | 6 | 2 | 0.853 | 0.402 | 0.430 | 0.811 | 0.538 | 0.542 |
| monks-2 | 601 | 6 | 2 | 0.896 | 0.389 | 0.429 | 0.527 | 0.416 | 0.427 |
| monks-3 | 554 | 6 | 2 | 0.626 | 0.486 | 0.485 | 0.661 | 0.519 | 0.515 |
| mushroom | 8124 | 21 | 2 | NA | NA | NA | NA | NA | NA |
| musk-1 | 476 | 166 | 2 | 0.651 | 0.481 | 0.549 | 0.454 | 0.421 | 0.382 |
| musk-2 | 6598 | 166 | 2 | 0.540 | 0.389 | 0.447 | 0.456 | 0.351 | 0.406 |
| nursery | 12960 | 8 | 5 | 0.427 | 0.227 | 0.235 | 0.598 | 0.349 | 0.306 |
| oocytes_merluccius_nucleus_4d | 1022 | 41 | 2 | 0.813 | 0.487 | 0.470 | 0.407 | 0.387 | 0.387 |
| oocytes_merluccius_states_2f | 1022 | 25 | 3 | 0.464 | 0.404 | 0.358 | 0.449 | 0.426 | 0.468 |
| oocytes_trisopterus_nucleus_2f | 912 | 25 | 2 | 0.572 | 0.443 | 0.436 | 0.384 | 0.370 | 0.386 |
| oocytes_trisopterus_states_5b | 912 | 32 | 3 | 0.481 | 0.374 | 0.420 | 0.408 | 0.382 | 0.383 |
| optical | 5620 | 62 | 10 | 0.332 | 0.307 | 0.313 | 0.418 | 0.407 | 0.363 |
| ozone | 2536 | 72 | 2 | 0.321 | 0.240 | 0.285 | 0.350 | 0.351 | 0.337 |
| page-blocks | 5473 | 10 | 5 | 0.446 | 0.365 | 0.320 | 0.405 | 0.376 | 0.354 |
| parkinsons | 195 | 22 | 2 | 0.785 | 0.562 | 0.533 | 0.538 | 0.453 | 0.509 |
| pendigits | 10992 | 16 | 10 | 0.328 | 0.309 | 0.314 | 0.397 | 0.371 | 0.281 |
| phishing | 1353 | 9 | 3 | 0.604 | 0.451 | 0.454 | 0.559 | 0.471 | 0.444 |
| pima | 768 | 8 | 2 | 0.488 | 0.472 | 0.465 | 0.469 | 0.466 | 0.464 |
| pittsburg-bridges-MATERIAL | 106 | 7 | 3 | 0.485 | 0.410 | 0.460 | 0.548 | 0.507 | 0.500 |
| pittsburg-bridges-REL-L | 103 | 7 | 3 | 0.642 | 0.569 | 0.525 | 0.590 | 0.535 | 0.531 |
| pittsburg-bridges-SPAN | 92 | 7 | 3 | 0.593 | 0.579 | 0.622 | 0.545 | 0.498 | 0.520 |
| pittsburg-bridges-T-OR-D | 102 | 7 | 2 | 0.585 | 0.462 | 0.526 | 0.466 | 0.437 | 0.468 |
| pittsburg-bridges-TYPE | 105 | 7 | 6 | 0.834 | 0.758 | 0.755 | 0.784 | 0.725 | 0.720 |
| planning | 182 | 12 | 2 | 0.331 | 0.315 | 0.350 | 0.331 | 0.329 | 0.323 |
| plant-margin | 1600 | 64 | 100 | 0.623 | 0.603 | 0.509 | 0.614 | 0.603 | 0.476 |
| plant-shape | 1600 | 64 | 100 | 0.814 | 0.788 | 0.743 | 0.685 | 0.646 | 0.571 |
| plant-texture | 1599 | 64 | 100 | 0.659 | 0.650 | 0.556 | 0.654 | 0.624 | 0.503 |
| post-operative | 90 | 8 | 3 | 0.345 | 0.326 | 0.308 | 0.360 | 0.352 | 0.366 |
| primary-tumor | 330 | 17 | 15 | 0.880 | 0.854 | 0.845 | 0.809 | 0.803 | 0.801 |
| ringnorm | 7400 | 20 | 2 | 0.381 | 0.344 | 0.304 | 0.415 | 0.350 | 0.377 |
| seeds | 210 | 7 | 3 | 0.662 | 0.654 | 0.734 | 0.717 | 0.654 | 0.657 |
| semeion | 1593 | 256 | 10 | 0.552 | 0.538 | 0.479 | 0.538 | 0.514 | 0.499 |
| soybean | 683 | 35 | 18 | 0.555 | 0.522 | 0.393 | 0.534 | 0.519 | 0.410 |
| spambase | 4601 | 57 | 2 | 0.332 | 0.318 | 0.331 | 0.359 | 0.355 | 0.377 |
| spect | 265 | 22 | 2 | 0.473 | 0.427 | 0.418 | 0.518 | 0.513 | 0.518 |
| spectf | 267 | 44 | 2 | 0.784 | 0.581 | 0.575 | 0.430 | 0.362 | 0.355 |
| statlog-australian-credit | 690 | 14 | 2 | 0.471 | 0.360 | 0.347 | 0.446 | 0.414 | 0.421 |
| statlog-german-credit | 1000 | 24 | 2 | 0.602 | 0.547 | 0.557 | 0.418 | 0.412 | 0.422 |
| statlog-heart | 270 | 13 | 2 | 0.459 | 0.432 | 0.400 | 0.402 | 0.376 | 0.395 |
| statlog-image | 2310 | 18 | 7 | 0.548 | 0.422 | 0.351 | 0.508 | 0.460 | 0.403 |
| statlog-landsat | 6435 | 36 | 6 | 0.491 | 0.457 | 0.447 | 0.445 | 0.415 | 0.380 |
| statlog-shuttle | 58000 | 9 | 7 | 0.541 | 0.284 | 0.448 | 0.669 | 0.447 | 0.483 |
| statlog-vehicle | 846 | 18 | 4 | 0.546 | 0.532 | 0.439 | 0.526 | 0.493 | 0.450 |
| steel-plates | 1941 | 27 | 7 | 0.605 | 0.539 | 0.481 | 0.546 | 0.516 | 0.491 |
| synthetic-control | 600 | 60 | 6 | 0.439 | 0.310 | 0.713 | 0.584 | 0.426 | 0.463 |
| teaching | 151 | 5 | 3 | 0.727 | 0.694 | 0.693 | 0.699 | 0.650 | 0.626 |
| thyroid | 7200 | 21 | 3 | 0.561 | 0.452 | 0.488 | 0.547 | 0.489 | 0.478 |
| tic-tac-toe | 958 | 9 | 2 | 0.863 | 0.257 | 0.398 | 0.693 | 0.458 | 0.558 |
| titanic | 2201 | 3 | 2 | 0.367 | 0.346 | 0.338 | 0.344 | 0.344 | 0.346 |
| trains | 10 | 29 | 2 | 1.000 | 0.833 | 0.917 | 0.979 | 0.812 | 0.812 |
| twonorm | 7400 | 20 | 2 | 0.376 | 0.351 | 0.360 | 0.379 | 0.373 | 0.374 |
| vertebral-column-2clases | 310 | 6 | 2 | 0.549 | 0.502 | 0.509 | 0.477 | 0.463 | 0.476 |
| vertebral-column-3clases | 310 | 6 | 3 | 0.679 | 0.632 | 0.599 | 0.476 | 0.472 | 0.472 |
| wall-following | 5456 | 24 | 4 | 0.427 | 0.379 | 0.372 | 0.439 | 0.396 | 0.413 |
| waveform | 5000 | 21 | 3 | 0.418 | 0.395 | 0.368 | 0.432 | 0.424 | 0.392 |
| waveform-noise | 5000 | 40 | 3 | 0.441 | 0.430 | 0.398 | 0.440 | 0.429 | 0.406 |
| wine | 178 | 13 | 3 | 0.675 | 0.648 | 0.732 | 0.843 | 0.820 | 0.695 |
| wine-quality-red | 1599 | 11 | 6 | 0.578 | 0.532 | 0.521 | 0.521 | 0.502 | 0.492 |
| wine-quality-white | 4898 | 11 | 7 | 0.509 | 0.497 | 0.475 | 0.510 | 0.503 | 0.477 |
| yeast | 1484 | 8 | 10 | 0.605 | 0.597 | 0.576 | 0.596 | 0.590 | 0.549 |
| zoo | 101 | 16 | 7 | 0.907 | 0.821 | 0.808 | 0.931 | 0.802 | 0.847 |

*Continued from previous page.*

Table 11: Comparison between RED and Counterparts Using AP-Success

| Dataset | N | M | K | RED+BNN | BNN MCP | BNN Entropy | RED+MC-D | MC-D MCP | MC-D Entropy |
|---|---|---|---|---|---|---|---|---|---|
| abalone | 4177 | 8 | 3 | 0.855 | 0.854 | 0.849 | 0.854 | 0.852 | 0.851 |
| acute-inflammation | 120 | 6 | 2 | NA | NA | NA | NA | NA | NA |
| acute-nephritis | 120 | 6 | 2 | NA | NA | NA | NA | NA | NA |
| adult | 48842 | 14 | 2 | 0.970 | 0.969 | 0.970 | 0.970 | 0.969 | 0.970 |
| annealing | 898 | 31 | 5 | 0.983 | 0.950 | 0.947 | 0.981 | 0.971 | 0.974 |
| arrhythmia | 452 | 262 | 13 | 0.805 | 0.756 | 0.811 | 0.831 | 0.814 | 0.827 |
| audiology-std | 196 | 59 | 18 | 0.920 | 0.881 | 0.939 | 0.943 | 0.906 | 0.927 |
| balance-scale | 625 | 4 | 3 | 0.999 | 0.996 | 0.997 | 0.999 | 0.998 | 0.999 |
| balloons | 16 | 4 | 2 | 0.869 | 0.845 | 0.845 | 0.880 | 0.861 | 0.861 |
| bank | 4521 | 16 | 2 | 0.985 | 0.980 | 0.981 | 0.981 | 0.982 | 0.983 |
| bioconcentration | 779 | 9 | 3 | 0.767 | 0.752 | 0.746 | 0.757 | 0.748 | 0.738 |
| blood | 748 | 4 | 2 | 0.903 | 0.885 | 0.905 | 0.902 | 0.903 | 0.905 |
| breast-cancer | 286 | 9 | 2 | 0.834 | 0.779 | 0.828 | 0.829 | 0.826 | 0.838 |
| breast-cancer-wisc | 699 | 9 | 2 | 0.998 | 0.992 | 0.996 | 0.998 | 0.998 | 0.997 |
| breast-cancer-wisc-diag | 569 | 30 | 2 | 0.997 | 0.990 | 0.997 | 0.998 | 0.994 | 0.998 |
| breast-cancer-wisc-prog | 198 | 33 | 2 | 0.879 | 0.856 | 0.840 | 0.863 | 0.862 | 0.864 |
| breast-tissue | 106 | 9 | 6 | 0.932 | 0.909 | 0.912 | 0.935 | 0.914 | 0.925 |
| car | 1728 | 6 | 4 | 0.997 | 0.998 | 0.998 | 0.999 | 1.000 | 0.999 |
| cardiotocography-10clases | 2126 | 21 | 10 | 0.960 | 0.954 | 0.952 | 0.962 | 0.956 | 0.955 |
| cardiotocography-3clases | 2126 | 21 | 3 | 0.990 | 0.988 | 0.989 | 0.992 | 0.991 | 0.992 |
| chess-krvk | 28056 | 6 | 18 | 0.923 | 0.919 | 0.912 | 0.806 | 0.753 | 0.738 |
| chess-krvkp | 3196 | 36 | 2 | 0.999 | 0.998 | 1.000 | 1.000 | 0.999 | 1.000 |
| climate | 540 | 20 | 2 | 0.989 | 0.986 | 0.993 | 0.987 | 0.987 | 0.990 |
| congressional-voting | 435 | 16 | 2 | 0.749 | 0.704 | 0.725 | 0.720 | 0.719 | 0.724 |
| conn-bench-sonar-mines-rocks | 208 | 60 | 2 | 0.940 | 0.931 | 0.953 | 0.955 | 0.951 | 0.957 |
| conn-bench-vowel-deterding | 990 | 11 | 11 | 0.998 | 0.994 | 0.990 | 0.998 | 0.996 | 0.996 |
| connect-4 | 67557 | 42 | 2 | 0.975 | 0.975 | 0.975 | 0.959 | 0.954 | 0.954 |
| contrac | 1473 | 9 | 3 | 0.744 | 0.741 | 0.739 | 0.741 | 0.740 | 0.731 |
| credit-approval | 690 | 15 | 2 | 0.953 | 0.937 | 0.936 | 0.945 | 0.942 | 0.943 |
| cylinder-bands | 512 | 35 | 2 | 0.848 | 0.822 | 0.832 | 0.871 | 0.865 | 0.874 |
| dermatology | 366 | 34 | 6 | 0.998 | 0.992 | 0.999 | 1.000 | 1.000 | 0.999 |
| echocardiogram | 131 | 10 | 2 | 0.947 | 0.935 | 0.923 | 0.951 | 0.946 | 0.950 |
| ecoli | 336 | 7 | 8 | 0.958 | 0.950 | 0.964 | 0.957 | 0.956 | 0.959 |
| energy-y1 | 768 | 8 | 3 | 0.997 | 0.978 | 0.979 | 0.996 | 0.993 | 0.993 |
| energy-y2 | 768 | 8 | 3 | 0.993 | 0.973 | 0.973 | 0.990 | 0.986 | 0.987 |
| fertility | 100 | 9 | 2 | 0.959 | 0.923 | 0.928 | 0.962 | 0.935 | 0.939 |
| flags | 194 | 28 | 8 | 0.743 | 0.694 | 0.716 | 0.756 | 0.708 | 0.728 |
| glass | 214 | 9 | 6 | 0.837 | 0.732 | 0.747 | 0.837 | 0.793 | 0.829 |
| haberman-survival | 306 | 3 | 2 | 0.873 | 0.778 | 0.801 | 0.851 | 0.847 | 0.848 |
| hayes-roth | 160 | 3 | 3 | 0.974 | 0.969 | 0.963 | 0.955 | 0.904 | 0.905 |
| heart-cleveland | 303 | 13 | 5 | 0.893 | 0.881 | 0.886 | 0.867 | 0.865 | 0.869 |
| heart-hungarian | 294 | 12 | 2 | 0.934 | 0.913 | 0.929 | 0.926 | 0.908 | 0.911 |
| heart-switzerland | 123 | 12 | 5 | 0.588 | 0.539 | 0.517 | 0.502 | 0.476 | 0.469 |
| heart-va | 200 | 12 | 5 | 0.415 | 0.349 | 0.357 | 0.468 | 0.438 | 0.436 |
| hepatitis | 155 | 19 | 2 | 0.968 | 0.950 | 0.972 | 0.959 | 0.966 | 0.968 |
| hill-valley | 1212 | 100 | 2 | 0.523 | 0.509 | 0.525 | 0.551 | 0.548 | 0.543 |
| horse-colic | 368 | 25 | 2 | 0.927 | 0.893 | 0.911 | 0.906 | 0.896 | 0.903 |
| ilpd-indian-liver | 583 | 9 | 2 | 0.873 | 0.845 | 0.853 | 0.880 | 0.876 | 0.876 |
| image-segmentation | 2310 | 18 | 7 | 0.997 | 0.993 | 0.997 | 0.998 | 0.997 | 0.998 |
| ionosphere | 351 | 33 | 2 | 0.984 | 0.965 | 0.982 | 0.991 | 0.991 | 0.991 |
| iris | 150 | 4 | 3 | 0.997 | 0.994 | 0.993 | 0.997 | 0.996 | 0.997 |
| led-display | 1000 | 7 | 10 | 0.886 | 0.882 | 0.870 | 0.887 | 0.885 | 0.879 |
| lenses | 24 | 4 | 3 | 0.893 | 0.820 | 0.832 | 0.990 | 0.945 | 0.935 |
| letter | 20000 | 16 | 26 | 0.996 | 0.997 | 0.997 | 0.988 | 0.985 | 0.980 |
| libras | 360 | 90 | 15 | 0.879 | 0.847 | 0.732 | 0.952 | 0.939 | 0.925 |
| low-res-spect | 531 | 100 | 9 | 0.973 | 0.966 | 0.979 | 0.979 | 0.974 | 0.984 |
| lung-cancer | 32 | 56 | 3 | 0.715 | 0.685 | 0.695 | 0.727 | 0.648 | 0.592 |
| lymphography | 148 | 18 | 4 | 0.950 | 0.933 | 0.943 | 0.932 | 0.936 | 0.934 |
| magic | 19020 | 10 | 2 | 0.969 | 0.968 | 0.968 | 0.968 | 0.967 | 0.967 |
| mammographic | 961 | 5 | 2 | 0.925 | 0.909 | 0.916 | 0.922 | 0.919 | 0.920 |
| messidor | 1151 | 19 | 2 | 0.812 | 0.809 | 0.810 | 0.872 | 0.871 | 0.870 |
| miniboone | 130064 | 50 | 2 | 0.993 | 0.992 | 0.993 | 0.992 | 0.992 | 0.993 |
| molec-biol-promoter | 106 | 57 | 2 | 0.934 | 0.838 | 0.908 | 0.928 | 0.901 | 0.901 |
| molec-biol-splice | 3190 | 60 | 3 | 0.978 | 0.976 | 0.975 | 0.970 | 0.969 | 0.972 |
| monks-1 | 556 | 6 | 2 | 1.000 | 0.996 | 0.997 | 1.000 | 0.999 | 0.999 |
| monks-2 | 601 | 6 | 2 | 0.964 | 0.809 | 0.818 | 0.871 | 0.822 | 0.822 |
| monks-3 | 554 | 6 | 2 | 0.997 | 0.996 | 0.996 | 0.999 | 0.997 | 0.997 |
| mushroom | 8124 | 21 | 2 | NA | NA | NA | NA | NA | NA |
| musk-1 | 476 | 166 | 2 | 0.938 | 0.915 | 0.919 | 0.980 | 0.976 | 0.980 |
| musk-2 | 6598 | 166 | 2 | 0.999 | 0.998 | 0.998 | 1.000 | 0.999 | 1.000 |
| nursery | 12960 | 8 | 5 | 1.000 | 1.000 | 1.000 | 1.000 | 1.000 | 1.000 |
| oocytes_merluccius_nucleus_4d | 1022 | 41 | 2 | 0.955 | 0.781 | 0.782 | 0.942 | 0.936 | 0.940 |
| oocytes_merluccius_states_2f | 1022 | 25 | 3 | 0.987 | 0.983 | 0.985 | 0.991 | 0.988 | 0.991 |
| oocytes_trisopterus_nucleus_2f | 912 | 25 | 2 | 0.914 | 0.850 | 0.866 | 0.936 | 0.926 | 0.934 |
| oocytes_trisopterus_states_5b | 912 | 32 | 3 | 0.985 | 0.977 | 0.980 | 0.993 | 0.991 | 0.993 |

*Continued on next page.*

| Dataset | N | M | K | RED+BNN | BNN MCP | BNN Entropy | RED+MC-D | MC-D MCP | MC-D Entropy |
|---|---|---|---|---|---|---|---|---|---|
| optical | 5620 | 62 | 10 | 0.997 | 0.997 | 0.999 | 0.999 | 0.998 | 0.999 |
| ozone | 2536 | 72 | 2 | 0.996 | 0.993 | 0.995 | 0.995 | 0.993 | 0.996 |
| page-blocks | 5473 | 10 | 5 | 0.998 | 0.997 | 0.998 | 0.998 | 0.998 | 0.998 |
| parkinsons | 195 | 22 | 2 | 0.988 | 0.975 | 0.977 | 0.986 | 0.987 | 0.987 |
| pendigits | 10992 | 16 | 10 | 0.999 | 0.999 | 0.999 | 0.999 | 0.999 | 0.999 |
| phishing | 1353 | 9 | 3 | 0.974 | 0.967 | 0.967 | 0.979 | 0.976 | 0.974 |
| pima | 768 | 8 | 2 | 0.897 | 0.890 | 0.906 | 0.901 | 0.899 | 0.904 |
| pittsburg-bridges-MATERIAL | 106 | 7 | 3 | 0.978 | 0.963 | 0.971 | 0.976 | 0.973 | 0.973 |
| pittsburg-bridges-REL-L | 103 | 7 | 3 | 0.809 | 0.772 | 0.731 | 0.798 | 0.786 | 0.791 |
| pittsburg-bridges-SPAN | 92 | 7 | 3 | 0.856 | 0.846 | 0.837 | 0.851 | 0.808 | 0.813 |
| pittsburg-bridges-T-OR-D | 102 | 7 | 2 | 0.956 | 0.910 | 0.937 | 0.965 | 0.944 | 0.945 |
| pittsburg-bridges-TYPE | 105 | 7 | 6 | 0.800 | 0.719 | 0.764 | 0.783 | 0.725 | 0.713 |
| planning | 182 | 12 | 2 | 0.775 | 0.761 | 0.774 | 0.713 | 0.707 | 0.699 |
| plant-margin | 1600 | 64 | 100 | 0.935 | 0.934 | 0.909 | 0.954 | 0.952 | 0.919 |
| plant-shape | 1600 | 64 | 100 | 0.714 | 0.685 | 0.622 | 0.874 | 0.857 | 0.804 |
| plant-texture | 1599 | 64 | 100 | 0.947 | 0.945 | 0.921 | 0.947 | 0.940 | 0.902 |
| post-operative | 90 | 8 | 3 | 0.685 | 0.664 | 0.666 | 0.692 | 0.688 | 0.688 |
| primary-tumor | 330 | 17 | 15 | 0.782 | 0.748 | 0.755 | 0.759 | 0.751 | 0.753 |
| ringnorm | 7400 | 20 | 2 | 0.999 | 0.999 | 0.999 | 0.998 | 0.997 | 0.998 |
| seeds | 210 | 7 | 3 | 0.980 | 0.976 | 0.991 | 0.990 | 0.986 | 0.985 |
| semeion | 1593 | 256 | 10 | 0.992 | 0.991 | 0.992 | 0.992 | 0.993 | 0.993 |
| soybean | 683 | 35 | 18 | 0.996 | 0.993 | 0.993 | 0.996 | 0.996 | 0.992 |
| spambase | 4601 | 57 | 2 | 0.987 | 0.986 | 0.990 | 0.989 | 0.988 | 0.992 |
| spect | 265 | 22 | 2 | 0.778 | 0.753 | 0.777 | 0.792 | 0.780 | 0.782 |
| spectf | 267 | 44 | 2 | 0.965 | 0.951 | 0.955 | 0.959 | 0.949 | 0.950 |
| statlog-australian-credit | 690 | 14 | 2 | 0.744 | 0.669 | 0.664 | 0.721 | 0.705 | 0.705 |
| statlog-german-credit | 1000 | 24 | 2 | 0.904 | 0.899 | 0.900 | 0.892 | 0.897 | 0.899 |
| statlog-heart | 270 | 13 | 2 | 0.937 | 0.926 | 0.930 | 0.933 | 0.926 | 0.931 |
| statlog-image | 2310 | 18 | 7 | 0.998 | 0.997 | 0.997 | 0.998 | 0.998 | 0.998 |
| statlog-landsat | 6435 | 36 | 6 | 0.984 | 0.982 | 0.981 | 0.985 | 0.985 | 0.985 |
| statlog-shuttle | 58000 | 9 | 7 | 1.000 | 0.999 | 1.000 | 1.000 | 1.000 | 1.000 |
| statlog-vehicle | 846 | 18 | 4 | 0.944 | 0.940 | 0.914 | 0.970 | 0.967 | 0.965 |
| steel-plates | 1941 | 27 | 7 | 0.923 | 0.911 | 0.900 | 0.923 | 0.913 | 0.916 |
| synthetic-control | 600 | 60 | 6 | 0.998 | 0.994 | 1.000 | 1.000 | 0.999 | 0.999 |
| teaching | 151 | 5 | 3 | 0.616 | 0.580 | 0.548 | 0.709 | 0.648 | 0.640 |
| thyroid | 7200 | 21 | 3 | 1.000 | 0.999 | 1.000 | 0.999 | 0.999 | 1.000 |
| tic-tac-toe | 958 | 9 | 2 | 1.000 | 0.998 | 0.999 | 1.000 | 0.999 | 0.999 |
| titanic | 2201 | 3 | 2 | 0.890 | 0.884 | 0.884 | 0.890 | 0.890 | 0.891 |
| trains | 10 | 29 | 2 | 1.000 | 0.833 | 0.917 | 0.979 | 0.812 | 0.812 |
| twonorm | 7400 | 20 | 2 | 0.999 | 0.999 | 0.999 | 0.999 | 0.998 | 0.999 |
| vertebral-column-2clases | 310 | 6 | 2 | 0.976 | 0.962 | 0.967 | 0.976 | 0.975 | 0.977 |
| vertebral-column-3clases | 310 | 6 | 3 | 0.971 | 0.965 | 0.965 | 0.976 | 0.976 | 0.976 |
| wall-following | 5456 | 24 | 4 | 0.990 | 0.987 | 0.990 | 0.992 | 0.992 | 0.994 |
| waveform | 5000 | 21 | 3 | 0.973 | 0.970 | 0.970 | 0.973 | 0.973 | 0.972 |
| waveform-noise | 5000 | 40 | 3 | 0.975 | 0.974 | 0.973 | 0.971 | 0.971 | 0.971 |
| wine | 178 | 13 | 3 | 0.997 | 0.990 | 0.999 | 0.999 | 0.998 | 0.999 |
| wine-quality-red | 1599 | 11 | 6 | 0.726 | 0.699 | 0.713 | 0.737 | 0.729 | 0.727 |
| wine-quality-white | 4898 | 11 | 7 | 0.640 | 0.626 | 0.609 | 0.665 | 0.654 | 0.635 |
| yeast | 1484 | 8 | 10 | 0.724 | 0.719 | 0.717 | 0.749 | 0.744 | 0.721 |
| zoo | 101 | 16 | 7 | 0.998 | 0.996 | 0.997 | 0.998 | 0.996 | 0.995 |

*Continued from previous page.*

Table 12: Comparison between RED and Counterparts Using AUPR-Error

| Dataset | N | M | K | RED+BNN | BNN MCP | BNN Entropy | RED+MC-D | MC-D MCP | MC-D Entropy |
|---|---|---|---|---|---|---|---|---|---|
| abalone | 4177 | 8 | 3 | 0.526 | 0.519 | 0.510 | 0.513 | 0.506 | 0.501 |
| acute-inflammation | 120 | 6 | 2 | NA | NA | NA | NA | NA | NA |
| acute-nephritis | 120 | 6 | 2 | NA | NA | NA | NA | NA | NA |
| adult | 48842 | 14 | 2 | 0.414 | 0.406 | 0.405 | 0.415 | 0.405 | 0.404 |
| annealing | 898 | 31 | 5 | 0.785 | 0.414 | 0.327 | 0.580 | 0.401 | 0.353 |
| arrhythmia | 452 | 262 | 13 | 0.713 | 0.603 | 0.726 | 0.539 | 0.531 | 0.529 |
| audiology-std | 196 | 59 | 18 | 0.758 | 0.683 | 0.744 | 0.748 | 0.637 | 0.674 |
| balance-scale | 625 | 4 | 3 | 0.790 | 0.474 | 0.518 | 0.656 | 0.519 | 0.580 |
| balloons | 16 | 4 | 2 | 0.752 | 0.744 | 0.744 | 0.801 | 0.759 | 0.759 |
| bank | 4521 | 16 | 2 | 0.523 | 0.438 | 0.442 | 0.426 | 0.403 | 0.405 |
| bioconcentration | 779 | 9 | 3 | 0.557 | 0.477 | 0.477 | 0.454 | 0.427 | 0.408 |
| blood | 748 | 4 | 2 | 0.505 | 0.457 | 0.485 | 0.429 | 0.428 | 0.430 |
| breast-cancer | 286 | 9 | 2 | 0.488 | 0.372 | 0.394 | 0.498 | 0.459 | 0.476 |
| breast-cancer-wisc | 699 | 9 | 2 | 0.475 | 0.440 | 0.381 | 0.399 | 0.365 | 0.360 |
| breast-cancer-wisc-diag | 569 | 30 | 2 | 0.362 | 0.333 | 0.379 | 0.511 | 0.441 | 0.500 |
| breast-cancer-wisc-prog | 198 | 33 | 2 | 0.528 | 0.460 | 0.500 | 0.477 | 0.469 | 0.460 |
| breast-tissue | 106 | 9 | 6 | 0.830 | 0.751 | 0.772 | 0.877 | 0.825 | 0.813 |
| car | 1728 | 6 | 4 | 0.591 | 0.475 | 0.316 | 0.628 | 0.418 | 0.277 |
| cardiotocography-10clases | 2126 | 21 | 10 | 0.542 | 0.509 | 0.457 | 0.524 | 0.482 | 0.457 |

*Continued on next page.*

| Dataset | N | M | K | RED+BNN | BNN MCP | BNN Entropy | RED+MC-D | MC-D MCP | MC-D Entropy |
|---|---|---|---|---|---|---|---|---|---|
| cardiotocography-3clases | 2126 | 21 | 3 | 0.490 | 0.413 | 0.409 | 0.458 | 0.388 | 0.392 |
| chess-krvk | 28056 | 6 | 18 | 0.514 | 0.500 | 0.473 | 0.751 | 0.674 | 0.648 |
| chess-krvkp | 3196 | 36 | 2 | 0.338 | 0.320 | 0.312 | 0.395 | 0.373 | 0.378 |
| climate | 540 | 20 | 2 | 0.523 | 0.486 | 0.489 | 0.406 | 0.374 | 0.431 |
| congressional-voting | 435 | 16 | 2 | 0.503 | 0.450 | 0.457 | 0.471 | 0.470 | 0.481 |
| conn-bench-sonar-mines-rocks | 208 | 60 | 2 | 0.463 | 0.460 | 0.466 | 0.494 | 0.496 | 0.510 |
| conn-bench-vowel-deterding | 990 | 11 | 11 | 0.648 | 0.439 | 0.337 | 0.665 | 0.541 | 0.435 |
| connect-4 | 67557 | 42 | 2 | 0.407 | 0.401 | 0.401 | 0.490 | 0.409 | 0.407 |
| contrac | 1473 | 9 | 3 | 0.631 | 0.626 | 0.619 | 0.601 | 0.595 | 0.590 |
| credit-approval | 690 | 15 | 2 | 0.367 | 0.334 | 0.344 | 0.393 | 0.380 | 0.386 |
| cylinder-bands | 512 | 35 | 2 | 0.506 | 0.423 | 0.406 | 0.443 | 0.447 | 0.454 |
| dermatology | 366 | 34 | 6 | 0.593 | 0.646 | 0.423 | 0.682 | 0.639 | 0.535 |
| echocardiogram | 131 | 10 | 2 | 0.394 | 0.360 | 0.378 | 0.477 | 0.435 | 0.452 |
| ecoli | 336 | 7 | 8 | 0.491 | 0.454 | 0.462 | 0.389 | 0.383 | 0.371 |
| energy-y1 | 768 | 8 | 3 | 0.861 | 0.398 | 0.346 | 0.552 | 0.353 | 0.365 |
| energy-y2 | 768 | 8 | 3 | 0.698 | 0.392 | 0.330 | 0.521 | 0.445 | 0.410 |
| fertility | 100 | 9 | 2 | 0.156 | 0.125 | 0.165 | 0.270 | 0.253 | 0.256 |
| flags | 194 | 28 | 8 | 0.785 | 0.737 | 0.732 | 0.754 | 0.705 | 0.715 |
| glass | 214 | 9 | 6 | 0.651 | 0.493 | 0.478 | 0.522 | 0.456 | 0.447 |
| haberman-survival | 306 | 3 | 2 | 0.455 | 0.315 | 0.321 | 0.468 | 0.465 | 0.457 |
| hayes-roth | 160 | 3 | 3 | 0.638 | 0.518 | 0.520 | 0.686 | 0.535 | 0.555 |
| heart-cleveland | 303 | 13 | 5 | 0.764 | 0.730 | 0.750 | 0.700 | 0.674 | 0.683 |
| heart-hungarian | 294 | 12 | 2 | 0.463 | 0.431 | 0.437 | 0.387 | 0.356 | 0.380 |
| heart-switzerland | 123 | 12 | 5 | 0.741 | 0.680 | 0.633 | 0.628 | 0.597 | 0.561 |
| heart-va | 200 | 12 | 5 | 0.827 | 0.803 | 0.822 | 0.744 | 0.703 | 0.720 |
| hepatitis | 155 | 19 | 2 | 0.575 | 0.523 | 0.536 | 0.583 | 0.539 | 0.508 |
| hill-valley | 1212 | 100 | 2 | 0.508 | 0.491 | 0.529 | 0.566 | 0.561 | 0.543 |
| horse-colic | 368 | 25 | 2 | 0.365 | 0.314 | 0.339 | 0.354 | 0.347 | 0.372 |
| ilpd-indian-liver | 583 | 9 | 2 | 0.487 | 0.382 | 0.415 | 0.465 | 0.454 | 0.457 |
| image-segmentation | 2310 | 18 | 7 | 0.527 | 0.352 | 0.352 | 0.517 | 0.432 | 0.382 |
| ionosphere | 351 | 33 | 2 | 0.314 | 0.283 | 0.387 | 0.380 | 0.287 | 0.314 |
| iris | 150 | 4 | 3 | 0.772 | 0.624 | 0.381 | 0.686 | 0.631 | 0.619 |
| led-display | 1000 | 7 | 10 | 0.571 | 0.559 | 0.513 | 0.568 | 0.565 | 0.513 |
| lenses | 24 | 4 | 3 | 0.757 | 0.553 | 0.559 | 0.974 | 0.833 | 0.807 |
| letter | 20000 | 16 | 26 | 0.458 | 0.452 | 0.397 | 0.705 | 0.626 | 0.562 |
| libras | 360 | 90 | 15 | 0.689 | 0.572 | 0.492 | 0.534 | 0.403 | 0.347 |
| low-res-spect | 531 | 100 | 9 | 0.610 | 0.523 | 0.588 | 0.557 | 0.517 | 0.543 |
| lung-cancer | 32 | 56 | 3 | 0.730 | 0.686 | 0.663 | 0.840 | 0.768 | 0.725 |
| lymphography | 148 | 18 | 4 | 0.411 | 0.381 | 0.363 | 0.460 | 0.442 | 0.423 |
| magic | 19020 | 10 | 2 | 0.356 | 0.353 | 0.354 | 0.357 | 0.348 | 0.351 |
| mammographic | 961 | 5 | 2 | 0.432 | 0.420 | 0.409 | 0.419 | 0.415 | 0.417 |
| messidor | 1151 | 19 | 2 | 0.466 | 0.447 | 0.436 | 0.405 | 0.391 | 0.389 |
| miniboone | 130064 | 50 | 2 | 0.372 | 0.360 | 0.360 | 0.374 | 0.364 | 0.366 |
| molec-biol-promoter | 106 | 57 | 2 | 0.608 | 0.518 | 0.526 | 0.506 | 0.435 | 0.450 |
| molec-biol-splice | 3190 | 60 | 3 | 0.435 | 0.420 | 0.392 | 0.465 | 0.426 | 0.439 |
| monks-1 | 556 | 6 | 2 | 0.827 | 0.354 | 0.377 | 0.788 | 0.474 | 0.477 |
| monks-2 | 601 | 6 | 2 | 0.895 | 0.379 | 0.419 | 0.515 | 0.401 | 0.413 |
| monks-3 | 554 | 6 | 2 | 0.596 | 0.446 | 0.446 | 0.627 | 0.468 | 0.462 |
| mushroom | 8124 | 21 | 2 | NA | NA | NA | NA | NA | NA |
| musk-1 | 476 | 166 | 2 | 0.640 | 0.460 | 0.534 | 0.428 | 0.389 | 0.340 |
| musk-2 | 6598 | 166 | 2 | 0.533 | 0.378 | 0.438 | 0.435 | 0.328 | 0.381 |
| nursery | 12960 | 8 | 5 | 0.340 | 0.166 | 0.162 | 0.578 | 0.318 | 0.275 |
| oocytes_merluccius_nucleus_4d | 1022 | 41 | 2 | 0.810 | 0.478 | 0.461 | 0.394 | 0.373 | 0.374 |
| oocytes_merluccius_states_2f | 1022 | 25 | 3 | 0.445 | 0.387 | 0.339 | 0.426 | 0.402 | 0.446 |
| oocytes_trisopterus_nucleus_2f | 912 | 25 | 2 | 0.564 | 0.432 | 0.426 | 0.371 | 0.356 | 0.373 |
| oocytes_trisopterus_states_5b | 912 | 32 | 3 | 0.466 | 0.352 | 0.400 | 0.378 | 0.348 | 0.346 |
| optical | 5620 | 62 | 10 | 0.315 | 0.291 | 0.289 | 0.402 | 0.390 | 0.344 |
| ozone | 2536 | 72 | 2 | 0.303 | 0.221 | 0.265 | 0.330 | 0.336 | 0.311 |
| page-blocks | 5473 | 10 | 5 | 0.436 | 0.352 | 0.308 | 0.394 | 0.360 | 0.342 |
| parkinsons | 195 | 22 | 2 | 0.769 | 0.521 | 0.482 | 0.465 | 0.371 | 0.454 |
| pendigits | 10992 | 16 | 10 | 0.309 | 0.287 | 0.292 | 0.382 | 0.355 | 0.261 |
| phishing | 1353 | 9 | 3 | 0.596 | 0.437 | 0.440 | 0.551 | 0.457 | 0.434 |
| pima | 768 | 8 | 2 | 0.477 | 0.459 | 0.451 | 0.457 | 0.454 | 0.452 |
| pittsburg-bridges-MATERIAL | 106 | 7 | 3 | 0.415 | 0.318 | 0.379 | 0.496 | 0.450 | 0.435 |
| pittsburg-bridges-REL-L | 103 | 7 | 3 | 0.618 | 0.535 | 0.486 | 0.558 | 0.501 | 0.492 |
| pittsburg-bridges-SPAN | 92 | 7 | 3 | 0.550 | 0.534 | 0.582 | 0.499 | 0.449 | 0.463 |
| pittsburg-bridges-T-OR-D | 102 | 7 | 2 | 0.530 | 0.399 | 0.466 | 0.387 | 0.355 | 0.397 |
| pittsburg-bridges-TYPE | 105 | 7 | 6 | 0.825 | 0.737 | 0.738 | 0.770 | 0.704 | 0.694 |
| planning | 182 | 12 | 2 | 0.298 | 0.284 | 0.318 | 0.297 | 0.296 | 0.291 |
| plant-margin | 1600 | 64 | 100 | 0.619 | 0.597 | 0.502 | 0.610 | 0.598 | 0.469 |
| plant-shape | 1600 | 64 | 100 | 0.813 | 0.787 | 0.740 | 0.682 | 0.643 | 0.566 |
| plant-texture | 1599 | 64 | 100 | 0.655 | 0.646 | 0.550 | 0.650 | 0.620 | 0.496 |
| post-operative | 90 | 8 | 3 | 0.301 | 0.277 | 0.253 | 0.307 | 0.300 | 0.323 |
| primary-tumor | 330 | 17 | 15 | 0.878 | 0.850 | 0.841 | 0.804 | 0.797 | 0.794 |
| ringnorm | 7400 | 20 | 2 | 0.370 | 0.331 | 0.290 | 0.405 | 0.341 | 0.367 |
| seeds | 210 | 7 | 3 | 0.640 | 0.644 | 0.711 | 0.687 | 0.610 | 0.610 |
| semeion | 1593 | 256 | 10 | 0.543 | 0.527 | 0.465 | 0.523 | 0.496 | 0.483 |

*Continued on next page.*

| Dataset | N | M | K | RED+BNN | BNN MCP | BNN Entropy | RED+MC-D | MC-D MCP | MC-D Entropy |
|---|---|---|---|---|---|---|---|---|---|
| soybean | 683 | 35 | 18 | 0.524 | 0.487 | 0.354 | 0.498 | 0.482 | 0.368 |
| spambase | 4601 | 57 | 2 | 0.324 | 0.311 | 0.323 | 0.351 | 0.347 | 0.369 |
| spect | 265 | 22 | 2 | 0.452 | 0.404 | 0.394 | 0.498 | 0.492 | 0.497 |
| spectf | 267 | 44 | 2 | 0.775 | 0.560 | 0.545 | 0.396 | 0.322 | 0.314 |
| statlog-australian-credit | 690 | 14 | 2 | 0.460 | 0.350 | 0.335 | 0.435 | 0.403 | 0.409 |
| statlog-german-credit | 1000 | 24 | 2 | 0.595 | 0.540 | 0.547 | 0.407 | 0.400 | 0.412 |
| statlog-heart | 270 | 13 | 2 | 0.440 | 0.409 | 0.367 | 0.366 | 0.338 | 0.359 |
| statlog-image | 2310 | 18 | 7 | 0.532 | 0.404 | 0.330 | 0.491 | 0.439 | 0.373 |
| statlog-landsat | 6435 | 36 | 6 | 0.488 | 0.453 | 0.444 | 0.441 | 0.411 | 0.375 |
| statlog-shuttle | 58000 | 9 | 7 | 0.526 | 0.261 | 0.427 | 0.657 | 0.431 | 0.465 |
| statlog-vehicle | 846 | 18 | 4 | 0.536 | 0.522 | 0.426 | 0.513 | 0.478 | 0.437 |
| steel-plates | 1941 | 27 | 7 | 0.601 | 0.533 | 0.475 | 0.541 | 0.510 | 0.485 |
| synthetic-control | 600 | 60 | 6 | 0.391 | 0.233 | 0.684 | 0.510 | 0.340 | 0.386 |
| teaching | 151 | 5 | 3 | 0.710 | 0.672 | 0.673 | 0.680 | 0.632 | 0.605 |
| thyroid | 7200 | 21 | 3 | 0.552 | 0.439 | 0.473 | 0.536 | 0.476 | 0.461 |
| tic-tac-toe | 958 | 9 | 2 | 0.851 | 0.187 | 0.344 | 0.642 | 0.379 | 0.505 |
| titanic | 2201 | 3 | 2 | 0.361 | 0.340 | 0.332 | 0.338 | 0.338 | 0.340 |
| trains | 10 | 29 | 2 | 1.000 | 0.750 | 0.875 | 0.969 | 0.719 | 0.719 |
| twonorm | 7400 | 20 | 2 | 0.364 | 0.338 | 0.348 | 0.367 | 0.361 | 0.361 |
| vertebral-column-2clases | 310 | 6 | 2 | 0.523 | 0.465 | 0.475 | 0.440 | 0.422 | 0.434 |
| vertebral-column-3clases | 310 | 6 | 3 | 0.659 | 0.613 | 0.570 | 0.436 | 0.431 | 0.437 |
| wall-following | 5456 | 24 | 4 | 0.422 | 0.373 | 0.365 | 0.432 | 0.388 | 0.406 |
| waveform | 5000 | 21 | 3 | 0.414 | 0.391 | 0.363 | 0.428 | 0.419 | 0.387 |
| waveform-noise | 5000 | 40 | 3 | 0.437 | 0.425 | 0.393 | 0.436 | 0.425 | 0.401 |
| wine | 178 | 13 | 3 | 0.635 | 0.626 | 0.623 | 0.822 | 0.810 | 0.648 |
| wine-quality-red | 1599 | 11 | 6 | 0.574 | 0.527 | 0.516 | 0.517 | 0.497 | 0.487 |
| wine-quality-white | 4898 | 11 | 7 | 0.507 | 0.495 | 0.473 | 0.508 | 0.502 | 0.475 |
| yeast | 1484 | 8 | 10 | 0.602 | 0.594 | 0.573 | 0.592 | 0.586 | 0.544 |
| zoo | 101 | 16 | 7 | 0.888 | 0.771 | 0.764 | 0.904 | 0.733 | 0.819 |

*Continued from previous page.*

Table 13: Comparison between RED and Counterparts Using AUPR-Success

| Dataset | N | M | K | RED+BNN | BNN MCP | BNN Entropy | RED+MC-D | MC-D MCP | MC-D Entropy |
|---|---|---|---|---|---|---|---|---|---|
| abalone | 4177 | 8 | 3 | 0.855 | 0.853 | 0.849 | 0.853 | 0.852 | 0.851 |
| acute-inflammation | 120 | 6 | 2 | NA | NA | NA | NA | NA | NA |
| acute-nephritis | 120 | 6 | 2 | NA | NA | NA | NA | NA | NA |
| adult | 48842 | 14 | 2 | 0.970 | 0.969 | 0.970 | 0.970 | 0.969 | 0.970 |
| annealing | 898 | 31 | 5 | 0.983 | 0.951 | 0.947 | 0.981 | 0.972 | 0.974 |
| arrhythmia | 452 | 262 | 13 | 0.801 | 0.750 | 0.806 | 0.827 | 0.809 | 0.823 |
| audiology-std | 196 | 59 | 18 | 0.918 | 0.912 | 0.938 | 0.942 | 0.902 | 0.925 |
| balance-scale | 625 | 4 | 3 | 0.999 | 0.996 | 0.997 | 0.999 | 0.998 | 0.999 |
| balloons | 16 | 4 | 2 | 0.815 | 0.780 | 0.780 | 0.829 | 0.819 | 0.819 |
| bank | 4521 | 16 | 2 | 0.985 | 0.980 | 0.981 | 0.981 | 0.982 | 0.983 |
| bioconcentration | 779 | 9 | 3 | 0.764 | 0.749 | 0.741 | 0.753 | 0.744 | 0.734 |
| blood | 748 | 4 | 2 | 0.902 | 0.884 | 0.905 | 0.901 | 0.902 | 0.904 |
| breast-cancer | 286 | 9 | 2 | 0.829 | 0.770 | 0.823 | 0.823 | 0.821 | 0.834 |
| breast-cancer-wisc | 699 | 9 | 2 | 0.998 | 0.995 | 0.995 | 0.998 | 0.998 | 0.997 |
| breast-cancer-wisc-diag | 569 | 30 | 2 | 0.997 | 0.995 | 0.997 | 0.998 | 0.996 | 0.998 |
| breast-cancer-wisc-prog | 198 | 33 | 2 | 0.875 | 0.855 | 0.833 | 0.858 | 0.859 | 0.860 |
| breast-tissue | 106 | 9 | 6 | 0.929 | 0.904 | 0.906 | 0.931 | 0.907 | 0.920 |
| car | 1728 | 6 | 4 | 0.997 | 0.998 | 0.998 | 0.999 | 1.000 | 0.999 |
| cardiotocography-10clases | 2126 | 21 | 10 | 0.960 | 0.954 | 0.952 | 0.962 | 0.956 | 0.955 |
| cardiotocography-3clases | 2126 | 21 | 3 | 0.990 | 0.988 | 0.989 | 0.992 | 0.991 | 0.992 |
| chess-krvk | 28056 | 6 | 18 | 0.923 | 0.919 | 0.912 | 0.806 | 0.753 | 0.738 |
| chess-krvkp | 3196 | 36 | 2 | 0.999 | 0.999 | 1.000 | 1.000 | 0.999 | 1.000 |
| climate | 540 | 20 | 2 | 0.988 | 0.991 | 0.993 | 0.987 | 0.989 | 0.990 |
| congressional-voting | 435 | 16 | 2 | 0.745 | 0.700 | 0.719 | 0.713 | 0.713 | 0.717 |
| conn-bench-sonar-mines-rocks | 208 | 60 | 2 | 0.939 | 0.948 | 0.952 | 0.954 | 0.951 | 0.956 |
| conn-bench-vowel-deterding | 990 | 11 | 11 | 0.998 | 0.994 | 0.990 | 0.998 | 0.996 | 0.996 |
| connect-4 | 67557 | 42 | 2 | 0.975 | 0.975 | 0.975 | 0.959 | 0.954 | 0.954 |
| contrac | 1473 | 9 | 3 | 0.743 | 0.739 | 0.737 | 0.740 | 0.739 | 0.729 |
| credit-approval | 690 | 15 | 2 | 0.953 | 0.936 | 0.935 | 0.944 | 0.942 | 0.943 |
| cylinder-bands | 512 | 35 | 2 | 0.846 | 0.820 | 0.830 | 0.869 | 0.865 | 0.872 |
| dermatology | 366 | 34 | 6 | 0.998 | 0.996 | 0.999 | 1.000 | 1.000 | 0.999 |
| echocardiogram | 131 | 10 | 2 | 0.945 | 0.934 | 0.918 | 0.949 | 0.944 | 0.949 |
| ecoli | 336 | 7 | 8 | 0.958 | 0.954 | 0.963 | 0.956 | 0.955 | 0.959 |
| energy-y1 | 768 | 8 | 3 | 0.997 | 0.978 | 0.979 | 0.996 | 0.993 | 0.993 |
| energy-y2 | 768 | 8 | 3 | 0.993 | 0.975 | 0.973 | 0.990 | 0.986 | 0.987 |
| fertility | 100 | 9 | 2 | 0.945 | 0.917 | 0.921 | 0.960 | 0.928 | 0.933 |
| flags | 194 | 28 | 8 | 0.734 | 0.679 | 0.702 | 0.747 | 0.697 | 0.718 |
| glass | 214 | 9 | 6 | 0.832 | 0.729 | 0.741 | 0.833 | 0.786 | 0.824 |
| haberman-survival | 306 | 3 | 2 | 0.869 | 0.769 | 0.795 | 0.845 | 0.840 | 0.841 |
| hayes-roth | 160 | 3 | 3 | 0.973 | 0.969 | 0.962 | 0.954 | 0.901 | 0.903 |

*Continued on next page.*

| Dataset | N | M | K | RED+BNN | BNN MCP | BNN Entropy | RED+MC-D | MC-D MCP | MC-D Entropy |
|---|---|---|---|---|---|---|---|---|---|
| heart-cleveland | 303 | 13 | 5 | 0.891 | 0.879 | 0.884 | 0.865 | 0.863 | 0.866 |
| heart-hungarian | 294 | 12 | 2 | 0.933 | 0.913 | 0.927 | 0.924 | 0.904 | 0.908 |
| heart-switzerland | 123 | 12 | 5 | 0.562 | 0.505 | 0.489 | 0.470 | 0.445 | 0.437 |
| heart-va | 200 | 12 | 5 | 0.379 | 0.312 | 0.323 | 0.443 | 0.410 | 0.407 |
| hepatitis | 155 | 19 | 2 | 0.968 | 0.958 | 0.972 | 0.957 | 0.966 | 0.967 |
| hill-valley | 1212 | 100 | 2 | 0.517 | 0.504 | 0.519 | 0.546 | 0.542 | 0.538 |
| horse-colic | 368 | 25 | 2 | 0.926 | 0.894 | 0.909 | 0.904 | 0.895 | 0.901 |
| ilpd-indian-liver | 583 | 9 | 2 | 0.872 | 0.845 | 0.852 | 0.879 | 0.875 | 0.875 |
| image-segmentation | 2310 | 18 | 7 | 0.997 | 0.995 | 0.997 | 0.998 | 0.998 | 0.998 |
| ionosphere | 351 | 33 | 2 | 0.984 | 0.977 | 0.982 | 0.991 | 0.991 | 0.991 |
| iris | 150 | 4 | 3 | 0.997 | 0.994 | 0.993 | 0.997 | 0.996 | 0.997 |
| led-display | 1000 | 7 | 10 | 0.886 | 0.881 | 0.870 | 0.887 | 0.884 | 0.878 |
| lenses | 24 | 4 | 3 | 0.860 | 0.783 | 0.798 | 0.988 | 0.934 | 0.922 |
| letter | 20000 | 16 | 26 | 0.996 | 0.997 | 0.997 | 0.988 | 0.985 | 0.980 |
| libras | 360 | 90 | 15 | 0.878 | 0.845 | 0.728 | 0.952 | 0.938 | 0.924 |
| low-res-spect | 531 | 100 | 9 | 0.973 | 0.973 | 0.979 | 0.978 | 0.978 | 0.984 |
| lung-cancer | 32 | 56 | 3 | 0.645 | 0.613 | 0.628 | 0.666 | 0.582 | 0.519 |
| lymphography | 148 | 18 | 4 | 0.948 | 0.941 | 0.942 | 0.929 | 0.934 | 0.932 |
| magic | 19020 | 10 | 2 | 0.969 | 0.968 | 0.968 | 0.968 | 0.967 | 0.967 |
| mammographic | 961 | 5 | 2 | 0.924 | 0.908 | 0.915 | 0.921 | 0.918 | 0.919 |
| messidor | 1151 | 19 | 2 | 0.811 | 0.808 | 0.809 | 0.871 | 0.871 | 0.870 |
| miniboone | 130064 | 50 | 2 | 0.993 | 0.992 | 0.993 | 0.992 | 0.992 | 0.993 |
| molec-biol-promoter | 106 | 57 | 2 | 0.932 | 0.896 | 0.904 | 0.925 | 0.897 | 0.897 |
| molec-biol-splice | 3190 | 60 | 3 | 0.977 | 0.976 | 0.975 | 0.970 | 0.970 | 0.972 |
| monks-1 | 556 | 6 | 2 | 0.999 | 0.996 | 0.997 | 1.000 | 0.999 | 0.999 |
| monks-2 | 601 | 6 | 2 | 0.964 | 0.808 | 0.816 | 0.870 | 0.821 | 0.820 |
| monks-3 | 554 | 6 | 2 | 0.997 | 0.996 | 0.996 | 0.999 | 0.997 | 0.997 |
| mushroom | 8124 | 21 | 2 | NA | NA | NA | NA | NA | NA |
| musk-1 | 476 | 166 | 2 | 0.937 | 0.914 | 0.918 | 0.980 | 0.979 | 0.980 |
| musk-2 | 6598 | 166 | 2 | 0.999 | 0.998 | 0.998 | 1.000 | 0.999 | 1.000 |
| nursery | 12960 | 8 | 5 | 1.000 | 1.000 | 1.000 | 1.000 | 1.000 | 1.000 |
| oocytes_merluccius_nucleus_4d | 1022 | 41 | 2 | 0.955 | 0.779 | 0.780 | 0.942 | 0.936 | 0.939 |
| oocytes_merluccius_states_2f | 1022 | 25 | 3 | 0.987 | 0.984 | 0.985 | 0.991 | 0.989 | 0.991 |
| oocytes_trisopterus_nucleus_2f | 912 | 25 | 2 | 0.914 | 0.849 | 0.866 | 0.936 | 0.925 | 0.933 |
| oocytes_trisopterus_states_5b | 912 | 32 | 3 | 0.984 | 0.977 | 0.980 | 0.993 | 0.992 | 0.993 |
| optical | 5620 | 62 | 10 | 0.997 | 0.998 | 0.999 | 0.999 | 0.998 | 0.999 |
| ozone | 2536 | 72 | 2 | 0.996 | 0.995 | 0.995 | 0.995 | 0.995 | 0.996 |
| page-blocks | 5473 | 10 | 5 | 0.998 | 0.997 | 0.998 | 0.998 | 0.998 | 0.998 |
| parkinsons | 195 | 22 | 2 | 0.988 | 0.976 | 0.977 | 0.985 | 0.987 | 0.987 |
| pendigits | 10992 | 16 | 10 | 0.999 | 0.999 | 0.999 | 0.999 | 0.999 | 0.999 |
| phishing | 1353 | 9 | 3 | 0.974 | 0.967 | 0.967 | 0.979 | 0.975 | 0.974 |
| pima | 768 | 8 | 2 | 0.896 | 0.889 | 0.906 | 0.900 | 0.898 | 0.903 |
| pittsburg-bridges-MATERIAL | 106 | 7 | 3 | 0.977 | 0.964 | 0.969 | 0.975 | 0.973 | 0.972 |
| pittsburg-bridges-REL-L | 103 | 7 | 3 | 0.798 | 0.756 | 0.709 | 0.784 | 0.771 | 0.775 |
| pittsburg-bridges-SPAN | 92 | 7 | 3 | 0.845 | 0.835 | 0.822 | 0.843 | 0.794 | 0.799 |
| pittsburg-bridges-T-OR-D | 102 | 7 | 2 | 0.954 | 0.913 | 0.933 | 0.964 | 0.941 | 0.940 |
| pittsburg-bridges-TYPE | 105 | 7 | 6 | 0.785 | 0.691 | 0.744 | 0.765 | 0.697 | 0.683 |
| planning | 182 | 12 | 2 | 0.766 | 0.750 | 0.765 | 0.699 | 0.692 | 0.682 |
| plant-margin | 1600 | 64 | 100 | 0.935 | 0.933 | 0.908 | 0.954 | 0.952 | 0.919 |
| plant-shape | 1600 | 64 | 100 | 0.713 | 0.683 | 0.620 | 0.874 | 0.856 | 0.804 |
| plant-texture | 1599 | 64 | 100 | 0.947 | 0.945 | 0.921 | 0.947 | 0.940 | 0.902 |
| post-operative | 90 | 8 | 3 | 0.659 | 0.633 | 0.635 | 0.667 | 0.663 | 0.662 |
| primary-tumor | 330 | 17 | 15 | 0.777 | 0.740 | 0.747 | 0.753 | 0.744 | 0.747 |
| ringnorm | 7400 | 20 | 2 | 0.999 | 0.999 | 0.999 | 0.998 | 0.997 | 0.998 |
| seeds | 210 | 7 | 3 | 0.980 | 0.984 | 0.990 | 0.990 | 0.985 | 0.985 |
| semeion | 1593 | 256 | 10 | 0.992 | 0.992 | 0.992 | 0.992 | 0.993 | 0.993 |
| soybean | 683 | 35 | 18 | 0.996 | 0.994 | 0.993 | 0.996 | 0.996 | 0.992 |
| spambase | 4601 | 57 | 2 | 0.987 | 0.986 | 0.990 | 0.989 | 0.989 | 0.992 |
| spect | 265 | 22 | 2 | 0.769 | 0.743 | 0.769 | 0.785 | 0.772 | 0.773 |
| spectf | 267 | 44 | 2 | 0.965 | 0.956 | 0.954 | 0.958 | 0.951 | 0.949 |
| statlog-australian-credit | 690 | 14 | 2 | 0.739 | 0.663 | 0.659 | 0.716 | 0.700 | 0.700 |
| statlog-german-credit | 1000 | 24 | 2 | 0.904 | 0.899 | 0.899 | 0.891 | 0.897 | 0.898 |
| statlog-heart | 270 | 13 | 2 | 0.936 | 0.928 | 0.929 | 0.931 | 0.924 | 0.930 |
| statlog-image | 2310 | 18 | 7 | 0.998 | 0.997 | 0.997 | 0.998 | 0.998 | 0.998 |
| statlog-landsat | 6435 | 36 | 6 | 0.984 | 0.982 | 0.981 | 0.985 | 0.986 | 0.985 |
| statlog-shuttle | 58000 | 9 | 7 | 1.000 | 0.999 | 1.000 | 1.000 | 1.000 | 1.000 |
| statlog-vehicle | 846 | 18 | 4 | 0.944 | 0.940 | 0.914 | 0.970 | 0.967 | 0.964 |
| steel-plates | 1941 | 27 | 7 | 0.923 | 0.912 | 0.900 | 0.923 | 0.913 | 0.916 |
| synthetic-control | 600 | 60 | 6 | 0.998 | 0.997 | 1.000 | 1.000 | 0.999 | 0.999 |
| teaching | 151 | 5 | 3 | 0.589 | 0.555 | 0.517 | 0.694 | 0.627 | 0.618 |
| thyroid | 7200 | 21 | 3 | 1.000 | 0.999 | 1.000 | 0.999 | 0.999 | 1.000 |
| tic-tac-toe | 958 | 9 | 2 | 1.000 | 0.998 | 0.999 | 1.000 | 0.999 | 0.999 |
| titanic | 2201 | 3 | 2 | 0.890 | 0.884 | 0.885 | 0.890 | 0.889 | 0.891 |
| trains | 10 | 29 | 2 | 1.000 | 0.750 | 0.875 | 0.969 | 0.719 | 0.719 |
| twonorm | 7400 | 20 | 2 | 0.999 | 0.999 | 0.999 | 0.999 | 0.998 | 0.999 |
| vertebral-column-2clases | 310 | 6 | 2 | 0.975 | 0.964 | 0.967 | 0.976 | 0.975 | 0.977 |
| vertebral-column-3clases | 310 | 6 | 3 | 0.971 | 0.967 | 0.965 | 0.976 | 0.975 | 0.975 |

*Continued on next page.*

| Dataset | N | M | K | RED+BNN | BNN MCP | BNN Entropy | RED+MC-D | MC-D MCP | MC-D Entropy |
|---|---|---|---|---|---|---|---|---|---|
| wall-following | 5456 | 24 | 4 | 0.990 | 0.987 | 0.990 | 0.992 | 0.992 | 0.994 |
| waveform | 5000 | 21 | 3 | 0.973 | 0.970 | 0.970 | 0.973 | 0.973 | 0.972 |
| waveform-noise | 5000 | 40 | 3 | 0.975 | 0.974 | 0.973 | 0.971 | 0.971 | 0.971 |
| wine | 178 | 13 | 3 | 0.997 | 0.994 | 0.999 | 0.999 | 0.998 | 0.999 |
| wine-quality-red | 1599 | 11 | 6 | 0.725 | 0.697 | 0.711 | 0.736 | 0.728 | 0.725 |
| wine-quality-white | 4898 | 11 | 7 | 0.639 | 0.624 | 0.607 | 0.664 | 0.653 | 0.634 |
| yeast | 1484 | 8 | 10 | 0.721 | 0.716 | 0.715 | 0.747 | 0.741 | 0.718 |
| zoo | 101 | 16 | 7 | 0.997 | 0.996 | 0.997 | 0.998 | 0.996 | 0.995 |

*Continued from previous page.*

Table 14: Comparison between RED and Counterparts Using AUROC

| Dataset | N | M | K | RED+BNN | BNN MCP | BNN Entropy | RED+MC-D | MC-D MCP | MC-D Entropy |
|---|---|---|---|---|---|---|---|---|---|
| abalone | 4177 | 8 | 3 | 0.731 | 0.727 | 0.718 | 0.725 | 0.723 | 0.716 |
| acute-inflammation | 120 | 6 | 2 | NA | NA | NA | NA | NA | NA |
| acute-nephritis | 120 | 6 | 2 | NA | NA | NA | NA | NA | NA |
| adult | 48842 | 14 | 2 | 0.845 | 0.842 | 0.843 | 0.846 | 0.841 | 0.841 |
| annealing | 898 | 31 | 5 | 0.928 | 0.798 | 0.764 | 0.883 | 0.850 | 0.836 |
| arrhythmia | 452 | 262 | 13 | 0.766 | 0.703 | 0.784 | 0.709 | 0.698 | 0.700 |
| audiology-std | 196 | 59 | 18 | 0.850 | 0.814 | 0.865 | 0.874 | 0.814 | 0.851 |
| balance-scale | 625 | 4 | 3 | 0.982 | 0.946 | 0.965 | 0.977 | 0.955 | 0.968 |
| balloons | 16 | 4 | 2 | 0.762 | 0.714 | 0.714 | 0.778 | 0.722 | 0.722 |
| bank | 4521 | 16 | 2 | 0.900 | 0.867 | 0.876 | 0.868 | 0.872 | 0.877 |
| bioconcentration | 779 | 9 | 3 | 0.669 | 0.636 | 0.639 | 0.619 | 0.609 | 0.586 |
| blood | 748 | 4 | 2 | 0.757 | 0.723 | 0.753 | 0.731 | 0.731 | 0.735 |
| breast-cancer | 286 | 9 | 2 | 0.662 | 0.581 | 0.652 | 0.672 | 0.668 | 0.678 |
| breast-cancer-wisc | 699 | 9 | 2 | 0.944 | 0.876 | 0.911 | 0.943 | 0.938 | 0.936 |
| breast-cancer-wisc-diag | 569 | 30 | 2 | 0.898 | 0.769 | 0.896 | 0.941 | 0.893 | 0.929 |
| breast-cancer-wisc-prog | 198 | 33 | 2 | 0.720 | 0.687 | 0.671 | 0.696 | 0.692 | 0.689 |
| breast-tissue | 106 | 9 | 6 | 0.891 | 0.860 | 0.862 | 0.909 | 0.883 | 0.888 |
| car | 1728 | 6 | 4 | 0.964 | 0.956 | 0.942 | 0.977 | 0.977 | 0.965 |
| cardiotocography-10clases | 2126 | 21 | 10 | 0.848 | 0.835 | 0.810 | 0.845 | 0.831 | 0.818 |
| cardiotocography-3clases | 2126 | 21 | 3 | 0.912 | 0.896 | 0.901 | 0.913 | 0.905 | 0.910 |
| chess-krvk | 28056 | 6 | 18 | 0.792 | 0.781 | 0.762 | 0.777 | 0.716 | 0.696 |
| chess-krvkp | 3196 | 36 | 2 | 0.917 | 0.883 | 0.954 | 0.968 | 0.927 | 0.956 |
| climate | 540 | 20 | 2 | 0.906 | 0.877 | 0.916 | 0.881 | 0.868 | 0.897 |
| congressional-voting | 435 | 16 | 2 | 0.632 | 0.594 | 0.605 | 0.616 | 0.615 | 0.617 |
| conn-bench-sonar-mines-rocks | 208 | 60 | 2 | 0.780 | 0.794 | 0.825 | 0.827 | 0.824 | 0.836 |
| conn-bench-vowel-deterding | 990 | 11 | 11 | 0.965 | 0.916 | 0.863 | 0.969 | 0.936 | 0.928 |
| connect-4 | 67557 | 42 | 2 | 0.852 | 0.851 | 0.851 | 0.832 | 0.807 | 0.808 |
| contrac | 1473 | 9 | 3 | 0.691 | 0.690 | 0.683 | 0.679 | 0.677 | 0.665 |
| credit-approval | 690 | 15 | 2 | 0.773 | 0.723 | 0.735 | 0.753 | 0.750 | 0.753 |
| cylinder-bands | 512 | 35 | 2 | 0.705 | 0.643 | 0.653 | 0.702 | 0.700 | 0.705 |
| dermatology | 366 | 34 | 6 | 0.937 | 0.882 | 0.969 | 0.988 | 0.987 | 0.982 |
| echocardiogram | 131 | 10 | 2 | 0.737 | 0.703 | 0.719 | 0.798 | 0.776 | 0.786 |
| ecoli | 336 | 7 | 8 | 0.817 | 0.807 | 0.835 | 0.780 | 0.777 | 0.800 |
| energy-y1 | 768 | 8 | 3 | 0.980 | 0.849 | 0.849 | 0.933 | 0.888 | 0.888 |
| energy-y2 | 768 | 8 | 3 | 0.952 | 0.840 | 0.821 | 0.921 | 0.889 | 0.894 |
| fertility | 100 | 9 | 2 | 0.594 | 0.439 | 0.471 | 0.707 | 0.609 | 0.608 |
| flags | 194 | 28 | 8 | 0.756 | 0.721 | 0.730 | 0.752 | 0.699 | 0.715 |
| glass | 214 | 9 | 6 | 0.739 | 0.578 | 0.589 | 0.668 | 0.612 | 0.660 |
| haberman-survival | 306 | 3 | 2 | 0.712 | 0.544 | 0.559 | 0.700 | 0.699 | 0.703 |
| hayes-roth | 160 | 3 | 3 | 0.888 | 0.863 | 0.837 | 0.880 | 0.786 | 0.787 |
| heart-cleveland | 303 | 13 | 5 | 0.838 | 0.827 | 0.834 | 0.795 | 0.795 | 0.800 |
| heart-hungarian | 294 | 12 | 2 | 0.785 | 0.746 | 0.766 | 0.743 | 0.718 | 0.728 |
| heart-switzerland | 123 | 12 | 5 | 0.637 | 0.592 | 0.543 | 0.521 | 0.487 | 0.444 |
| heart-va | 200 | 12 | 5 | 0.627 | 0.568 | 0.590 | 0.583 | 0.545 | 0.561 |
| hepatitis | 155 | 19 | 2 | 0.853 | 0.812 | 0.859 | 0.843 | 0.849 | 0.850 |
| hill-valley | 1212 | 100 | 2 | 0.511 | 0.495 | 0.528 | 0.560 | 0.559 | 0.543 |
| horse-colic | 368 | 25 | 2 | 0.733 | 0.664 | 0.688 | 0.702 | 0.693 | 0.708 |
| ilpd-indian-liver | 583 | 9 | 2 | 0.723 | 0.655 | 0.671 | 0.723 | 0.716 | 0.717 |
| image-segmentation | 2310 | 18 | 7 | 0.934 | 0.898 | 0.938 | 0.946 | 0.936 | 0.952 |
| ionosphere | 351 | 33 | 2 | 0.853 | 0.777 | 0.851 | 0.882 | 0.874 | 0.878 |
| iris | 150 | 4 | 3 | 0.975 | 0.931 | 0.908 | 0.961 | 0.953 | 0.961 |
| led-display | 1000 | 7 | 10 | 0.760 | 0.753 | 0.728 | 0.760 | 0.756 | 0.738 |
| lenses | 24 | 4 | 3 | 0.798 | 0.667 | 0.702 | 0.979 | 0.885 | 0.865 |
| letter | 20000 | 16 | 26 | 0.940 | 0.950 | 0.943 | 0.935 | 0.920 | 0.895 |
| libras | 360 | 90 | 15 | 0.804 | 0.739 | 0.594 | 0.818 | 0.767 | 0.720 |
| low-res-spect | 531 | 100 | 9 | 0.873 | 0.859 | 0.885 | 0.862 | 0.853 | 0.885 |
| lung-cancer | 32 | 56 | 3 | 0.669 | 0.601 | 0.596 | 0.722 | 0.613 | 0.564 |
| lymphography | 148 | 18 | 4 | 0.770 | 0.741 | 0.742 | 0.767 | 0.758 | 0.748 |
| magic | 19020 | 10 | 2 | 0.816 | 0.814 | 0.816 | 0.813 | 0.811 | 0.811 |
| mammographic | 961 | 5 | 2 | 0.753 | 0.739 | 0.740 | 0.757 | 0.754 | 0.755 |
| messidor | 1151 | 19 | 2 | 0.661 | 0.655 | 0.648 | 0.691 | 0.689 | 0.686 |

*Continued on next page.*

| Dataset | N | M | K | RED+BNN | BNN MCP | BNN Entropy | RED+MC-D | MC-D MCP | MC-D Entropy |
|---|---|---|---|---|---|---|---|---|---|
| miniboone | 130064 | 50 | 2 | 0.906 | 0.903 | 0.904 | 0.905 | 0.904 | 0.905 |
| molec-biol-promoter | 106 | 57 | 2 | 0.803 | 0.676 | 0.745 | 0.779 | 0.720 | 0.713 |
| molec-biol-splice | 3190 | 60 | 3 | 0.858 | 0.853 | 0.845 | 0.851 | 0.839 | 0.849 |
| monks-1 | 556 | 6 | 2 | 0.989 | 0.877 | 0.918 | 0.991 | 0.963 | 0.975 |
| monks-2 | 601 | 6 | 2 | 0.934 | 0.581 | 0.611 | 0.742 | 0.639 | 0.641 |
| monks-3 | 554 | 6 | 2 | 0.959 | 0.930 | 0.934 | 0.970 | 0.950 | 0.950 |
| mushroom | 8124 | 21 | 2 | NA | NA | NA | NA | NA | NA |
| musk-1 | 476 | 166 | 2 | 0.838 | 0.769 | 0.792 | 0.835 | 0.821 | 0.828 |
| musk-2 | 6598 | 166 | 2 | 0.968 | 0.957 | 0.961 | 0.971 | 0.937 | 0.971 |
| nursery | 12960 | 8 | 5 | 0.957 | 0.965 | 0.974 | 0.994 | 0.968 | 0.967 |
| oocytes_merluccius_nucleus_4d | 1022 | 41 | 2 | 0.910 | 0.649 | 0.641 | 0.771 | 0.761 | 0.764 |
| oocytes_merluccius_states_2f | 1022 | 25 | 3 | 0.891 | 0.873 | 0.872 | 0.902 | 0.892 | 0.905 |
| oocytes_trisopterus_nucleus_2f | 912 | 25 | 2 | 0.793 | 0.677 | 0.697 | 0.752 | 0.735 | 0.750 |
| oocytes_trisopterus_states_5b | 912 | 32 | 3 | 0.875 | 0.832 | 0.854 | 0.903 | 0.898 | 0.914 |
| optical | 5620 | 62 | 10 | 0.897 | 0.907 | 0.963 | 0.949 | 0.941 | 0.960 |
| ozone | 2536 | 72 | 2 | 0.899 | 0.866 | 0.882 | 0.889 | 0.872 | 0.909 |
| page-blocks | 5473 | 10 | 5 | 0.945 | 0.932 | 0.947 | 0.948 | 0.942 | 0.950 |
| parkinsons | 195 | 22 | 2 | 0.932 | 0.851 | 0.865 | 0.890 | 0.869 | 0.873 |
| pendigits | 10992 | 16 | 10 | 0.936 | 0.934 | 0.947 | 0.946 | 0.937 | 0.941 |
| phishing | 1353 | 9 | 3 | 0.884 | 0.841 | 0.840 | 0.876 | 0.863 | 0.855 |
| pima | 768 | 8 | 2 | 0.755 | 0.750 | 0.752 | 0.751 | 0.750 | 0.750 |
| pittsburg-bridges-MATERIAL | 106 | 7 | 3 | 0.839 | 0.775 | 0.797 | 0.833 | 0.816 | 0.813 |
| pittsburg-bridges-REL-L | 103 | 7 | 3 | 0.682 | 0.625 | 0.576 | 0.656 | 0.624 | 0.629 |
| pittsburg-bridges-SPAN | 92 | 7 | 3 | 0.716 | 0.698 | 0.702 | 0.675 | 0.618 | 0.643 |
| pittsburg-bridges-T-OR-D | 102 | 7 | 2 | 0.793 | 0.715 | 0.742 | 0.780 | 0.730 | 0.740 |
| pittsburg-bridges-TYPE | 105 | 7 | 6 | 0.791 | 0.728 | 0.750 | 0.771 | 0.715 | 0.712 |
| planning | 182 | 12 | 2 | 0.518 | 0.496 | 0.519 | 0.452 | 0.446 | 0.441 |
| plant-margin | 1600 | 64 | 100 | 0.835 | 0.829 | 0.767 | 0.855 | 0.850 | 0.768 |
| plant-shape | 1600 | 64 | 100 | 0.759 | 0.730 | 0.677 | 0.801 | 0.773 | 0.706 |
| plant-texture | 1599 | 64 | 100 | 0.856 | 0.853 | 0.796 | 0.853 | 0.840 | 0.760 |
| post-operative | 90 | 8 | 3 | 0.380 | 0.364 | 0.357 | 0.428 | 0.415 | 0.414 |
| primary-tumor | 330 | 17 | 15 | 0.831 | 0.811 | 0.803 | 0.781 | 0.777 | 0.777 |
| ringnorm | 7400 | 20 | 2 | 0.953 | 0.943 | 0.952 | 0.940 | 0.915 | 0.946 |
| seeds | 210 | 7 | 3 | 0.875 | 0.869 | 0.932 | 0.929 | 0.896 | 0.902 |
| semeion | 1593 | 256 | 10 | 0.916 | 0.915 | 0.903 | 0.913 | 0.917 | 0.914 |
| soybean | 683 | 35 | 18 | 0.932 | 0.907 | 0.901 | 0.950 | 0.948 | 0.904 |
| spambase | 4601 | 57 | 2 | 0.853 | 0.847 | 0.873 | 0.861 | 0.858 | 0.889 |
| spect | 265 | 22 | 2 | 0.616 | 0.576 | 0.592 | 0.656 | 0.646 | 0.649 |
| spectf | 267 | 44 | 2 | 0.895 | 0.845 | 0.848 | 0.777 | 0.751 | 0.744 |
| statlog-australian-credit | 690 | 14 | 2 | 0.628 | 0.502 | 0.490 | 0.582 | 0.557 | 0.561 |
| statlog-german-credit | 1000 | 24 | 2 | 0.793 | 0.781 | 0.786 | 0.717 | 0.720 | 0.722 |
| statlog-heart | 270 | 13 | 2 | 0.739 | 0.701 | 0.703 | 0.736 | 0.712 | 0.724 |
| statlog-image | 2310 | 18 | 7 | 0.949 | 0.938 | 0.938 | 0.953 | 0.947 | 0.954 |
| statlog-landsat | 6435 | 36 | 6 | 0.890 | 0.876 | 0.874 | 0.884 | 0.881 | 0.877 |
| statlog-shuttle | 58000 | 9 | 7 | 0.999 | 0.823 | 0.998 | 0.999 | 0.905 | 0.998 |
| statlog-vehicle | 846 | 18 | 4 | 0.829 | 0.818 | 0.741 | 0.861 | 0.857 | 0.837 |
| steel-plates | 1941 | 27 | 7 | 0.824 | 0.788 | 0.757 | 0.802 | 0.786 | 0.773 |
| synthetic-control | 600 | 60 | 6 | 0.861 | 0.738 | 0.980 | 0.974 | 0.965 | 0.966 |
| teaching | 151 | 5 | 3 | 0.656 | 0.618 | 0.614 | 0.690 | 0.624 | 0.611 |
| thyroid | 7200 | 21 | 3 | 0.977 | 0.950 | 0.985 | 0.968 | 0.951 | 0.984 |
| tic-tac-toe | 958 | 9 | 2 | 0.990 | 0.875 | 0.961 | 0.985 | 0.951 | 0.975 |
| titanic | 2201 | 3 | 2 | 0.677 | 0.664 | 0.665 | 0.669 | 0.668 | 0.674 |
| trains | 10 | 29 | 2 | 1.000 | 0.667 | 0.833 | 0.958 | 0.625 | 0.625 |
| twonorm | 7400 | 20 | 2 | 0.958 | 0.956 | 0.957 | 0.950 | 0.945 | 0.955 |
| vertebral-column-2clases | 310 | 6 | 2 | 0.871 | 0.836 | 0.846 | 0.851 | 0.847 | 0.858 |
| vertebral-column-3clases | 310 | 6 | 3 | 0.888 | 0.877 | 0.870 | 0.861 | 0.860 | 0.859 |
| wall-following | 5456 | 24 | 4 | 0.894 | 0.879 | 0.889 | 0.909 | 0.900 | 0.919 |
| waveform | 5000 | 21 | 3 | 0.845 | 0.834 | 0.833 | 0.850 | 0.849 | 0.845 |
| waveform-noise | 5000 | 40 | 3 | 0.858 | 0.856 | 0.849 | 0.850 | 0.850 | 0.849 |
| wine | 178 | 13 | 3 | 0.923 | 0.857 | 0.976 | 0.979 | 0.949 | 0.954 |
| wine-quality-red | 1599 | 11 | 6 | 0.657 | 0.629 | 0.622 | 0.640 | 0.626 | 0.611 |
| wine-quality-white | 4898 | 11 | 7 | 0.583 | 0.568 | 0.544 | 0.598 | 0.588 | 0.560 |
| yeast | 1484 | 8 | 10 | 0.675 | 0.667 | 0.648 | 0.690 | 0.687 | 0.645 |
| zoo | 101 | 16 | 7 | 0.980 | 0.966 | 0.965 | 0.989 | 0.967 | 0.963 |

*Continued from previous page.*

