# OpenReview forum: "Detecting Misclassification Errors in Neural Networks with a Gaussian Process Model"
_ICLR.cc/2021/Conference — Reject_

### Official Review · AnonReviewer3 · 2020-10-18
**Interesting problem and results, although the approach seems a bit incremental**

**Rating:** 6
**Confidence:** 3

**Review:**

#######################################################
SUMMARY

This paper introduces RED, a new methodology to produce reliable confidence scores to detect missclassification errors in neural networks. The idea is to combine kernels based on both input and output spaces (as in RIO) to define a (sparse) GP that estimates the residual between the correctness of the original prediction and the maximum class probability. The authors show enhanced performance against other related methods and the ability of RED to detect OOD and adversarial data through the variance of the confidence score.

#####################################################
PROS

1) Obtaining confidence scores for neural network predictions is a timely and very relevant topic for the ICLR community, since it is one of the main limitations of real-world applications of current neural nets.

2) The related literature review is clear and, to the best of my knowledge, the proposed metholody based on Gaussian Processes is novel.

3) The experimental validation of the proposed method on the UCI datasets is strong. It uses a wide range of datasets and several statistical tests, and RED obtains superior performance.

4) The idea of using the variance of the proposed confidence score to identify OOD and adversarial data is interesting and promising.

######################################################
CONS

1) My main concern is that the contribution in RED can be regarded somehow incremental given the RIO approach. It utilizes the same rationale behind RIO, and just adapts the necessary components so that it works in classification. The adaptation of these components is also straightforward: the output kernel now works on several dimensions (instead of the scalar dimension of regression) and the target is now the correctness of the original prediction.

2) The experimental validation focuses on several competitors which can be considered "of the same family" as the proposed approach. Namely, all of them calibrate the predictions of a pre-trained neural network. I think it would be interesting to also compare to a different "family" of methods. For instance, (Functional) Bayesian Neural Networks are meant to obtain calibrated predictions by leveraging epistemic uncertainty (that coming from the model parameters).

3) I do not fully understand the relevance of the experiment with the large deep learning architecture given by the VGG16 model. Since the proposed method works on the pre-trained neural network, my understanding is that the complexity of the neural network itself is not relevant for the performance of the proposed approach. Also, in this experiment I miss several independent runs to assess the results variability.

####################################
Additional questions/feedback:

1) In the second paragraph of section 4.2., there seems to be a typo when reporting the margin. It is said 0.42 and 0.55 for ConfidNet and RED respectively, but I think it should be 0.042 and 0.055 by looking at Table 3.

2) It is not entirely clear to me why the process described in section 4.3. (second paragraph) produces proper OOD and adversarial data. For instance, some of the intended OOD data could be similar to training data (specially because the latter is being normalized to mean 0 and std 1). And similarly for the adversarial case. I think this could be better explained.

3) When it comes to real practice, a key decision is to set a threshold on the confidence score to decide what instances should be supervised by an expert. Is there any recommendation on this?

#######################################
AFTER REBUTTAL

The new baselines added make the experimental validation more convincing. Therefore, I have raised my rating to 6 (Marginally above the acceptance threshold). However, I still believe that the contribution is incremental, and I think the paper would gain in terms of novelty if it focused more on the detection of OOD data and adversarial attacks (which right now is more like a preliminary test).

---

> ### Author Response · Authors · 2020-11-24
> **Responses to Reviewer 3 (1 out of 2)**
>
> Thanks for your encouraging comments and constructive suggestions. We have added more baselines as suggested to make the experimental evaluations more comprehensive. Please see our detailed responses below:
> ***
> Comment 1: “My main concern is that the contribution in RED can be regarded somehow incremental given the RIO approach. It utilizes the same rationale behind RIO, and just adapts the necessary components so that it works in classification. The adaptation of these components is also straightforward: the output kernel now works on several dimensions (instead of the scalar dimension of regression) and the target is now the correctness of the original prediction.”
>
> A1: It is notable that the original RIO is only limited to standard regression problems, but RED extends it to solve an important yet underexplored direction in classification domain: detecting misclassification errors. The main contribution of RED is to capture the connection between these two seemingly unrelated topics (a regression method and error detection in classification). We believe the implementation is natural and compelling, and the experimental results (in the revised version RED is compared to 9 approaches in 100+ datasets) show that the proposed approach indeed works significantly better than existing approaches in this new problem.
> ***
> Comment 2: “The experimental validation focuses on several competitors which can be considered "of the same family" as the proposed approach. Namely, all of them calibrate the predictions of a pre-trained neural network. I think it would be interesting to also compare to a different "family" of methods. For instance, (Functional) Bayesian Neural Networks are meant to obtain calibrated predictions by leveraging epistemic uncertainty (that coming from the model parameters).”
>
> A2. We chose these competitors because they are state-of-the-art in the same problem as RED is intended to solve: providing a quantitative metric to detect misclassification errors of a pre-trained neural network. We agree, however, that including more traditional approaches makes the evaluations more convincing. Therefore, as suggested, we added a comparison to Bayesian Neural Networks (BNN). More specifically, we trained a BNN and applied RED on top of it to see whether RED is able to provide better confidence scores in misclassification detection compared to the internal confidence scores returned by BNN. The comparisons were run on all 125 UCI datasets, and the results show that RED outperforms BNN significantly (see Table 3 in the revised manuscript). In addition, four more approaches were included for comparison, as suggested by other reviewers: entropy of the softmax outputs, MC-dropout [1], original SVGP [2], and DNGO[3]. RED significantly outperforms all of them (see Table 1, 2, 3 and 4 in the revised manuscript), thus strengthening the conclusions. Thanks for the suggestion!
> ***
> Comment 3: “I do not fully understand the relevance of the experiment with the large deep learning architecture given by the VGG16 model. Since the proposed method works on the pre-trained neural network, my understanding is that the complexity of the neural network itself is not relevant for the performance of the proposed approach. Also, in this experiment I miss several independent runs to assess the results variability.”
>
> A3: Indeed  this experiment is not strictly necessary given the results on the 125 UCI datasets. The reason it was included was to verify empirically that RED also works well on more complex vision tasks with large, deep architectures. Since the submission we have improved the training pipeline for the VGG16 model (it now achieves state-of-the-art accuracy), and ran 10 independent runs to verify that the results are reliable. Statistical tests were run against the original and newly added baselines (MC-dropout and BNN is independent of VGG16 model, so they are not included in the  CIFAR-10 experiment). The results verify that RED significantly outperforms all counterparts (see Table 4 in the revised manuscript). These new results are included in the revision, strengthening the conclusions.

---

> > ### Author Response · Authors · 2020-11-24
> > **Responses to Reviewer 3 (2 out of 2)**
> >
> > Comment 4: “In the second paragraph of section 4.2., there seems to be a typo when reporting the margin. It is said 0.42 and 0.55 for ConfidNet and RED respectively, but I think it should be 0.042 and 0.055 by looking at Table 3.”
> >
> > A4: Thanks for pointing out this typo, it should be 0.042 and 0.055. We have updated the results and descriptions.
> > ***
> > Comment 5: “It is not entirely clear to me why the process described in section 4.3. (second paragraph) produces proper OOD and adversarial data. For instance, some of the intended OOD data could be similar to training data (specially because the latter is being normalized to mean 0 and std 1). And similarly for the adversarial case. I think this could be better explained.”
> >
> > A5: While the main focus of this work is on misclassification detection, This case study is included to highlight a significant and promising avenue for future work. It is a preliminary test of RED on detecting OOD and adversarial data under the most difficult scenaria, i.e., when the OOD data is similar to the original data, and the adversarial samples have almost identical input features with the original data but generate different predictions. The results are promising, suggesting that RED provides a new foundation to this difficult problem, hopefully inspiring further work in this area.  We have added a more thorough explanation to section 4.4 (previously section 4.3) in the revised version to clarify this point.
> > ***
> > Comment 6: “When it comes to real practice, a key decision is to set a threshold on the confidence score to decide what instances should be supervised by an expert. Is there any recommendation on this?”
> >
> > A6: Yes, our suggestion is to use a validation dataset to check how the tradeoffs between precision and recall (in terms of misclassification detection) change over different thresholds. The users can then decide the threshold based on their preference on the precision-recall tradeoff. There is also some literature discussing the threshold choice [4][5][6] (references to it are included  in “Related Work” section). This point is discussed more extensively in “Discussion and Future Work” section in the revised version.
> > ***
> > [1] Yarin Gal and Zoubin Ghahramani. “Dropout as a bayesian approximation: Representing model uncertainty in deep learning”. In Proceedings of the 33rd International Conference on International Conference on Machine Learning - Volume 48, ICML’16
> > [2] James Hensman, Nicol`o Fusi, and Neil D. Lawrence. “Gaussian processes for big data”. In Proceedings of the Twenty-Ninth Conference on Uncertainty in Artificial Intelligence, UAI’13
> > [3] Jasper Snoek, Oren Rippel, Kevin Swersky, Ryan Kiros, Nadathur Satish, Narayanan Sundaram, Md. Mostofa Ali Patwary, Prabhat Prabhat, and Ryan P. Adams. 2015. “Scalable Bayesian optimization using deep neural networks”. In Proceedings of the 32nd International Conference on International Conference on Machine Learning - Volume 37 (ICML'15)
> > [4] Bernard Dubuisson and Mylne Masson. A statistical decision rule with incomplete knowledge about classes. Pattern Recognition, 26(1):155 – 165, 1993.
> > [5] Carla M. Santos-Pereira and Ana M. Pires. On optimal reject rules and roc curves. Pattern Recogn. Lett., 26(7):943952, May 2005.
> > [6] C. Chow. On optimum recognition error and reject tradeoff. IEEE Trans. Inf. Theor., 16(1):4146, September 2006.

---

### Official Review · AnonReviewer4 · 2020-10-27
**Adding confidence score to NN classifiers without retraining or modifying the model**

**Rating:** 6
**Confidence:** 2

**Review:**

This paper solves an interesting problem of predicting uncertainty in NN without re-raining/modifying the existing NN. The authors propose a framework to calculate a confidence score for detecting misclassification errors by calibrating the NN classifier’s confidence scores and estimates uncertainty around the calibrated scores using Gaussian processes. This framework is called RED (Residual i/o Error Detection).

This paper is also technically sound and to the best of my knowledge is novel and relevant to the community.

It would be good to apply SVGP directly to some of these datasets and compare the results against NN+SVGP results.

You use the term “calibrated” confidence score/prediction. Could you explain what do you mean by calibrated?

I find the presentation of results very confusing. For example, in Table 1, AP-Error is smallest for the RED method and in Table 3 AP-error is the largest for the RED method. In both cases, it is mentioned that the RED method outperforms other methods.

You mentioned ConfidNet outperformed the MCP baseline by a margin of 0.42. I do not see this number on the table.

It would be good if the authors could mention in the paper what is RIO short for.

You mentioned that you need to extend the kernel to multiple output kernel. Could you explain a bit more about that and how you build it?

---

> ### Author Response · Authors · 2020-11-24
> **Responses to Reviewer 4**
>
> Thanks for your positive comment. As suggested, we have included more comparisons to other approaches in the revised version. Please see detailed responses below:
> ***
> Comment 1: “It would be good to apply SVGP directly to some of these datasets and compare the results against NN+SVGP results.”
>
> A1: As suggested, we applied SVGP to all 125 UCI datasets and CIFAR-10. More specifically, SVGP was used to predict whether the original prediction is correct or not. Based on these new experimental results,  RED (NN+SVGP together) performs significantly better than SVGP alone in the misclassification detection task (see Table 1, 2 and 4 in the revised manuscript). As suggested by other reviewers, four other comparisons were also included: entropy of the softmax outputs, Bayesian Neural networks,  MC-dropout [1], and DNGO[2]. Experimental results on 125 UCI datasets and CIFAR-10 confirms that RED performs significantly better than all of these approaches (see Table 1, 2, 3 and 4 in the revised manuscript), significantly strengthening the conclusions.
> ***
> Comment 2: “You use the term “calibrated” confidence score/prediction. Could you explain what do you mean by calibrated?”
>
> A2: “Calibrated”means that RED is applied on top of the internal confidence score returned by the original classifier, e.g., the maximum softmax output. RED thus estimates the residuals between the originally predicted confidence score and target confidence score (1 for correct prediction, 0 for incorrect prediction). After that, RED adds the estimated residual back to the original confidence score, and generates a new confidence score in order to detect misclassifications. This new confidence score returned by RED is the “calibrated” version of the original confidence score. Note that this “calibrated” confidence score is only used for misclassification detection. It does not affect the outputs or prediction accuracy of the original classifier. The introduction section has been revised to clarify this point.
> ***
> Comment 3: “I find the presentation of results very confusing. For example, in Table 1, AP-Error is smallest for the RED method and in Table 3 AP-error is the largest for the RED method. In both cases, it is mentioned that the RED method outperforms other methods.”
>
> A3: The results in Table 1 present  the mean rank of the algorithm in 125 UCI datasets, in terms of different metrics like AP-Error, so the smaller the better. The results in Table 3 instead show the absolute values of different metrics like AP-Error in CIFAR-10 dataset, and larger values are better. We have made this concern clear in the revised version.
> ***
> Comment 4: “You mentioned ConfidNet outperformed the MCP baseline by a margin of 0.42. I do not see this number on the table.”
>
> A4: We are sorry for the typo. It should be 0.042. We have updated the experimental results and descriptions accordingly.
> ***
> Comment 5: “It would be good if the authors could mention in the paper what is RIO short for.”
>
> A5: Thanks for the suggestion. We have added a note in the revised version to specify that RIO stands for Residual Input/Output.
> ***
> Comment 6: “You mentioned that you need to extend the kernel to multiple output kernel. Could you explain a bit more about that and how you build it?”
>
> A6: The output kernel in the original RIO model is limited to single-output regression problems. However, for classification problems, the original model usually has multiple outputs, each one corresponding to one class. In RED, this output kernel is extended to multiple outputs of the original classifier. Utilizing information from all outputs should be beneficial in misclassification detection compared to simply considering the single output of the predicted class. To build this kernel, the calculation of covariances (based on GP kernel) is extended from single dimension to multiple dimensions. The feature for output kernel is thus a vector containing multiple softmax outputs (one for each class). A description of this process is included in both the texts and Algorithm1 in section 3.3.
> ***
>
> [1] Yarin Gal and Zoubin Ghahramani. “Dropout as a bayesian approximation: Representing model uncertainty in deep learning”. In Proceedings of the 33rd International Conference on International Conference on Machine Learning - Volume 48, ICML’16
> [2] Jasper Snoek, Oren Rippel, Kevin Swersky, Ryan Kiros, Nadathur Satish, Narayanan Sundaram, Md. Mostofa Ali Patwary, Prabhat Prabhat, and Ryan P. Adams. 2015. “Scalable Bayesian optimization using deep neural networks”. In Proceedings of the 32nd International Conference on International Conference on Machine Learning - Volume 37 (ICML'15)

---

### Official Review · AnonReviewer2 · 2020-10-29
**Well-performing, simple to implement method for classification error detection; limited set of baselines may not establish generality**

**Rating:** 6
**Confidence:** 3

**Review:**

Update: Following the authors' clarifications and additional experimental work, I'm increasing my rating to 6.

This paper proposes RED, a framework for detecting misclassification errors, based on regression of target confidence scores and application of a Gaussian process for uncertainty in predicted confidence scores. It builds upon RIO, a framework for predicting residuals of regression models and their uncertainties using GPs. Compared with other confidence metrics, RED aims for greater separability between correct and incorrect predictions.

The method is straightforward to implement and performs well against the baselines considered on classification tasks for 125 UCI datasets. However, I question whether the baselines are sufficient; it is not demonstrated whether RED would outperform other confidence scoring and OOD detection methods mentioned in the related work section, such as temperature scaling (or the related method ODIN, proposed in Liang, S., Li, Y., and Srikant, R., 2017. Enhancing the reliability of out-of-distribution image detection in neural networks.) or simply the entropy of the softmax predictions. Unless there is a good justification for the limited set of baselines, I believe the paper's claims to generality are limited.

Additionally, for the OOD detection results shown in Figure 3, why were AUROC and AUPRC not reported? While the scatterplots show separability of OOD data visually, these metrics (used elsewhere in the paper) would give a better indication of performance (and again, I think a greater range of baselines and tasks would be necessary to make any firm claims about OOD detection).

---

> ### Author Response · Authors · 2020-11-24
> **Responses to Reviewer 2 (1 out of 2)**
>
> Thanks for your constructive suggestions. We have added experimental comparisons to several more approaches in the revised version. Please see detailed responses below:
> ***
> Comment 1: “However, I question whether the baselines are sufficient; it is not demonstrated whether RED would outperform other confidence scoring and OOD detection methods mentioned in the related work section, such as temperature scaling (or the related method ODIN, proposed in Liang, S., Li, Y., and Srikant, R., 2017. Enhancing the reliability of out-of-distribution image detection in neural networks.) or simply the entropy of the softmax predictions. Unless there is a good justification for the limited set of baselines, I believe the paper's claims to generality are limited.”
>
> A1: It is important to clarify that the focus of this work is on misclassification detection (an underexplored [1] and challenging [2] new area), instead of OOD or adversarial detection. We have considered all the approaches mentioned in the related work section from this perspective, however, most of them do not apply to the misclassification detection problem. Taking temperature scaling as an example, the main idea is to scale all the logit outputs by a scalar T, and the same T is applied to all predictions. As a result, the relative ranking of the predictions are still preserved after re-scaling, so it makes no difference for misclassification detection compared to using original softmax outputs (the MCP baseline in our experiments). It is notable that approaches like temperature scaling focus on reducing the difference between reported class probability and true accuracy, and the separability between correct and incorrect predictions is not improved. In contrast, RED aims at deriving a score that can differentiate incorrect predictions from correct ones. This point is emphasized in the “related work” section. Similarly, ODIN is particularly designed for OOD detection in image tasks, which is a different problem from misclassification detection, as is now clarified in the related work section.
>
> However, to put the results in context, we added the entropy of the softmax predictions as a baseline (it was not originally included because according to literature [1], its performance is similar to maximum class probability baseline, which was already included). Experiments on 125 UCI datasets and CIFAR-10 show that RED significantly outperforms the entropy baseline (see Table 1, 2 and 4 in the revised manuscript). In addition, four more baselines were included for comparison as suggested by other reviewers: Bayesian Neural Networks, MC-dropout [3], original SVGP [4], and DNGO[5]. According to experimental results (see Table 1, 2, 3 and 4 in the revised manuscript), RED performs significantly better than all of them, which, we believe convincingly demonstrates the value of the approach.
> ***
> Comment 2: “Additionally, for the OOD detection results shown in Figure 3, why were AUROC and AUPRC not reported? While the scatterplots show separability of OOD data visually, these metrics (used elsewhere in the paper) would give a better indication of performance (and again, I think a greater range of baselines and tasks would be necessary to make any firm claims about OOD detection).”
>
> A2: Because the focus of the paper is on misclassification detection, the case study in section 4.4 (previously section 4.3) is not yet intended to make substantial claims about the superiority of RED in detecting OOD examples. However, we decided to show the results in Figure 3 because this preliminary finding shows an intriguing possibility for future work: Since RED provides both the mean and variance of the confidence scores, it is possible to construct a 2-dimensional space for error detection (as shown in Figure 3). This space is different from the 1-dimensional detection space in traditional approaches, which only provide a single number for the confidence score. With this new 2D space, it is possible to not only detect the errors, but also differentiate between different types of errors, i.e. separate correct, incorrect (misclassification), OOD, and adversarial samples. Traditional metrics like AUROC and AUPRC are only designed for binary classification problems, but this classification problem has four classes in total. Therefore, AUROC and AUPRC cannot be directly applied, and new metrics will need to be developed for this new domain (differentiating different types of errors). This case study is intended to show the potential of further extending RED to broader error detection tasks, and we hope it can inspire other researchers in their future work. We have placed this case study at the end of the experiments section to avoid distraction from the main topic, and included discussion to clarify its purpose. We also included discussions in the future work section to point out this new direction.

---

> > ### Author Response · Authors · 2020-11-24
> > **Responses to Reviewer 2 (2 out of 2)**
> >
> > references
> > ***
> > [1] Dan Hendrycks and Kevin Gimpel. “A baseline for detecting misclassified and out-of-distribution examples in neural networks”. Proceedings of International Conference on Learning Representations, 2017.
> >
> > [2] Jonathan Aigrain and Marcin Detyniecki. “Detecting adversarial examples and other misclassifications in neural networks by introspection”. CoRR, abs/1905.09186, 2019.
> >
> > [3] Yarin Gal and Zoubin Ghahramani. “Dropout as a bayesian approximation: Representing model uncertainty in deep learning”. In Proceedings of the 33rd International Conference on International Conference on Machine Learning - Volume 48, ICML’16
> >
> > [4] James Hensman, Nicol`o Fusi, and Neil D. Lawrence. “Gaussian processes for big data”. In Proceedings of the Twenty-Ninth Conference on Uncertainty in Artificial Intelligence, UAI’13
> >
> > [5] Jasper Snoek, Oren Rippel, Kevin Swersky, Ryan Kiros, Nadathur Satish, Narayanan Sundaram, Md. Mostofa Ali Patwary, Prabhat Prabhat, and Ryan P. Adams. 2015. “Scalable Bayesian optimization using deep neural networks”. In Proceedings of the 32nd International Conference on International Conference on Machine Learning - Volume 37 (ICML'15)

---

### Official Review · AnonReviewer1 · 2020-10-31
**Key comparison methods missing**

**Rating:** 6
**Confidence:** 3

**Review:**

In this paper, their goal is to improve calibration and accuracy by augmenting a classification model with a GP. They base their model off RIO (ICLR 2020) which targets regression problems and tries to predict the residual between predicted value and true value. They propose a model, RED, which instead tries to predict the residual between the predicted confidence score for the true class and 1 — the true class target confidence score using a GP. They show strong improvements over the methods they compare to for 125 UCI datasets and CIFAR-10 dataset.

I find the approach interesting though the novelty is incremental over the RIO paper. My main concern is that I think some additional methods need to be compared with. For example [1] uses a bayesian last layer which is something that should be compared with. Using an ensemble of single layer NNs for the last layer or using MC-dropout at test time (which is known to approximate Bayesian inference under certain conditions) would also be interesting.

[1] “Scalable Bayesian Optimization Using Deep Neural Networks” by Snoek et al.
[2] “Dropout as a Bayesian Approximation: Representing Model Uncertainty in Deep Learning” by Gal et al.

Edit: Based on the author response in terms of adding additional experiments, I'm raising my score to a 6.

---

> ### Author Response · Authors · 2020-11-24
> **Responses to Reviewer1**
>
> Thank you for constructive suggestions. We have carefully considered your comments, and added the suggested experiments. Please see detailed responses below:
> ***
> Comment 1: “In this paper, their goal is to improve calibration and accuracy by augmenting a classification model with a GP.”
>
> A1: We would like to emphasize that the proposed method (RED) does not change the prediction accuracy of the original classification model. Instead, RED is a supporting tool that can provide a quantitative metric for detecting misclassification errors of the original classification model. We have revised the Introduction section to make this point clear.
> ***
> Comment 2: “They propose a model, RED, which instead tries to predict the residual between the predicted confidence score for the true class and 1 — the true class target confidence score using a GP.”
>
> A2: Actually, “the predicted confidence score for the true class” should be “the predicted confidence score for the predicted class”; this score corresponds to the maximum class probability returned by the original classification model (the predicted class may not necessarily be the true class).
> ***
> Comment 3: “My main concern is that I think some additional methods need to be compared with. For example [1] uses a bayesian last layer which is something that should be compared with. Using an ensemble of single layer NNs for the last layer or using MC-dropout at test time (which is known to approximate Bayesian inference under certain conditions) would also be interesting.”
>
> A3: Thanks for the suggestion---we added several new comparisons in the revised paper, and they strengthen the conclusions significantly. First, a comparison with the approach in your reference [1] was included. Although the original approach [1] models the surrogate function in Bayesian Optimization setup, we managed to extend it to error detection problems by adding a Bayesian linear regression layer after the logits layer of the original classification model to predict whether an original prediction is correct or not. This approach was tested over all 125 UCI datasets and CIFAR-10 (with VGG-16 architecture). RED outperforms this approach by a significant margin (see Table 1, 2 and 4 in the revised manuscript). Second, a comparison with the MC-dropout approach [2] was included. More specifically, the original standard NN classifier (without dropout layer) was replaced with an NN classifier (adding dropout layers after each hidden layer) with dropout running in both train and test time. RED was then applied on top of the MC-dropout NN classifiers to see whether RED is able to provide better performance in error detection. Experiments were again run on all 125 UCI datasets (the modified MC-dropout NN classifiers does not directly apply to CIFAR-10), and again RED significantly outperformed MC-dropout (see Table 3 in the revised manuscript). Third, comparisons were added with Bayesian Neural Networks, original SVGP, and entropy baseline as suggested by other reviewers. Please check Table 1, 2, 3, 4 in the revised manuscript for the newly added results. Based on the experiments, RED performs significantly better than all of them, supporting the conclusions of the paper strongly.
> ***
> Comment 4: “I find the approach interesting though the novelty is incremental over the RIO paper.”
>
> A4: Actually, the original RIO approach applies only to  standard regression problems; it cannot be directly applied to classification problems. Moreover, detecting misclassification errors is a distinctly different problem from improving the accuracy/calibrating the predictions, on which most previous works have been done. This problem is still underexplored [3] and challenging [4], yet it is critical for improving AI safety in real-world applications. Our main innovation is to capture the connection between a method for regression (RIO) and a new problem in classification, and successfully extend the method to this new problem. The experimental results (now compared with 9 approaches in 100+ datasets) show that the RED indeed works significantly better than state-of-the-art approaches. We have revised the introduction to make this framing clear.
> ***
> [1] “Scalable Bayesian Optimization Using Deep Neural Networks” by Snoek et al.
> [2] “Dropout as a Bayesian Approximation: Representing Model Uncertainty in Deep Learning” by Gal et al.
> [3] Dan Hendrycks and Kevin Gimpel. “A baseline for detecting misclassified and out-of-distribution examples in neural networks”. Proceedings of International Conference on Learning Representations, 2017.
> [4] Jonathan Aigrain and Marcin Detyniecki. “Detecting adversarial examples and other misclassifications in neural networks by introspection”. CoRR, abs/1905.09186, 2019.

---

### Author Response · Authors · 2020-11-24
**General response to all reviewers**

We want to thank all the reviewers for their time and effort in evaluating our work. Thank you for your constructive comments. We have carefully read and considered all your suggestions. A common concern by all reviewers is the need for including more baseline approaches. As suggested, we have added five more baseline approaches, and tested all of them on all 125 UCI datasets and CIFAR-10 (when applicable). RED significantly outperforms all these approaches, significantly strengthening the conclusions of the paper. Below is a list briefly summarizing the main revisions:
- new experiments
   - comparison with DNGO [1]
   - comparison with original SVGP [2]
   - comparison with entropy of softmax outputs [3]
   - comparison with MC-dropout [4]
   - comparison with Bayesian Neural Networks [5]
   - more comprehensive evaluation on CIFAR-10 datasets with VGG16 models

- new discussions
   - clarifying the purpose and contribution of the approach in section 1
   - clarifying the intention of the preliminary case study in section 4.4 (previously section 4.3)
   - suggestion on how to choose the warning threshold in section 5

We have also responded to the specific comments of each reviewer by directly replying to their original reviews.
***
[1] Jasper Snoek, Oren Rippel, Kevin Swersky, Ryan Kiros, Nadathur Satish, Narayanan Sundaram, Md. Mostofa Ali Patwary, Prabhat Prabhat, and Ryan P. Adams. 2015. “Scalable Bayesian optimization using deep neural networks”. In Proceedings of the 32nd International Conference on International Conference on Machine Learning - Volume 37 (ICML'15)

[2] James Hensman, Nicol`o Fusi, and Neil D. Lawrence. “Gaussian processes for big data”. In Proceedings of the Twenty-Ninth Conference on Uncertainty in Artificial Intelligence, UAI’13

[3] Jacob Steinhardt and Percy Liang. “Unsupervised risk estimation using only conditional independence structure”. In Proceedings of the 30th International Conference on Neural Information Processing Systems, NIPS’16

[4] Yarin Gal and Zoubin Ghahramani. “Dropout as a bayesian approximation: Representing model uncertainty in deep learning”. In Proceedings of the 33rd International Conference on International Conference on Machine Learning - Volume 48, ICML’16

[5] Yeming Wen, Paul Vicol, Jimmy Ba, Dustin Tran, and Roger Grosse. “Flipout: Efficient pseudo independent weight perturbations on mini-batches”. In International Conference on Learning Representations, 2018.

---

### Author Response · Authors · 2021-01-15
**Reply to program chair**

We would like to thank the program chair for spending time reading our manuscript and providing suggestions. We also want to clarify some points raised by the program chair, in order to avoid possible confusions:

1. Regarding the program chair’s comment “In the tables, it can be hinted that this might be happening, as about 80% of the cases MCP and RED are indistinguishable in the AUROC values.” Actually we have already run two statistical tests on the AUROC values, and the results are already summarized in Table 2 in the original manuscript. RED is statistically significantly better than MCP in ~50% of the datasets, not 20% only. Moreover, AUROC is sometimes less informative [1] and not ideal when the positive class and negative class have greatly differing base rates [2] (this happens when the base classifier has high prediction accuracy so we only have few misclassified examples). Since the focus of this work is on error detection, we stated in the paper that we would focus more on AP-error and AUPR-error. In terms of AP-error and AUPR-error, RED indeed significantly outperforms MCP in most datasets.

2. Actually we have already added two new types of base classifiers during the rebuttal (please see section 4.2), and the experimental results validate the robustness of our approach (summarized results in Table 3). A deep NN model with a large-scale dataset was also included in section 4.3 in the original manuscript.

3. Regarding program chair’s recommendation to compare with methods in paper: https://arxiv.org/abs/1706.04599, actually we have already cited this paper in section 2, and made extensive discussions about the difference between their work and ours: These methods focus on reducing the difference between reported class probability and true accuracy, and generally the rankings of samples are preserved after calibration. As a result, the separability between correct and incorrect predictions is not improved. In contrast, RED aims at deriving a score that can differentiate incorrect predictions from correct ones better. RED is solving a totally different problem. Directly applying the methods in https://arxiv.org/abs/1706.04599 to error detection is the same as using MCP baseline, which we already included in the current manuscript.

4. Regarding the program chair’s comment “using a GP might be an overkill”, we are using a sparse GP variant called SVGP, which has much lower computational complexity compared to original GP. Moreover, as discussed in the original manuscript, the main reason for using a probabilistic model like GP is to further provide uncertainty information regarding the returned “confidence score”. This can overcome the limitations of original NN classifiers: the standard NNs do not provide any information regarding uncertainty of their inherent confidence scores, resulting in misleading “overconfident” predictions. This is critical to the safety and reliability of real-world AI models.

[1] Chris Manning and Hinrich Schütze. “Foundations of Statistical Natural Language Processing”. MIT Press, 1999.

[2] Dan Hendrycks and Kevin Gimpel. “A baseline for detecting misclassified and out-of-distribution examples in neural networks”. Proceedings of International Conference on Learning Representations, 2017.

---

### Decision · Program_Chairs · 2021-01-07
**Final Decision**

**Decision:**

Reject

**Comment:**

In this paper, the authors use a GP classifier to detect if the output of a NN classifier has been decided correctly. The GP takes as input the original input vector x and the output of the NN, i.e. the calibrated posterior probabilities given by the NN. It uses that as an input vector for the GP classifier to decide if the sample was correctly decided. The output of the GP will serve as confidence in the output of the NN. The results are comparable/superior with the state-of-the-art and the authors have repeated the experiments with over 125 different datasets. The reviewers of this paper were all cautiously positive about the paper, but all of them pointed towards the reduced novelty of the paper. Also, none of the reviewers were willing to champion this paper as a must-have at ICLR 2021.

For my reading of the paper, I would tend to agree with the reviewers’ comments. Also, I find that using the same NN, rather shallow, with the same configuration for all the datasets seems rather limited. Given that this method is independent of the underlying classifier and that the databases used are low dimensions and a low number of training examples, I would have liked to see what a random forest or a GP can accomplish. Also, I would have used bigger NNs that can be trained to overfit the sigmoid outputs for classification of higher accuracy. I believe that having a diversity of underlying classifiers is more relevant than having 125 datasets. We need to find the best classifier or ensemble and then apply the different mechanisms for estimating if the output is the correct one. Otherwise, the proposed method might only be workable for this specific NN configuration. In the tables, it can be hinted that this might be happening, as about 80% of the cases MCP and RED are indistinguishable in the AUROC values.

Also, for all of these datasets a GP could be used as an underlying classifier, and given the premises of this paper, the authors could check how well calibrate a GP classifier is. Also, there has been considerable work on calibrating NNs when they are trained to overfit. Comparing with those methods should be straightforward, as they provide more information than just a confidence score. This is probably the most influential paper: https://arxiv.org/abs/1706.04599 (1000+ references), but there are some recent papers too.

Finally, if the goal is to use a GP to detect if the classification done by the NNs is accurate, using a GP might be an overkill, as the complexity of the GP, especially for large datasets might end up being larger than the underlying classifier.